# PCoreSet: Effective Active Learning through Knowledge Distillation from Vision-Langauge Models

## Abstract

Knowledge distillation (KD) is a widely used framework for training compact, task-specific models by transferring the knowledge from teacher models. However, its application to *active learning* (AL), which aims to minimize annotation costs through iterative sample selection, remains underexplored. This gap stems from the fact that KD typically assumes access to sufficient labeled data, whereas AL operates in data-scarce scenarios where task-specific teacher models are often unavailable. In this paper, we first introduce **ActiveKD**, a framework that integrates AL with KD by leveraging the zero- and few-shot capabilities of large vision-language models (VLMs). A key aspect of ActiveKD is the *structured prediction bias* of VLMs—*i.e.*, their predictions form clusters in the probability space. We regard this structure as an inductive bias of the teacher model, capturing generalizable output patterns beneficial to student learning. To exploit this bias, we propose **P**robabilistic **CoreSet** (**PCoreSet**), a selection strategy that maximizes coverage in the probability space rather than the feature space. **PCoreSet** strategically selects probabilistically diverse unlabeled samples, facilitating more efficient transfer of teacher knowledge under limited annotation budgets. Extensive evaluations on 11 datasets show that ActiveKD consistently improves performance across selection methods (*e.g.*, +29.07% on ImageNet, averaged over methods). Under ActiveKD, **PCoreSet** ranks first in 64/73 settings ($\approx$87.7%) across 5 student and 3 teacher networks, always achieving the best performance except for first 2 AL rounds.

## 1 Introduction

Recent advances in deep neural networks have focused on developing powerful generalist models capable of solving diverse tasks (Achiam et al., 2023; OpenAI, 2025; Liu et al., 2023; 2024b) or creating robust transferable models through intensive pretraining (Koroteev, 2021; Radford et al., 2019; He et al., 2016; Dosovitskiy et al., 2020; Radford et al., 2021; Jia et al., 2021). However, real-world applications often necessitate training compact task-specific models, with a significant challenge of obtaining task-specific labeled data due to high annotation costs. Active learning (AL) addresses this by iteratively selecting the most informative samples for oracle annotation (Ren et al., 2021; Li et al., 2024), particularly in pool-based scenarios where informative samples can be identified from large unlabeled data. The availability of unlabeled data allows semi-supervised learning (SSL) to be applied to AL settings during training (Assran et al., 2021; Cai et al., 2022; Zheng et al., 2023; Kang et al., 2025), accompanying researches at the intersection of AL and SSL (Gao et al., 2020; Lim et al., 2023; Rangnekar et al., 2023; Singh et al., 2024).

Knowledge distillation (KD), meanwhile, is a widely used framework for training models by transferring knowledge from large pretrained teacher models (Hinton et al., 2015). Beyond its original purpose, KD can be naturally integrated into SSL framework as teacher models can provide its knowledge in the form of soft labels for unlabeled data (Chen et al., 2020b; He et al., 2021; Du et al., 2023; Yang et al., 2025). However, despite its effectiveness, the application of KD in AL remains *underexplored*. One major reason is the mismatch in their assumptions: AL assumes that labels must be manually acquired through selective annotation, while KD typically assumes the existence of powerful task-specific teacher models trained on sufficient labeled data. However, considering the data-scarce settings common in AL, such teacher models are rarely available or practical to obtain.

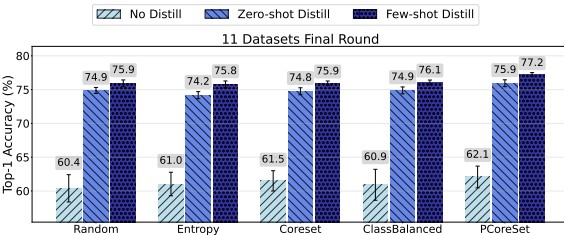
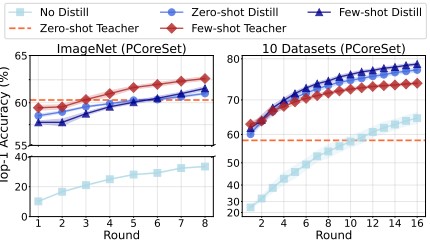
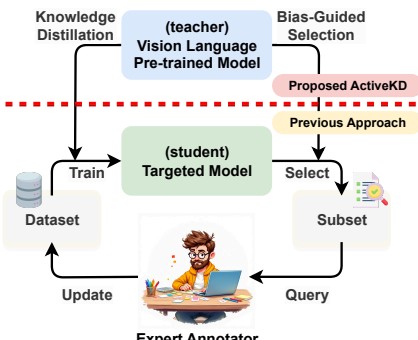

Figure 1: **(Left):** The proposed **ActiveKD framework consistently improves the final-round accuracy of all selection methods**, while **PCoreSet further outperforms baselines** when combined with ActiveKD. **(Right):** ActiveKD consistently improves **PCoreSet** across active learning rounds (No Distill vs. Zero-shot Distill), with further gains when using few-shot teachers (Few-shot Distill). The proposed ActiveKD + **PCoreSet** even surpasses VLM teachers, outperforming **zero-shot teachers** on the average of 10 datasets after the **first AL round** and **few-shot teachers** after the **second**.

Recently, vision-language models (VLMs) have emerged as powerful teachers in KD frameworks, as their rich representations—acquired through large-scale, unsupervised pretraining—enable effective knowledge transfer across both modalities and tasks (Vemulapalli et al., 2023; Wu et al., 2024; Mistretta et al., 2024). Recent work (Kang et al., 2025) has further demonstrated that transferring the knowledge of VLMs into task-specific models is effective in label-scarce scenarios, such as few-shot or low-shot semi-supervised settings. Motivated by these, we introduce **ActiveKD**, a framework that integrates AL with KD by leveraging the zero- and few-shot capabilities of VLMs. Specifically, in each AL round, we train a task-specific student model using both labeled data acquired up to that round and unlabeled data with soft predictions generated by the VLM teacher, as illustrated in Fig. 2.

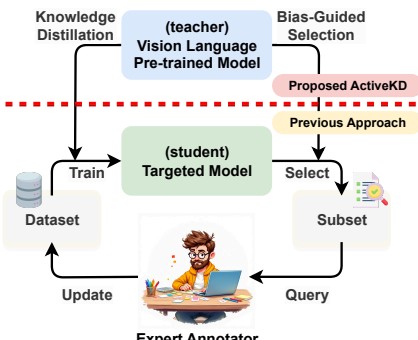

Figure 2: An overview of ActiveKD.

In ActiveKD settings, we identify an aspect of VLM teachers: they exhibit *structured prediction biases* acquired during pretraining and shaped by language prompts (Bang et al., 2024), resulting in predictions that form distinct clusters in the probability space as illustrated in Fig. 3. We observe that this bias propagates through distillation, influencing both the active selection mechanism and student model performance. Rather than viewing this as a limitation, we interpret this structure as an *inductive bias* of the teacher model—capturing generalizable patterns in the output space that, when complemented by labeled data, can benefit student learning. To exploit this, we propose a *simple and effective* method, **P**robabilistic **CoreSet** (**PCoreSet**), which selects coresets in the probability space rather than the feature space (Sener & Savarese, 2017). Specifically, it maximizes diversity in the categorical probability space by targeting underrepresented regions with limited budgets.

To evaluate the empirical effectiveness of **ActiveKD** and **PCoreSet**, we conduct extensive experiments on 11 datasets spanning diverse domains and tasks detailed in §C. As summarized in Fig. 1, **ActiveKD consistently improves** performance across selection methods on all datasets (*e.g.*, **+29.07%** on ImageNet, averaged over methods), with further gains when using few-shot teachers. Moreover, under ActiveKD, **PCoreSet outperforms alternative selection strategies**, ranking first in **64/73** settings ($\approx$ **87.67%**) across 5 student and 3 teacher architectures, with either 8 or 16 rounds.

Our contributions and findings are summarized as follows:

- We first introduce *active knowledge distillation* (**ActiveKD**), a framework that integrates knowledge distillation into active learning by leveraging the zero- and few-shot capabilities of vision-language models to train efficient task-specific models with limited labeled data.

- We identify *structured prediction biases* in VLM teachers that are propagated to the student model after KD. We thus propose **P**robabilistic **CoreSet** (**PCoreSet**), which utilizes this *inductive bias* to select samples in the underrepresented regions of the probability space.

- Extensive experiments on 11 datasets demonstrate the effectiveness of the proposed ActiveKD and **PCoreSet**: ActiveKD consistently improves performance across all selection methods on all datasets (*e.g.*, **+29.07% on ImageNet**, averaged over methods). **PCoreSet** also consistently outperforms alternative selection strategies under the ActiveKD framework, **ranks first in 64/73** settings ($\approx$ **87.67%**) across 5 student and 3 teacher architectures.

## 2 RELATED WORKS

**Vision-Language Models (VLMs).** Recent advances in machine learning have focused on training large foundation models that either transfer pretrained knowledge to specific tasks (He et al., 2016; Dosovitskiy et al., 2020; Radford et al., 2021; Jia et al., 2021; Radford et al., 2019; Koroteev, 2021) or directly apply to target tasks (Wei et al., 2021; Liu et al., 2023; 2024b; Radford et al., 2021; Jia et al., 2021) via task instructions or zero-shot prediction using language prompts. In this paper, we utilize VLMs (Radford et al., 2021; Jia et al., 2021; Silva-Rodriguez et al., 2024) as teachers, which learn a shared embedding space for images and texts through a contrastive objective. A key advantage of such language-aligned training is the ability to perform zero-shot predictions (Radford et al., 2021; Jia et al., 2021) *without task-specific training* and few-shot predictions (Zhou et al., 2022b;a; Khattak et al., 2023a; Zhu et al., 2023; Khattak et al., 2023b; Zhao et al., 2024; Roy & Etemad, 2023; Zhang et al., 2024; Lafon et al., 2025; Gao et al., 2024; Zhang et al., 2021; Yu et al., 2023) *with few examples*, qualifying VLMs as generalist models capable of handling diverse visual recognition tasks. Despite their strong task-agnostic capabilities, effectively leveraging task-specific datasets remains crucial for developing compact models that perform well on downstream tasks, especially under labeled data scarcity—a common challenge in real-world applications.

**Active learning** (**AL**; Ren et al., 2021; Zhan et al., 2022; Li et al., 2024) is a framework specifically designed to address such labeled data scarcity issues. It assumes that acquiring domain-specific datasets through manual annotation can be prohibitively expensive in practical scenarios. Specifically, AL addresses this by strategically selecting the most informative samples from unlabeled data on the target task for annotation, maximizing model performance with minimal labeled data. Previous approaches include uncertainty-based methods (Lewis, 1995; Joshi et al., 2009; Holub et al., 2008; Houlsby et al., 2011; Gal et al., 2017; Kirsch et al., 2019; Rakesh & Jain, 2021), diversity-based techniques (Sener & Savarese, 2017; Parvaneh et al., 2022; Yehuda et al., 2022; Hacohen et al., 2022), and hybrid approaches (Ash et al., 2019; Kirsch et al., 2019; Hacohen & Weinshall, 2023; Giouroukis et al., 2025) that combine multiple criteria for sample selection. Several studies have focused on the class imbalance problem (Aggarwal et al., 2020; Bengar et al., 2022; Huang et al., 2024), where unbalanced datasets can lead selection algorithms to pick class-imbalanced samples. Recent research has integrated AL with prompt tuning (Lester et al., 2021; Jia et al., 2022; Zhou et al., 2022a) of VLMs, leveraging foundational knowledge with efficient parameter updates (Bang et al., 2024; Safaei & Patel, 2024; Kim et al., 2024). PCB (Bang et al., 2024) addressed skewed predictions of VLMs by balancing class distributions, while CB+SQ (Kim et al., 2024) enhanced this with class-guided clustering and adaptive thresholds. In contrast, we regard structured predictions as an *inductive bias*, and propose to leverage it to select samples to annotate.

**Knowledge Distillation from VLMs.** Knowledge Distillation (KD) (Hinton et al., 2015) transfers knowledge from large teachers to compact students. With the emergence of vision foundation models through self-supervised learning (Chen et al., 2020a; He et al., 2020; Grill et al., 2020; Chen & He, 2021; Caron et al., 2021; He et al., 2022) and vision-language pretraining (Radford et al., 2021; Jia et al., 2021), researchers have explored KD methods to leverage knowledge embedded within these models, moving beyond conventional KD that require training student models on identical datasets with substantial data. Early work focused on self-supervised models (Fang et al., 2021; Abbasi Koohpayegani et al., 2020; Xu et al., 2021; Wang et al., 2022a; Navaneet et al., 2022; Singh & Wang, 2024), while the advent of VLMs inspired further research, including distillation of compact VLMs from larger counterparts (Wu et al., 2023; Sun et al., 2023; Udandarao et al., 2024; Vasu et al., 2024; Yang et al., 2024), unsupervised distillation from VLM predictions (Vemulapalli et al., 2023; Wu et al., 2024; Mistretta et al., 2024), and few-shot semi-supervised distillation of VLMs (Kang et al., 2025). Some research has explored using teacher models instead of human annotation to address incorrect predictions (Baykal et al., 2022) or generate data efficiently (Wang et al., 2020; Liu et al., 2024a). Our work focuses on human-in-the-loop scenarios, aiming to identify the most informative samples for annotation within a KD framework, by leveraging the zero- or few-shot capabilities of VLMs, to maximize learning efficiency under practical constraints of AL settings.

## 3 METHOD

### 3.1 PRELIMINARIES

**Backgrounds on VLMs.** We utilize Vision-Language Models (VLMs) like CLIP (Radford et al., 2021) and ALIGN (Jia et al., 2021) as teacher models. These models jointly optimize an image

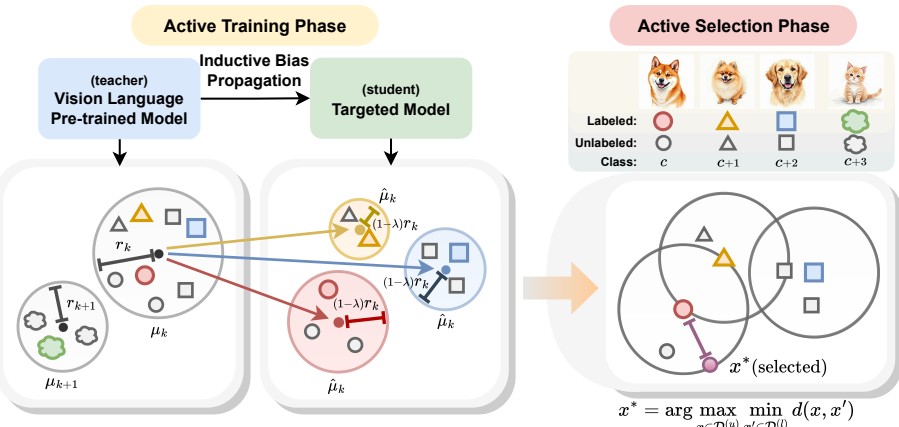

$$x^* = \arg\max_{x \in \mathcal{D}^{(u)}} \min_{x' \in \mathcal{D}^{(l)}} d(x, x')$$

Figure 3: **(Left):** Teacher model *prediction biases* $((\mu_1, r_1), \ldots, (\mu_k, r_k))$ are transferred to student models via distillation, where $\hat{\mu}_k = \mu_k + y_c$ and $\hat{r}_k = (1 - \lambda)r_k$ ($\mu$ denotes centroids, $r$ denotes radii). **(Right):** `PCoreSet` selects samples maximizing distance to labeled points in probability simplex $\Delta^{C-1}$, uncovering underrepresented regions.

encoder $f_{\mathcal{X}} : \mathcal{X} \to \mathbb{R}^d$ and a text encoder $f_{\mathcal{T}} : \mathcal{T} \to \mathbb{R}^d$ to map corresponding image-text pairs into a shared embedding space $\mathbb{R}^d$. This cross-modal alignment enables zero-shot transfer through natural language supervision. For $C$-way classification tasks, we create textual prompts (*e.g.*, "a photo of a [CLASS]") to generate class-specific text descriptions $\{t_1, t_2, \ldots, t_C\}$. The output probability vector of categorical distribution over $C$ classes is computed as the following:

$$f(x) = \sigma \left( \frac{1}{\zeta} [\texttt{CosSim}(f_{\mathcal{X}}(x), f_{\mathcal{T}}(t_1)), \ldots, \texttt{CosSim}(f_{\mathcal{X}}(x), f_{\mathcal{T}}(t_C))]^\top \right) \in \Delta^{C-1}. \quad (1)$$

Here, $\Delta^{C-1}$ is the $(C-1)$-dimensional probability simplex, $\sigma : \mathbb{R}^C \to \Delta^{C-1}$ denotes the softmax function, and $\zeta \in \mathbb{R}_{>0}$ is the temperature scaling factor (Hinton et al., 2015). Cosine similarity is defined as $\texttt{CosSim}(x, y) = \frac{x^\top y}{\|x\|_2 \|y\|_2}$. The final predicted class is given by $\arg\max_{c \in \{1, \ldots, C\}} [f(x)]_c$.

---

**Algorithm 1** ActiveKD Framework

**Require:** Initial labeled dataset $\mathcal{D}^{(l)}$, unlabeled pool $\mathcal{D}^{(u)}$, teacher VLM $f$, a selection algorithm $A$, query size $Q$, number of rounds $R$
**Ensure:** Final model $f_r$
1: **for** $r = 1$ to $R$ **do**
2:     Initialize the $r$-th round model $f_r$
3:     (Optional for few-shot teacher) adapt $f$ on $\mathcal{D}^{(l)}$
4:     Train $f_r$ with $\mathcal{L}_{\text{CE}}$ in Eq. 2 and $\mathcal{L}_{\text{KD}}$ in Eq. 3
5:     $\{x_q^*\}_{q=1}^Q \leftarrow A(\mathcal{D}^{(l)}, \mathcal{D}^{(u)}, f_r; Q)$
6:     Obtain labels $\{y_q^*\}_{q=1}^Q$ for $\{x_q^*\}_{q=1}^Q$
7:     $\mathcal{D}^{(l)} \leftarrow \mathcal{D}^{(l)} \cup \{(x_q^*, y_q^*)\}_{q=1}^Q$
8:     $\mathcal{D}^{(u)} \leftarrow \mathcal{D}^{(u)} \setminus \{x_q^*\}_{q=1}^Q$
9: **end for**
10: **return** $f_r$

**Algorithm 2** `PCoreSet` Algorithm

**Require:** Labeled dataset $\mathcal{D}^{(l)}$, unlabeled pool $\mathcal{D}^{(u)}$, model $f_r$, query size $Q$, distance $d(x, x') := \|f_r(x) - f_r(x')\|_2$
**Ensure:** Coreset $\mathcal{S} \subset \mathcal{D}^{(u)}$ with $|\mathcal{S}| = Q$
1: Initialize $\mathcal{S} \leftarrow \mathcal{D}^{(l)}$
2: Initialize $D[x] \leftarrow \min_{s \in \mathcal{S}} d(x, s)$ for all $x \in \mathcal{D}^{(u)}$
3: **while** $|\mathcal{S}| < Q$ **do**
4:     $x^* \leftarrow \arg\max_{x \in \mathcal{D}^{(u)} \setminus \mathcal{S}} D[x]$
5:     $\mathcal{S} \leftarrow \mathcal{S} \cup \{x^*\}$
6:     **for** each $x \in \mathcal{D}^{(u)} \setminus \mathcal{S}$ **do**
7:         $D[x] \leftarrow \min(D[x], d(x, x^*))$
8:     **end for**
9: **end while**
10: **return** $\mathcal{S}$

---

**Problem formulation.** In this paper, we consider a pool-based active learning (AL), a framework for building a $C$-way classifier $f_r : \mathcal{X} \mapsto \Delta^{C-1}$ for each round $r \in \{1, \ldots, R\}$ while minimizing annotation costs. We start with a labeled dataset $\mathcal{D}^{(l)} = \{(x_n^{(l)}, y_n)\}_{n=1}^N$ and an unlabeled dataset $\mathcal{D}^{(u)} = \{x_m^{(u)}\}_{m=1}^M$, where $y_n \in \{0, 1\}^C$ is the one-hot encoding label of $x_n$ and typically $N \ll M$. We first train a model $f_r$ using the current labeled dataset $\mathcal{D}^{(l)}$ for each round $r$. Then, we select subset of unlabeled data that will be requested for annotation, *i.e.*, $\{x_q^*\}_{q=1}^Q \leftarrow A(\mathcal{D}^{(l)}, \mathcal{D}^{(u)}, f_r; Q) \subset \mathcal{D}^{(u)}$, where $Q$ is the number of query datapoints and $A$ is an algorithm for selection. These selected queries are then annotated by an oracle (typically human experts) to obtain their true labels $\{y_q^*\}_{q=1}^Q$. The labeled dataset is updated as $\mathcal{D}^{(l)} \leftarrow \mathcal{D}^{(l)} \cup \{(x_q^*, y_q^*)\}_{q=1}^Q$, and the unlabeled pool is reduced by $\mathcal{D}^{(u)} \leftarrow \mathcal{D}^{(u)} \setminus \{x_q^*\}_{q=1}^Q$. See Algorithm 1 without underlines for an overview.

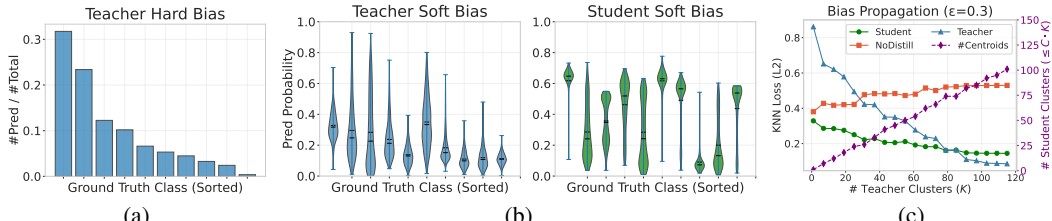

(a)                    (b)                    (c)

Figure 4: **Visualization** of prediction bias and its propagation. **(a)**: **hard prediction bias** of the teacher; **(b)**: **soft prediction bias** of the **teacher (left)** and the **student (right)** after distillation; and **(c)**: **the bias propagated from the teacher to the student**, quantified by KNN loss ($\ell_2$) across different numbers of clusters ($K$).

## 3.2 ACTIVEKD

**Active knowledge distillation (ActiveKD) framework.** In this paper, we propose the **ActiveKD** framework that leverages knowledge distillation into active learning. Our framework consists of two key components: 1) training the student model with zero-/few-shot teachers, and 2) performing sample selection using the distilled student model, potentially in collaboration with the teacher model. Specifically, we train the student model for each round using both supervised learning and knowledge distillation (Hinton et al., 2015; Kang et al., 2025):

$$\mathcal{L}_{\text{CE}} = \frac{1}{N} \sum_n \ell\left(f_r(x_n), y_n\right),\tag{2}$$

$$\mathcal{L}_{\text{KD}} = \frac{1}{N} \sum_n D_{\text{KL}}\left[f(x_n^{(l)}) \| f_r(x_n^{(l)})\right] + \frac{1}{M} \sum_m D_{\text{KL}}\left[f(x_m^{(u)}) \| f_r(x_m^{(u)})\right],\tag{3}$$

where the final loss is $\lambda\mathcal{L}_{\text{CE}} + (1-\lambda)\mathcal{L}_{\text{KD}}$, $\ell$ denotes $D_{\text{KL}}$ represent cross-entropy and Kullback-Leibler divergence, respectively. For each round $r$, we use a mini-batch version of the above objective with stochastic gradient descent to optimize the parameters of $f_r$ by leveraging the teacher prediction $f(\cdot)$. See underlines in Algorithm 1 for additional parts of ActiveKD.

**Structured prediction bias propagation.** VLMs (*e.g.*, CLIP ResNet-50; Radford et al., 2021) inherently exhibit class imbalance (Bang et al., 2024) due to their pre-training data and prompt design, as shown in Fig. 4a. Beyond imbalance, we observe that **teacher predictions form distinct clusters within constrained regions of the probability simplex**. The violin plots in Fig. 4b-**(left)** show the distributions for ground-truth (GT) classes, indicating that teacher predictions concentrate in specific regions rather than uniformly covering the space, creating *blind spots*. This **clustering behavior propagates to student models during knowledge distillation**, as students inherit similar biased patterns (Fig. 4b-**(right)**). To formalize this, we define the *structured prediction bias* as follows:

**Definition 1** (Structured prediction bias). *Teacher predictions exhibit structured prediction bias if there exist $K \in \mathbb{N}$, centroids $\{\mu_k\}_{k=1}^K \subset \Delta^{C-1}$, and radii $\{r_k\}_{k=1}^K \subset \mathbb{R}_{>0}$, such that:*

$$\forall x \in \mathcal{X}, \quad f(x) \in \bigcup_{k=1}^K \left\{p \in \Delta^{C-1} : \|p - \mu_k\|_2 \leq r_k\right\}.\tag{4}$$

In other words, all teacher predictions lie within a finite union of $\ell_2$-balls in the probability simplex. We now show that **student predictions trained via KD also exhibit similar structured bias**:

**Proposition 1** (Bias propagation through KD). *Let the teacher model $f$ exhibit structured prediction bias as defined in Definition 1, with $\{\mu_k\}_{k=1}^K$ and $\{r_k\}_{k=1}^K$. Assume the student model $f_r$ is trained via KD from $f$ using the loss $\lambda\mathcal{L}_{CE} + (1-\lambda)\mathcal{L}_{KD}$, and satisfies $\|f_r(x) - f^*(x)\|_1 \leq \epsilon$ for all $x \in \mathcal{X}$, where $f^*(x) = \lambda y + (1-\lambda)f(x)$ and $y \in \{0,1\}^C$ denotes the label of $x$. Then student predictions $f_r(x)$ also exhibit structured prediction bias defined in Definition 1. Specifically, $\forall x \in \mathcal{X}$, there exists $k \in \{1, \ldots, K\}$ such that:*

$$f_r(x) \in \left\{p \in \Delta^{C-1} : \|p - \hat{\mu}_k(x)\|_2 \leq \hat{r}_k\right\},\tag{5}$$

*where the propagated centriod is defined as $\hat{\mu}_k(x) = \lambda y + (1-\lambda)\mu_k$, and the adjusted radius is $\hat{r}_k = (1-\lambda)r_k + \epsilon$. Since $y$ is one-hot and $\mu_k$ is fixed, the set of possible centers $\{\hat{\mu}_k(x)\}$ is finite and contained in $\Delta^{C-1}$, thus satisfying the condition of Definition 1 with at most $C \cdot K$ clusters.*

We defer the proof of Proposition 1 to §A.2. Proposition 1 is illustrated in Fig. 3-(left), where a teacher cluster $(\mu_k, r_k)$ propagates to corresponding student clusters $(\hat{\mu}_k, \hat{r}_k)$ across three classes.

**Empirical validation of Proposition 1.** We compare: 1) a baseline model using $\mathcal{L}_{\text{CE}}$ on $\mathcal{D}^{(l)}$; and 2) a student model trained via KD with loss $\lambda\mathcal{L}_{\text{CE}} + (1-\lambda)\mathcal{L}_{\text{KD}}$ on $\mathcal{D}^{(l)}$ and $\mathcal{D}^{(u)}$, under the 1-shot setting. We cluster teacher predictions on $\mathcal{D}^{(u)}$ via $k$-means (Hartigan & Wong, 1979), yielding centroids $\{\mu_k\}_{k=1}^K$. For each $x \in \mathcal{D}^{(u)}$, we compute the propagated student centroid $\hat{\mu}_k = \lambda y + (1-\lambda)\mu_k$, where $y$ is the one-hot GT label, and measure the average $\ell_2$ distance between $f_r(x)$ and $\hat{\mu}_k$ (threshold $\epsilon = 0.3$). As shown in Fig. 4c, the **distilled student shows consistently lower distances** than the baseline across varying $K$, **validating Proposition 1**. The number of active propagated centroids also closely matches that of the teacher, further supporting the existence of *structured prediction bias* of student model.

### 3.3 PROBABILISTIC CORESET (PCORESET)

**Motivation.** During KD, the student model leverages both labels and teacher guidance, to generalize rather than merely mimicking teachers. As a result, the **prediction structure of the teacher acts as a prior** that shapes the hypothesis space of the student. Our key insight is that when KD is effective (with bounded error $\epsilon$), the student inherits the clustered structure of teacher (Proposition 1). Samples that deviate from this structure are particularly informative, as they fall outside regions captured by *inductive bias*. Rather than discarding this structure, we **leverage it to guide AL**: selecting samples that deviate from the established structure and expand the student's predictive capacity.

**PCoreSet selection.** To this end, we propose **P**robabilistic **C**oreset (PCoreSet), a *simple yet effective* selection strategy inspired by coreset selection (Sener & Savarese, 2017). While conventional coreset methods aim to maximize coverage in the feature space, PCoreSet instead maximizes coverage in the probability simplex $\Delta^{C-1}$ by targeting underrepresented probability regions. This enables the student to inherit the teacher's *inductive bias* more completely while actively exploring beyond it. Formally, given a labeled dataset $\mathcal{D}^{(l)}$, we greedily select $x^* \in \mathcal{D}^{(u)}$ as:

$$x^* = \arg\max_{x \in \mathcal{D}^{(u)}} \min_{x' \in \mathcal{D}^{(l)}} d(x, x'), \tag{6}$$

where $d(x, x') := \|f_r(x) - f_r(x')\|_2$ measures distance in the probability space. We then request the label $y^*$ for $x^*$ and update the labeled set: $\mathcal{D}^{(l)} \leftarrow \mathcal{D}^{(l)} \cup \{(x^*, y^*)\}$. See Algorithm 2 for details.

**Error of teacher bias.** Although PCoreSet maximally leverages the structured bias of VLM teachers, which can be **erroneous** in zero-/few-shot settings, any resulting mistakes can be corrected through the **GT labels** ($\mathcal{D}^{(l)}$), ensuring reliable learning under ActiveKD.

**The computational complexity** of PCoreSet is $\mathcal{O}(C \cdot M \cdot N)$, whereas the feature-space core-set (Sener & Savarese, 2017) has complexity $\mathcal{O}(H \cdot M \cdot N)$, where $H$ is the feature dimensionality. Thus, in most cases ($C \ll H$), PCoreSet is more efficient. While PCoreSet is less efficient than the $\mathcal{O}(C \cdot M)$ complexity of **Entropy** (Holub et al., 2008), Entropy exhibits degradation in §4.2.

## 4 EXPERIMENTS

### 4.1 EXPERIMENTAL SETUP

**Datasets.** We evaluate our approach across 11 diverse datasets: generic image recognition benchmarks (Russakovsky et al., 2015; Fei-Fei et al., 2004), and fine-grained and domain-specific benchmarks (Krause et al., 2013; Nilsback & Zisserman, 2008; Maji et al., 2013; Bossard et al., 2014; Xiao et al., 2010; Cimpoi et al., 2014; Helber et al., 2019; Soomro et al., 2012). To reduce computational cost, we subsample the unlabeled ImageNet (Russakovsky et al., 2015) pool to 100,000 images. Note that the datasets used are among the most diverse (Bang et al., 2024). See more details in §C.

**Active learning frameworks.** 1) **No Distill**: standard AL without knowledge distillation, where the model is trained only on labeled data $\mathcal{D}^{(l)}$ using cross-entropy loss $\mathcal{L}_{\text{CE}}$; 2) **ActiveKD (Zero-Shot)**: our proposed **ActiveKD** framework with a fixed zero-shot VLM teacher (Radford et al., 2021) that provides soft targets for distillation on both $\mathcal{D}^{(l)}$ and $\mathcal{D}^{(u)}$; and 3) **ActiveKD (Few-Shot)**: as in 2) **ActiveKD (Zero-Shot)** but with a few-shot VLM teacher (Silva-Rodriguez et al., 2024) fine-tuned on $\mathcal{D}^{(l)}$ using both the newly selected samples and existing labeled data.

**Active selection baselines.** We compare PCoreSet with five baselines. 1) **Random**: uniform sampling from the unlabeled pool $\mathcal{D}^{(u)}$; 2) **Entropy** (Holub et al., 2008): selects points with highest predictive entropy; 3) **CoreSet** (Sener & Savarese, 2017): maximizes diversity in feature space; 4) **BADGE** (Ash et al., 2019): combines uncertainty and diversity via gradient embeddings; and 5) **Class-Balanced** (Bang et al., 2024): promotes class diversity in the queried set.

Table 1: **Results on ImageNet and the average over 10 datasets under different AL frameworks** at the final AL round. We report the mean and 95% CI over five runs; values in parentheses denote the gain over No Distill. **ActiveKD** consistently improves over No Distill across all strategies and datasets.

| Methods | ImageNet (ResNet-50 student & ResNet-50 teacher) | | | Avg. over 10 Datasets (ResNet-18 student & ResNet-50 teacher) | | |
|---|---|---|---|---|---|---|
| | No Distill | **ActiveKD** (Zero-Shot) | **ActiveKD** (Few-Shot) | No Distill | **ActiveKD** (Zero-Shot) | **ActiveKD** (Few-Shot) |
| Random | 33.36±0.45 | 60.69±0.16 (**+27.33**) | 60.49±0.12 (**+27.13**) | 63.10±4.33 | 76.31±0.88 (**+13.21**) | 77.48±0.86 (**+14.38**) |
| Entropy | 30.76±0.24 | 60.43±0.16 (**+29.67**) | 58.87±0.14 (**+28.11**) | 64.06±3.47 | 75.58±0.86 (**+11.52**) | 77.47±0.90 (**+13.41**) |
| Coreset | 26.61±0.90 | 60.58±0.07 (**+33.97**) | 59.01±0.42 (**+32.40**) | 64.99±3.01 | 76.20±0.91 (**+11.21**) | 77.63±0.57 (**+12.64**) |
| Badge | - | - | - | 63.14±4.66 | 76.18±0.74 (**+13.04**) | 77.50±0.67 (**+14.36**) |
| ClassBalanced | 34.44±0.60 | 61.07±0.11 (**+26.63**) | 61.13±0.07 (**+26.69**) | 63.57±4.33 | 76.28±0.87 (**+12.71**) | 77.56±0.66 (**+13.99**) |
| **PCoreSet** (ours) | 33.41±0.41 | 61.16±0.14 (**+27.75**) | 61.57±0.24 (**+28.16**) | 64.94±3.25 | 77.44±0.84 (**+12.50**) | 78.81±0.42 (**+13.87**) |
| *Avg. over Methods* | 31.72 | 60.79 (**+29.07**) | 60.21 (**+28.50**) | 63.97 | 76.33 (**+12.37**) | 77.74 (**+13.78**) |

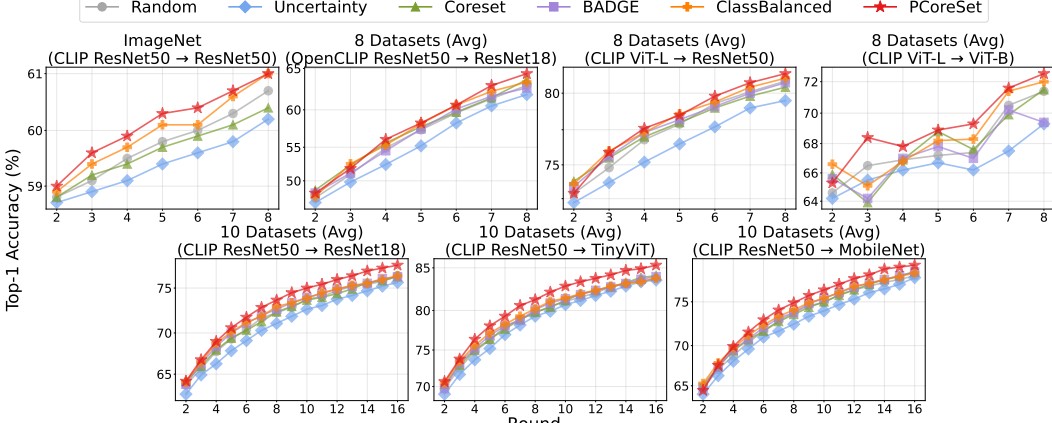

Figure 5: **Results on ImageNet and the average over 8 or 10 datasets across 5 student and 3 teacher architectures** under ActiveKD (Zero-Shot) with either 8 or 16 rounds. We report the mean and 95% CI over five runs. **PCoreSet** achieves **the best performance in 64/73** settings (≈**87.7%**)
.

**Implementation details.** All AL methods share the **same experimental setup** when evaluated under the same AL framework: We use a query size of $Q = C$ per round (1-shot), with $R = 16$ rounds for all datasets except ImageNet, which uses $R = 8$. We adopt DHO (Kang et al., 2025) as the KD method in all experiments to leverage the unlabeled pool. For ImageNet, we use a self-supervised ResNet-50 (Caron et al., 2021) as the student model. For the other 10 datasets, we use ResNet-18 (He et al., 2016), MobileNetV2 (Sandler et al., 2018), TinyViT (Wu et al., 2022), and ViT-B/16 (Dosovitskiy et al., 2020), all pretrained on ImageNet. We use CLIP ResNet-50 (Radford et al., 2021), OpenCLIP (Cherti et al., 2023), and ViT-L/14 as the zero-shot teacher and CLAP (Silva-Rodriguez et al., 2024) as the few-shot teacher. We exclude BADGE (Ash et al., 2019) on ImageNet due to memory limitations when computing gradients over its 1,000 classes. We report the mean performance with 95% confidence intervals across 5 random seeds. We defer further details to §B.

## 4.2 MAIN RESULTS

**Effectiveness of ActiveKD (Zero-Shot).** We first evaluate **ActiveKD** with ResNet-18/ResNet-50 students and a ResNet-50 zero-shot teacher. Tab. 1 shows that **ActiveKD (Zero-Shot)** **consistently improves No Distill across all datasets and selection strategies**: *e.g.*, it improves accuracy by +27.33% on ImageNet with *Random* selection (33.36% → 60.69%) and by +29.07% on average across strategies. This improvement extends across 10 datasets with an average gain of +13.21% (63.10% → 76.31%) for *Random* selection, and +12.37% on average across all strategies. These substantial gains demonstrates the effectiveness of our proposed ActiveKD under limited supervision.

**Effectiveness of ActiveKD (Few-Shot).** **ActiveKD (Few-Shot)** also consistently improves over No Distill across all datasets and strategies; relative to **ActiveKD (Zero-Shot)** it yields +1.17% for *Random* and +1.41% on average across the 10 datasets, demonstrating **the benefit of strong few-shot teachers**. However, the improvements of **ActiveKD (Few-Shot)** are not consistent on ImageNet; *e.g.*, the gains are only +0.06% with *ClassBalanced* and **+0.41% with PCoreSet**. We attribute this to the large number of classes in ImageNet ($C$=1000), which increases labeled samples per round, reducing the added value of teacher signals compared with other 10 datasets having $C \approx 100$. For the same reason, *Random* often outperforms *Entropy*/*Coreset* on ImageNet—also noted by Emam et al. (2021); Bang et al. (2024)—making *Random* a strong baseline.

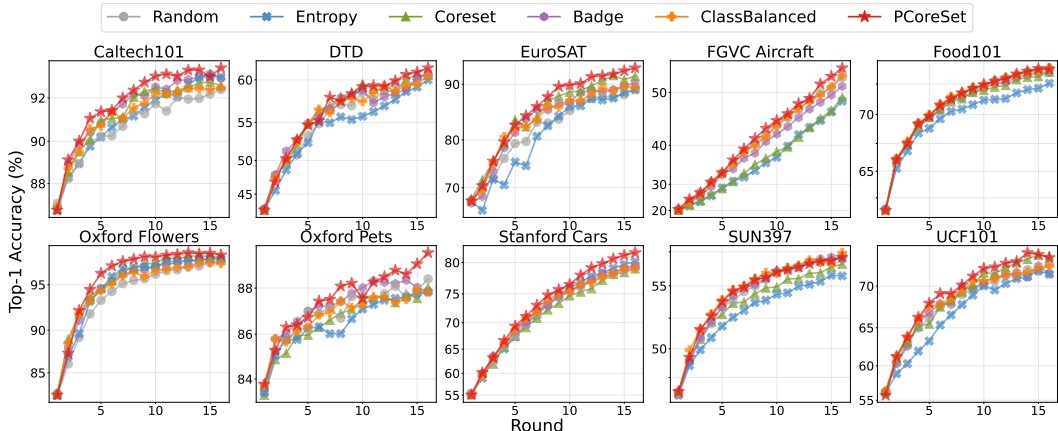

Figure 6: Results on 10 datasets using **ResNet-18** with ResNet-50 teacher under ActiveKD (Zero-Shot).

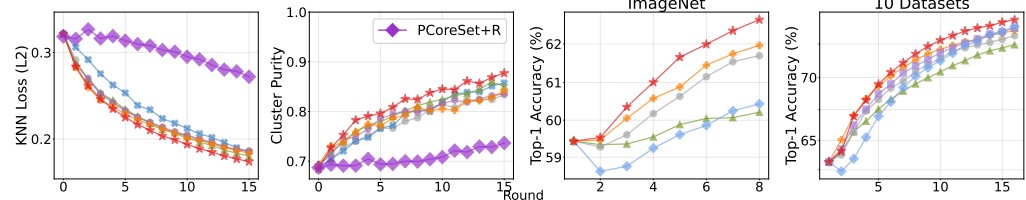

Figure 7: **Left two panels:** KNN loss (L2) and cluster purity of **PCoreSet** and other selection strategies across 10 datasets; **right two panels:** performance of few-shot teachers on 11 datasets across rounds.

**Effectiveness of PCoreSet under ActiveKD (Zero-Shot).** We now compare **PCoreSet** with selection baselines under **the same ActiveKD (Zero-Shot)**. Fig. 5 reports results on ImageNet and averages over 10 or 8 datasets across 8 or 16 AL rounds. In total, we consider **73 settings**, computed as $(8-1) \times 4 + (16-1) \times 3$, excluding the first round where all methods share the same $\mathcal{D}^{(l)}$. The factors 4 and 3 denote the number of configurations for the 8-round and 16-round setups, including ImageNet and cross-dataset averages. We observe slight degradation of **PCoreSet** in the first two rounds: **PCoreSet** relies on structured teacher bias, but at this stage there may be **insufficient GT labels to reliably correct errors** introduced by imperfect VLM teachers. Beyond the first two rounds, however, **PCoreSet ranks first in 64 of 73 cases (87.7%)**, consistently achieving the best performance and becoming increasingly effective as active learning progresses. These results are not marginal but **significant**, given that all methods fairly use the same setup except for **data selection**.

**In-depth comparison.** Fig. 6 presents an in-depth evaluation of selection strategies under ActiveKD (Zero-Shot) across 10 datasets, using a ResNet-18 student and a ResNet-50 teacher. We observe that **PCoreSet** achieves the best performance or is at least on par with the best. Similar trends appear in other in-depth evaluations across different settings; see §D for details.

### 4.3 ANALYSIS

**The acceleration of structured bias propagation.** To assess whether **PCoreSet** promotes *structured bias propagation*, we track the same metric, *i.e.*, KNN loss ($\ell_2$) and clster purity, used in Fig. 4c across AL rounds. We also include a **counterfactual baseline**, **PCoreSet**+R (Reverse), which suppresses probability-space coverage by replacing $\arg\max$ with $\arg\min$ in line 4 of Algorithm 2. As shown in the **two leftmost panels** of Fig. 7, **PCoreSet** achieves consistently **the lowest KNN loss and the highest cluster purity**, confirming that it promotes *structured bias propagation* as intended. In contrast, **PCoreSet**+R shows the highest KLL loss and the lowest cluster purity, which corroborates that maximizing sample diversity in the probability space encourages bias propagation.

**A virtuous cycle with few-shot teachers.** Beyond accelerating structured bias propagation, we evaluate the few-shot teacher under ActiveKD. Because acquisition is driven by the student predictions, we can test whether the selected samples by each selection method also benefit the teacher. As shown in the **two rightmost** panels of Fig. 7, **PCoreSet** consistently outperforms competing selection strategies for the teacher. This supports that the *structured prediction bias* **of teacher models propagates to the student**: by targeting regions underrepresented in the probability space of student models, **PCoreSet** also covers underrepresented teacher modes, **improving both models and inducing a virtuous cycle of better acquisition and stronger distillation**.

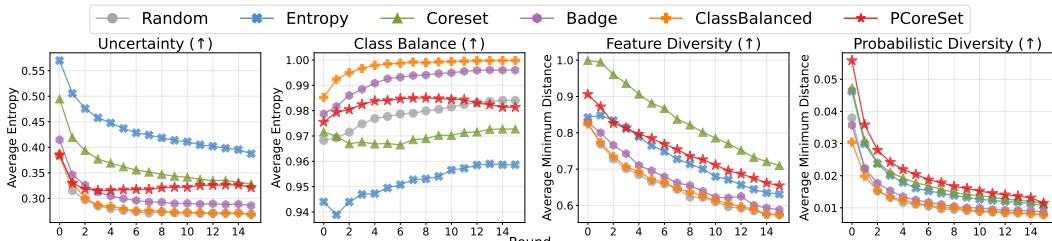

Figure 8: Comparison of four selection criteria: 1) Uncertainty, 2) Class balance, 3) Feature space diversity, and 4) Probability space diversity. The results are averaged over the 10 datasets excluding ImageNet.

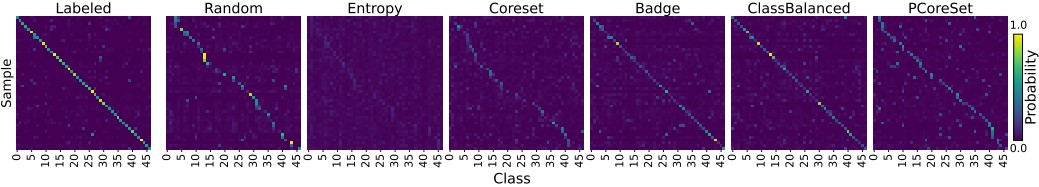

Figure 9: The heatmap of output probability vectors from different selection strategies in the first active learning round using the DTD dataset. See §E.1 for results on other datasets.

**Active selection criteria.** We analyze the progression of four strategies designed to be maximized throughout AL rounds: 1) uncertainty, 2) class balance, 3) feature space diversity, and 4) probability space diversity (**PCoreSet**). Criteria 1) and 2) are measured using normalized Shannon entropy, while 3) and 4) are evaluated using normalized average minimum distances. We report the average of each metric across 10 datasets, excluding ImageNet. As shown in Fig. 8, each method performs best on its respective objective: *Entropy* maximizes uncertainty, *ClassBalanced* optimizes the entropy of the class distribution, and *Coreset* maximizes feature space diversity. As expected, our **PCoreSet** method **achieves the highest diversity in probability space**, while also ranking third in uncertainty and class balance, and second in feature diversity. Although *Coreset* is effective in covering the feature space and offers moderate probabilistic diversity, it performs poorly in class balance, which may be the reason that it underperforms than *ClassBalanced* baseline.

**Sample diversity of PCoreSet.** To visualize the sample diversity of **PCoreSet**, we visualize probability heatmaps for the initial labeled set and for the samples selected after round 1 by each active-learning strategy (Fig. 9). The heatmaps show that **PCoreSet** selects samples with highly diverse probability profiles, maximizing distributional coverage relative to the initial set. We also visualize selected images from Caltech101 in Fig. 10, where **PCoreSet** chooses visually distinct examples spanning many classes (see §E.2 for more cases). We believe this probabilistic and visual diversity helps us **select samples underrepresented in the probability space**, thereby more efficiently propagating the teacher's *structured prediction bias* during distillation.

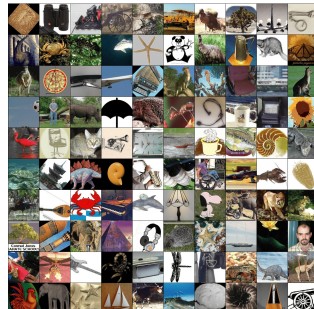

Figure 10: Samples selected by **PCoreSet** on Caltech101.

## 5 CONCLUSION, LIMITATION, AND FUTURE WORK

We introduced ActiveKD, integrating AL with KD by leveraging zero-/few-shot capabilities of VLMs. We discovered VLMs exhibit *structured prediction bias* that benefits student learning, and proposed **PCoreSet** to maximize diversity in probability space when selecting samples. Extensive experiments across 11 datasets show the effectiveness of ActiveKD and **PCoreSet**.

**Limitations and future work.** While our experiments focused on VLMs, we believe AL has even greater potential when leveraging the growing capabilities of generalist foundation models (Wei et al., 2021; Liu et al., 2023; Bai et al., 2023; Achiam et al., 2023; Team et al., 2023). Many task-specific applications face challenges due to limited labels, and using generalist models could provide valuable opportunities for future exploration. Our implementation is limited to visual recognition tasks, as they are representative and provide an ideal testbed. In addition, the zero-/few shot capabilities of VLMs in visual recognition make them natural candidates for KD. However, extending our approach to object detection and segmentation, is a promising direction that may require architectural adaptations. Furthermore, while various KD and semi-supervised learning methods (*e.g.*, Wang et al., 2022b) are orthogonal to ActiveKD, assessing their effectiveness within this framework remains future work.

## Reproducibility Statement

We are committed to ensuring the reproducibility of our research. To facilitate replication and extension of our work, we provide comprehensive implementation details throughout the paper and supplementary materials. Specifically, we include: 1) detailed experimental setup in §4.1 and §B, covering all hyperparameters, optimization settings, and training procedures; 2) complete algorithmic descriptions of our DHO training framework (Algorithm 3) and inference procedure (Algorithm 4); 3) specifications of all model architectures tested, including ResNet18, ResNet50, ViT-T/16, ViT-S/16, and MobileNetV2, along with teacher models CLIP ViT-B/32, CLIP ViT-B/16, and ALIGN; 4) comprehensive dataset information for all 11 benchmarks used in our evaluation (§C); and 5) statistical rigor through reporting mean performance with 95% confidence intervals across 5 random seeds for all experiments. Our implementation leverages publicly available pretrained models and standard datasets. We will release our code and experimental configurations upon publication to enable full reproducibility of our results.

## Ethics Statement

Our work presents no new ethical concerns as **ActiveKD** is a purely technical contribution for active learning with knowledge distillation using existing publicly available datasets (ImageNet, Caltech101, StanfordCars, Flowers102, FGVCAircraft, Food101, SUN397, DTD, EuroSAT, UCF101, and Ox-fordPets) that contain no personally identifiable information. No additional data collection, human subjects research, or sensitive information processing is involved in this work. We acknowledge that vision-language models (CLIP and ALIGN) may contain biases from their pre-training data, which our distillation framework preserves without amplification. The computational requirements are modest (single GPU for training student models across all benchmarks), significantly lower than training vision-language models from scratch, promoting research accessibility while minimizing environmental impact.

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

APPENDIX OVERVIEW

This appendix contains supplementary information organized as follows:

- **Theoretical Analysis** (§A): provides formal analysis of bias propagation in knowledge distillation, including optimal student prediction analysis (§A.1), formal characterization of teacher prediction bias (§A.2), and mathematical proof of how biases transfer from teacher to student models (Proposition 2).
- **Implementation** (§B): detailed experimental setup including model configurations, training parameters.
- **Datasets** (§C): comprehensive description of the datasets used in our experiments, including ImageNet and other benchmarks used for evaluating bias propagation, as well as detailed analysis of class distributions across all datasets.
- **Additional Experiments** (§D): supplementary experimental results including evaluations on alternative architectures (e.g., MobileNetV2 in §D.1), active learning without knowledge distillation (§D.2), few-shot teacher distillation scenarios (§D.3), and logit adjustment for calibrated teacher prediction (§D.4).
- **Additional Qualitative Results** (§E): additional results of the proposed method, including heatmap visualizations of selected samples across datasets (§E.1) and qualitative analysis of samples selected by our **PCoreSet** method (§E.2).

# A  THEORETICAL ANALYSIS

This section presents a formal analysis of bias propagation in knowledge distillation, examining how teacher model biases influence student model predictions. We first analyze the optimal student prediction after distillation from a teacher model in Section A.1, building upon the analytical framework established in DHO (Kang et al., 2025). Subsequently, in Section A.2, we formally characterize the mechanisms through which teacher prediction biases propagate to and constrain the distilled student model, providing the theoretical foundation for our debiasing strategies.

## A.1  OPTIMAL STUDENT PREDICTION AFTER DISTILLATION

Let us first establish our notation. We consider a student model with a feature extractor $h : \mathcal{X} \to \mathbb{R}^d$ that maps inputs to feature representations, and a classification head $g : \mathbb{R}^d \to \mathbb{R}^C$ that maps features to logits, with the final prediction obtained by applying the softmax function $\sigma : \mathbb{R}^C \to \Delta^{C-1}$ to these logits, resulting in $f_r(x) = \sigma(g(h(x)))$, where $r$ denotes the active learning round. Following DHO (Kang et al., 2025), we consider two target probability distributions: the ground truth label distribution $y$, typically one-hot encoded vectors where $y_c = 1$ for the true class $c$ and 0 elsewhere, and the teacher's softened distribution $f(x)$ for input $x \in \mathcal{X}$.

**Theorem 1** (Optimal Distribution for Knowledge Distillation). *The distribution $f^*(x)$ that minimizes the weighted combination of cross-entropy loss with respect to $y$ and Kullback-Leibler divergence with respect to $f(x)$:*

$$\mathcal{L}(f^*(x)) = \lambda \ell(f^*(x), y) + (1 - \lambda) D_{\mathrm{KL}}(f(x) \| f^*(x)) \tag{7}$$

*is the weighted arithmetic mean:*

$$f^*(x) = \lambda y + (1 - \lambda) f(x) \tag{8}$$

*where $\lambda \in [0, 1]$ is the weighting hyperparameter.*

This optimal prediction $f^*(x)$ represents the theoretical target that a single-head distillation approach should converge to.

**Assumption 1** ($\varepsilon$-Convergence). *We assume that after sufficient training, a student model has converged to the optimal prediction with bounded error:*

$$\sup_x \| f_r(x) - f^*(x) \|_1 \leq \varepsilon \tag{9}$$

*where $f_r(x) = \sigma(g(h(x)))$ is the student model's prediction, $f^*(x) = \lambda y + (1-\lambda)f(x)$ is the optimal prediction, $\|\cdot\|_1$ denotes the L1 norm, and $\varepsilon > 0$ is a small constant representing generalization error.*

This assumption is reasonable for both single-head and dual-head approaches:

For the **Single-Head Approach**, a single output head $g(h(x))$ is trained directly on the combined loss $\mathcal{L}(f^*(x))$, and with sufficient capacity and training, it approximates $f^*(x)$ with error bounded by $\varepsilon$.

For the **Dual-Head Approach**, in the Dual-Head Optimization (DHO) framework (Kang et al., 2025), shared features $h(x)$ are extracted from input $x$ and two specialized classification heads are employed: $g_{\text{CE}}(h(x))$, optimized for ground truth labels using cross-entropy loss, and $g_{\text{KD}}(h(x))$, optimized for teacher predictions using KL divergence. This approach addresses the constrained optimization problem:

$$\min_{h, g_{\text{CE}}, g_{\text{KD}}} \lambda \mathbb{E}_{x,y}[\ell(\sigma(g_{\text{CE}}(h(x))), y)] + (1-\lambda)\mathbb{E}_x[D_{\text{KL}}(f(x)\|\sigma(g_{\text{KD}}(h(x))))] \tag{10}$$

At inference time, their outputs are combined to form the final prediction:

$$f_r(x)_{\text{DHO}} = \alpha \cdot \sigma(g_{\text{CE}}(h(x))) + (1-\alpha) \cdot \sigma(g_{\text{KD}}(h(x))/\beta) \tag{11}$$

The DHO framework (Kang et al., 2025) provides a formal proof that under appropriate optimization conditions, the dual-head inference is equivalent to single-head inference when setting $\alpha = \lambda$ and $\beta = \tau$, meaning that for any choice of distillation weight $\lambda$ and temperature $\tau$, the two approaches converge to the same prediction.

**Theorem 2** (Optimal Prediction as Linear Combination). *Under Assumption 1, the optimal prediction $f^*(x) = \lambda y + (1-\lambda)f(x)$ is a linear combination of ground truth and teacher prediction, with bounded error $\varepsilon$, regardless of whether a single-head or dual-head approach is employed. This formulation provides the basis for analyzing bias propagation in knowledge distillation.*

This theoretical foundation establishes that regardless of the architectural choice (single-head or dual-head), the optimal student prediction converges to a linear combination of the ground truth and teacher prediction, providing a principled basis for analyzing bias propagation in knowledge distillation.

## A.2 Formal Definition of Teacher Prediction Bias

**Definition 2** (Structured prediction bias). *Teacher predictions exhibit structured prediction bias if there exist $K \in \mathbb{N}$, centroids $\{\mu_k\}_{k=1}^K \subset \Delta^{C-1}$, and radii $\{r_k\}_{k=1}^K \subset \mathbb{R}_{>0}$, such that:*

$$\forall x \in \mathcal{X}, \quad f(x) \in \bigcup_{k=1}^K \left\{ p \in \Delta^{C-1} : \|p - \mu_k\|_2 \leq r_k \right\}. \tag{12}$$

This formulation captures the empirical phenomenon that foundation teacher models tend to produce predictions clustered within a limited number of regions in the probability simplex $\Delta^{C-1}$ (where $C$ is the number of classes), rather than utilizing the entire space of possible probability distributions. The number of biased regions $K$ is typically much smaller than the model's theoretical expressivity would allow, reflecting inherent biases in the teacher's pretraining data and architecture.

**Proposition 2** (Bias propagation through KD). *Let the teacher model $f$ exhibit structured prediction bias as defined in Definition 2, with $\{\mu_k\}_{k=1}^K$ and $\{r_k\}_{k=1}^K$. Assume the student model $f_r$ is trained via KD from $f$ using the loss $\lambda\mathcal{L}_{CE} + (1-\lambda)\mathcal{L}_{KD}$, and satisfies $\sup_x \|f_r(x) - f^*(x)\|_1 \leq \varepsilon$ for all $x \in \mathcal{X}$, where $f^*(x) = \lambda y + (1-\lambda)f(x)$ and $y \in \{0,1\}^C$ denotes the one-hot label of $x$. Then student predictions $f_r(x)$ also exhibit structured prediction bias as in Definition 2. Specifically, for every $x \in \mathcal{X}$, there exists $k \in [K]$ such that:*

$$f_r(x) \in \left\{ p \in \Delta^{C-1} : \|p - \hat{\mu}_k(x)\|_2 \leq \hat{r}_k \right\}, \tag{13}$$

*where the propagated centroid is defined as $\hat{\mu}_k(x) = \lambda y + (1-\lambda)\mu_k$, and the adjusted radius is $\hat{r}_k = (1-\lambda)r_k + \varepsilon$. Since $y$ is one-hot and $\mu_k$ is fixed, the set of possible centers $\{\hat{\mu}_k(x)\}$ is finite and contained in $\Delta^{C-1}$, thus satisfying the condition of Definition 2 with at most $C \cdot K$ clusters.*

*Proof.* From Assumption 1, we have:

$$\|f_r(x) - f^*(x)\|_1 \leq \varepsilon \tag{14}$$

For any L1 norm bounded by $\varepsilon$, the maximum deviation in any single component is also bounded by $\varepsilon$, yielding:

$$f^*(x) - \varepsilon\mathbf{1} \leq f_r(x) \leq f^*(x) + \varepsilon\mathbf{1} \tag{15}$$

From Definition 2, for input $x$, the teacher's prediction $f(x)$ satisfies:

$$\mu_i - r_i\mathbf{1} \leq f(x) \leq \mu_i + r_i\mathbf{1} \tag{16}$$

for some bias region center $\mu_i$. Substituting this into the formula for $f^*(x) = \lambda y + (1 - \lambda)f(x)$:

$$\lambda y + (1 - \lambda)(\mu_i - r_i\mathbf{1}) \leq f^*(x) \leq \lambda y + (1 - \lambda)(\mu_i + r_i\mathbf{1}) \tag{17}$$

Combining these inequalities:

$$\lambda y + (1 - \lambda)(\mu_i - r_i\mathbf{1}) - \varepsilon\mathbf{1} \leq f_r(x) \leq \lambda y + (1 - \lambda)(\mu_i + r_i\mathbf{1}) + \varepsilon\mathbf{1} \tag{18}$$

To show that student predictions remain constrained, we note that for each teacher bias region centered at $\mu_k$, the student's predictions are bounded by:

$$\hat{\mu}_k(x) - (1 - \lambda)r_k\mathbf{1} - \varepsilon\mathbf{1} \leq f_r(x) \leq \hat{\mu}_k(x) + (1 - \lambda)r_k\mathbf{1} + \varepsilon\mathbf{1} \tag{19}$$

where $\hat{\mu}_k(x) = \lambda y + (1 - \lambda)\mu_k$. This implies that student predictions are constrained within regions centered at $\hat{\mu}_k(x)$ with radius $\hat{r}_k = (1 - \lambda)r_k + \varepsilon$.

While the teacher's predictions are constrained to $K$ distinct regions, the student's predictions may not maintain exactly $K$ distinct regions due to potential merging or splitting effects when combining with ground truth labels. However, the student's predictions remain bounded and cannot freely explore the entire probability simplex.

Setting $\hat{r}_k = (1 - \lambda)r_k + \varepsilon$, we can express the student's prediction constraint as:

$$f_r(x) \in \bigcup_{k'=1}^{K'} \{p \in \Delta^{C-1} : \|p - \hat{\mu}_k(x)\|_2 \leq \hat{r}_k\} \tag{20}$$

$\square$

This demonstrates that regardless of the exact number of distinct regions, the student model inherits constrained prediction patterns from the teacher model, with centers shifted by the ground truth component and radii scaled by $(1 - \lambda)$ plus the convergence error $\varepsilon$.

**Corollary 1** (Student Bias Inheritance). *Let $K'$ denote the number of distinct prediction regions in the student model after distillation. The student model's predictions are constrained to these $K'$ distinct regions in the probability simplex, where:*

$$K' \leq K \cdot C \tag{21}$$

*where $C$ is the number of classes.*

*Proof.* Consider the student's optimal prediction $f^*(x) = \lambda y + (1 - \lambda)f(x)$ for any input $x$. From Definition 2, the teacher's prediction $f(x)$ belongs to one of $K$ distinct regions in the probability simplex, each centered at some $\mu_k$ with radius $r_k$.

For each teacher bias region centered at $\mu_k$, the student's prediction center becomes $\hat{\mu}_{k,c} = \lambda e_c + (1 - \lambda)\mu_k$, where $e_c$ is the one-hot vector for class $c$. This is a function of both the teacher's bias center $\mu_k$ and the ground truth label.

Since there are $K$ distinct teacher bias regions and $C$ distinct ground truth label distributions (one per class), there can be at most $K \cdot C$ distinct combinations of $(\mu_k, e_c)$. Each such combination produces a potential distinct student bias region centered at $\hat{\mu}_{k,c} = \lambda e_c + (1 - \lambda)\mu_k$.

Therefore, the maximum number of distinct student prediction regions is bounded by:

$$K' \leq K \cdot C \tag{22}$$

These student regions are convex combinations of ground truth labels and the teacher's biased regions, with an additional error margin of $\varepsilon$. Specifically, for each teacher bias region with center $\mu_k$ and radius $r_k$, and each class $c$, there corresponds a student bias region with center $\hat{\mu}_{k,c} = \lambda e_c + (1-\lambda)\mu_k$ and radius $\hat{r}_k = (1-\lambda)r_k + \varepsilon$. $\qquad\square$

This establishes a comprehensive mathematical relationship between teacher bias and its propagation to the student model during knowledge distillation. The student model inherits a structured and constrained prediction space from the teacher, which constitutes a direct and quantifiable form of bias propagation through the distillation process.

## B IMPLEMENTATION DETAILS

---

**Algorithm 3** DHO Training with zero-shot CLIP (Radford et al., 2021) teacher

---

1: **Input:** labeled set $\mathcal{D}^{(l)} = \{(x_i^{(l)}, y_i)\}_{i=1}^N$, unlabeled set $\mathcal{D}^{(u)} = \{x_j^{(u)}\}_{j=1}^M$,
2: $\quad$ student feature extractor $g$, prediction heads $h_{\text{CE}}, h_{\text{KD}}$, teacher encoders $f_{\mathcal{X}}, f_{\mathcal{T}}$,
3: $\quad$ prompt template "A photo of [CLASS]", temperature scaling factors $\zeta, \eta$,
4: $\quad$ balancing hyperparameter $\lambda$,
5: $\quad$ supervised mini-batch size $B$, and unsupervised mini-batch size $B'$.
6: **while** not converged **do**
7: $\quad$ Sample mini-batch $\mathcal{B}^{(l)} = \{(x_b^{(l)}, y_b)\}_{b=1}^B$ from $\mathcal{D}^{(l)}$, $\mathcal{B}^{(u)} = \{x_{b'}^{(u)}\}_{b'=1}^{B'}$ from $\mathcal{D}^{(l)} \cup \mathcal{D}^{(u)}$.
8: $\quad$ // Process labeled data
9: $\quad$ **for** each $(x_b^{(l)}, y_b) \in \mathcal{B}^{(l)}$ **do**
10: $\quad\quad$ $z_b^{(l)} \leftarrow g(x_b^{(l)})$
11: $\quad\quad$ $\hat{p}_{\text{CE},b}^{(l)} \leftarrow \sigma(h_{\text{CE}}(z_b^{(l)}))$
12: $\quad\quad$ $\hat{p}_{\text{KD},b}^{(l)} \leftarrow \sigma(\frac{1}{\eta} h_{\text{KD}}(z_b^{(l)}))$
13: $\quad\quad$ $p_b^{(l)} \leftarrow \sigma\left(\frac{1}{\zeta\cdot\eta}[\texttt{CosSim}(f_{\mathcal{X}}(x_b^{(l)}), f_{\mathcal{T}}(t_1)), \ldots, \texttt{CosSim}(f_{\mathcal{X}}(x_b^{(l)}), f_{\mathcal{T}}(t_C))]^\top\right)$
14: $\quad$ **end for**
15: $\quad$ // Process unlabeled data
16: $\quad$ **for** each $x_{b'}^{(u)} \in \mathcal{B}^{(u)}$ **do**
17: $\quad\quad$ $z_{b'}^{(u)} \leftarrow g(x_{b'}^{(u)})$
18: $\quad\quad$ $\hat{p}_{\text{KD},b'}^{(u)} \leftarrow \sigma(\frac{1}{\eta} h_{\text{KD}}(z_{b'}^{(u)}))$
19: $\quad\quad$ $p_{b'}^{(u)} \leftarrow \sigma\left(\frac{1}{\zeta\cdot\eta}[\texttt{CosSim}(f_{\mathcal{X}}(x_{b'}^{(u)}), f_{\mathcal{T}}(t_1)), \ldots, \texttt{CosSim}(f_{\mathcal{X}}(x_{b'}^{(u)}), f_{\mathcal{T}}(t_C))]^\top\right)$
20: $\quad$ **end for**
21: $\quad$ // Compute losses and update
22: $\quad$ $\mathcal{L}_{\text{CE}} \leftarrow \frac{1}{B}\sum_{b=1}^B \ell(\hat{p}_{\text{CE},b}^{(l)}, y_b)$
23: $\quad$ $\mathcal{L}_{\text{KD}} \leftarrow \frac{1}{B}\sum_{b=1}^B D_{\text{KL}}(\hat{p}_{\text{KD},b}^{(l)} || p_b^{(l)}) + \frac{1}{B'}\sum_{b'=1}^{B'} D_{\text{KL}}(\hat{p}_{\text{KD},b'}^{(u)} || p_{b'}^{(u)})$
24: $\quad$ $\mathcal{L} \leftarrow \lambda\mathcal{L}_{\text{CE}} + (1-\lambda)\mathcal{L}_{\text{KD}}$
25: $\quad$ Update parameters of $g, h_{\text{CE}}, h_{\text{KD}}$ using $\nabla\mathcal{L}$
26: **end while**

---

---

**Algorithm 4** DHO Inference

1: **Input:** an image $x$, feature extractor $g$, prediction heads $h_{CE}, h_{KD}$, linear coefficient $\alpha$, temperature scaling $\beta$
2: $z \leftarrow g(x)$
3: $\hat{p}_{CE} \leftarrow \sigma(h_{CE}(z))$
4: $\hat{p}_{KD} \leftarrow \sigma(h_{KD}(z)/\beta)$
5: $\hat{p} \leftarrow \alpha \cdot \hat{p}_{CE} + (1 - \alpha) \cdot \hat{p}_{KD}$
6: $\hat{y} \leftarrow \arg\max_c(\hat{p}_c)$
7: **Return:** $\hat{y}$

---

We employ the DHO (Kang et al., 2025) framework for knowledge distillation in our approach, as it provides a principled and effective mechanism for transferring knowledge from foundation teacher models to student models. DHO's key advantage lies in its dual-head architecture that maintains separated optimization paths for labeled data and teacher knowledge, thereby preserving the distinct learning signals throughout training. Algorithm 3 details the DHO training procedure with our zero-shot CLIP teacher, while Algorithm 4 outlines the inference process that effectively combines both optimization pathways.

Table 2 presents the comprehensive implementation details for our experiments across three distinct settings. For ImageNet experiments, we employed a DINO self-supervised ResNet-50 student model with CLIP ResNet-50 as the teacher, conducting 8 active learning rounds starting from a 1-shot setting with 1,000 queries per round, using AdamW optimization with carefully tuned hyperparameters. Our experiments on 10 additional datasets utilized various student architectures (ResNet-18, MobileNetV2, and ViT-Tiny) with CLIP ResNet-50 as the teacher, extending to 16 active learning rounds with class-count-based query sizes and longer training epochs. For the Few-Shot Teacher Active Distillation setting, we implemented CLAP on CLIP ResNet-50 following the original paper's methodology with a reduced learning rate (0.01) to ensure better convergence in our specific experimental context. All experiments were conducted across 5 different random seeds without utilizing validation splits to simulate realistic low-data scenarios.

**Few-Shot Teacher Active Distillation.**   Recognizing the potential for improving teacher network performance, particularly in low-data regimes, we explored few-shot learning techniques to enhance the distillation process. After considering various approaches (Jia et al., 2022; Zhou et al., 2022b;a; Khattak et al., 2023a; Zhu et al., 2023; Khattak et al., 2023b; Menghini et al., 2023; Zhao et al., 2024; Roy & Etemad, 2023; Zhang et al., 2024; Lafon et al., 2024), we incorporated CLAP (Silva-Rodriguez et al., 2024), a method that notably does not require a validation set (Silva-Rodriguez et al., 2024; Murugesan et al., 2024; Morales-Álvarez et al., 2024), as our teacher model. For training the teacher, we followed the setting from the original CLAP paper (Silva-Rodriguez et al., 2024), with only one modification: reducing the learning rate from 0.1 to 0.01 for better convergence in our setting. This configuration represents a more realistic scenario wherein both teacher and student networks evolve simultaneously within the active distillation framework, better reflecting practical applications where pre-trained teacher models may not be available.

**Implementation of BADGE under DHO.**   BADGE (Ash et al., 2019) adopts a hybrid approach to maximize uncertainty and diversity simultaneously, utilizing the gradient of the final linear classifier layer which has $H \times C$ elements, where $H$ is the hidden dimension of the backbone feature and $C$ is the number of classes in the teacher model. However, DHO (Kang et al., 2025) extends beyond conventional single classifier architectures to dual classifiers that follow different optimization paths for labeled ground truth and teacher distillation signals. The dual classifier approach provides complementary information from both learning objectives, enhancing model performance through this specialized learning framework. In order to adopt DHO into the BADGE selection method, we manually calculate the gradients of both classifiers and concatenate them to form a gradient representation with $2 \times H \times C$ elements. This concatenated gradient naturally extends the original BADGE selection algorithm while preserving its core k-means++ clustering mechanism.

Table 2: Implementation details for our experiments across different settings.

| | |
|---|---|
| *Active Learning on ImageNet* | |
| **Model Configuration** | **Training Details** |
| • **Student:** DINO self-supervised ResNet-50 (Caron et al., 2021)
• **Teacher:** CLIP ResNet-50 (Radford et al., 2021)
• **Active learning rounds:** 8
• **Initial setting:** 1-shot (single image per class)
• **Query size:** 1,000
• **Unlabeled pool:** 100,000 samples randomly selected with seed
• **Random seeds:** 5 different seeds
• **KD parameters:** $\zeta = 0.01$, $\eta = 2$, $\lambda = 0.5$
• **DHO parameters:** $\alpha = 0.4$, $\beta = 0.5$
• **Validation:** No validation split used | • **Epochs:** 20 for first 7 rounds, 50 for final round
• **Optimizer:** AdamW ($\beta_1 = 0.9$, $\beta_2 = 0.999$)
• **Learning rate:** $1 \times 10^{-3}$
• **Weight decay:** $1 \times 10^{-2}$
• **Batch size:** 512 (labeled: 256, unlabeled: 256) |
| *Active Learning on 10 Additional Datasets* | |
| **Model Configuration** | **Training Details** |
| • **Student:** ImageNet pre-trained ResNet-18 (He et al., 2016), MobileNetV2 (Sandler et al., 2018), and ImageNet-21k (Deng et al., 2009) pre-trained ViT-Tiny (Wu et al., 2022)
• **Teacher:** CLIP ResNet-50 (Radford et al., 2021)
• **Active learning rounds:** 16
• **Initial setting:** 1-shot (single image per class)
• **Query size:** Equal to number of classes per round
• **Unlabeled pool:** All training samples except labeled set
• **Random seeds:** 5 different seeds
• **KD parameters:** $\zeta = 0.01$, $\eta = 2$, $\lambda = 0.5$
• **DHO parameters:** $\alpha = 0.5$, $\beta = 1$
• **Validation:** No validation split used | • **Epochs:** 200 for all rounds
• **Optimizer:** AdamW ($\beta_1 = 0.9$, $\beta_2 = 0.999$)
• **Learning rate:** $1 \times 10^{-3}$
• **Weight decay:** $1 \times 10^{-2}$
• **Batch size:** 128 (labeled: 64, unlabeled: 64) |
| *Few-Shot Teacher Active Distillation* | |
| **Model Configuration** | **Training Details** |
| • **Teacher:** CLAP (Silva-Rodriguez et al., 2024) on CLIP ResNet-50 (Radford et al., 2021) | • **Setup:** Following original CLAP paper (Silva-Rodriguez et al., 2024)
• **Modification:** Learning rate reduced from 0.1 to 0.01 for better convergence |

---

**Algorithm 5** Class-Balanced Selection

**Require:** Unlabeled pool $\mathcal{D}^{(u)} = \{x_1, \ldots, x_m\} \subset \mathbb{R}^d$, model outputs $f_r = \{f_r(x_1), f_r(x_2), \ldots, f_r(x_n)\}$, labeled dataset $\mathcal{D}^{(l)}$, query size $K$, number of classes $C$
**Ensure:** Selected set $S \subset \mathcal{D}^{(u)}$ with $|S| = K$
1: Initialize $S = \emptyset$
2: Set $y_i = \arg\max_c [f_r(x_i)]_c$ for all $x_i \in \mathcal{D}^{(u)}$
3: Count $n_c = |\{x_i \in \mathcal{D}^{(l)} : y_i = c\}|$ for each $c \in [C]$
4: Set weights $w_c = \frac{1}{n_c}$ if $n_c > 0$, else $w_c = 1$
5: Compute $K_c = \text{round}(\frac{w_c}{\sum_{j=1}^{C} w_j} \cdot K)$ for each $c$
6: Partition $\mathcal{D}^{(u)}$ into $\mathcal{D}_c^{(u)} = \{x_i \in \mathcal{D}^{(u)} : y_i = c\}$
7: **for** each class $c \in [C]$ **do**
8:     Add $\min(K_c, |\mathcal{D}_c^{(u)}|)$ random samples from $\mathcal{D}_c^{(u)}$ to $S$
9: **end for**
10: **return** $S$

**Implementation of Class-Balanced Selection.** We implement the Class-Balanced Selection algorithm (Algorithm 5) as our baseline method to address class distribution bias. The algorithm assigns pseudo-labels to unlabeled samples based on model predictions, analyzes class distribution in the labeled set, and computes inverse weights proportional to each class's representation—giving higher weights to underrepresented classes. It then allocates the query budget across classes according to these weights and randomly selects samples from each class partition. This approach effectively counteracts class imbalance by prioritizing underrepresented classes.

## C SUMMARY OF DATASETS

Table 3: Overview of datasets used in our experiments. Note that validation split is not used during the active learning process, and is only shown for completeness.

| Dataset | #Classes | #Train | #Val | #Test | Domain |
|---|---|---|---|---|---|
| Caltech101 (Fei-Fei et al., 2004) | 100 | 4,128 | 1,649 | 2,465 | General objects |
| DTD (Cimpoi et al., 2014) | 47 | 2,820 | 1,128 | 1,692 | Textures |
| EuroSAT (Helber et al., 2019) | 10 | 13,500 | 5,400 | 8,100 | Satellite |
| FGVCAircraft (Maji et al., 2013) | 100 | 3,334 | 3,333 | 3,333 | Aircraft |
| Food101 (Bossard et al., 2014) | 101 | 50,500 | 20,200 | 30,300 | Food |
| Flowers102 (Nilsback & Zisserman, 2008) | 102 | 4,093 | 1,633 | 2,463 | Plants |
| OxfordPets (Parkhi et al., 2012) | 37 | 2,944 | 736 | 3,669 | Animals |
| StanfordCars (Krause et al., 2013) | 196 | 6,509 | 1,635 | 8,041 | Vehicles |
| SUN397 (Xiao et al., 2010) | 397 | 15,880 | 3,970 | 19,850 | Scenes |
| UCF101 (Soomro et al., 2012) | 101 | 7,639 | 1,898 | 3,783 | Actions |
| ImageNet (Russakovsky et al., 2015) | 1,000 | 1.28M | - | 50,000 | General objects |

We provide the details of datasets we used in our experiments in this section. Our experimental evaluation encompasses 11 diverse datasets, including ImageNet (Russakovsky et al., 2015) and 10 additional datasets spanning various domains and classification challenges. These datasets represent a broad spectrum of visual recognition tasks including fine-grained classification across multiple domains: generic object recognition (Fei-Fei et al., 2004), automobile classification (Krause et al., 2013), flower species identification (Nilsback & Zisserman, 2008), aircraft categorization (Maji et al., 2013), pet breed classification (Parkhi et al., 2012), food recognition (Bossard et al., 2014), scene understanding (Xiao et al., 2010), texture classification (Cimpoi et al., 2014), satellite imagery analysis (Helber et al., 2019), and human action recognition (Soomro et al., 2012). Details of number of samples in each training, validation and test splits are illustrated in the Tab. 3.

We adhere to the train, validation, and test splits established in prior work (Kang et al., 2025). Our experimental protocol begins with a 1-shot setting, where only a single labeled example per class is available, with all remaining images in the training set treated as unlabeled. This approach extends to subsequent active learning rounds, where the unlabeled set consists of all training images not currently in the labeled set. To further simulate realistic constraints where validation data may be inaccessible, particularly in scenarios beginning with minimal labeled examples and progressively acquiring annotations, we conduct our experiments without utilizing the validation sets.

We further presents class balance statistics for each dataset in the Tab. 4 to provide additional insight into inherent class imbalances, a phenomenon that several researchers have addressed within active learning selection processes (Aggarwal et al., 2020; Bengar et al., 2022; Huang et al., 2024). It is important to note that our research addresses a distinct problem formulation; we primarily investigate teacher model bias rather than dataset bias. Notably, our findings demonstrate that the teacher bias emerges even in balanced datasets, showing the effectiveness of our approach in active knowledge distillation scenario.

Table 4: Class balance of datasets used in our experiments. The total number specified in the table is the number of samples of training splits as we do active selection on them.

| Dataset | Mean | Min | Max | Std | Total |
|---|---|---|---|---|---|
| Caltech101 (Fei-Fei et al., 2004) | 41.28 | 16 | 400 | 56.81 | 4,128 |
| DTD (Cimpoi et al., 2014) | 60.00 | 60 | 60 | 0.00 | 2,820 |
| EuroSAT (Helber et al., 2019) | 1,350.00 | 1,000 | 1,500 | 174.80 | 13,500 |
| FGVCAircraft (Maji et al., 2013) | 33.34 | 33 | 34 | 0.48 | 3,334 |
| Food101 (Bossard et al., 2014) | 500.00 | 500 | 500 | 0.00 | 50,500 |
| Flowers102 (Nilsback & Zisserman, 2008) | 40.13 | 20 | 129 | 22.20 | 4,093 |
| OxfordPets (Parkhi et al., 2012) | 79.57 | 74 | 80 | 1.26 | 2,944 |
| StanfordCars (Krause et al., 2013) | 33.21 | 19 | 54 | 3.47 | 6,509 |
| SUN397 (Xiao et al., 2010) | 40.00 | 40 | 40 | 0.00 | 15,880 |
| UCF101 (Soomro et al., 2012) | 75.63 | 58 | 97 | 10.72 | 7,639 |
| ImageNet (Russakovsky et al., 2015) | 1,281.17 | 732 | 1,300 | 70.22 | 1,281,167 |

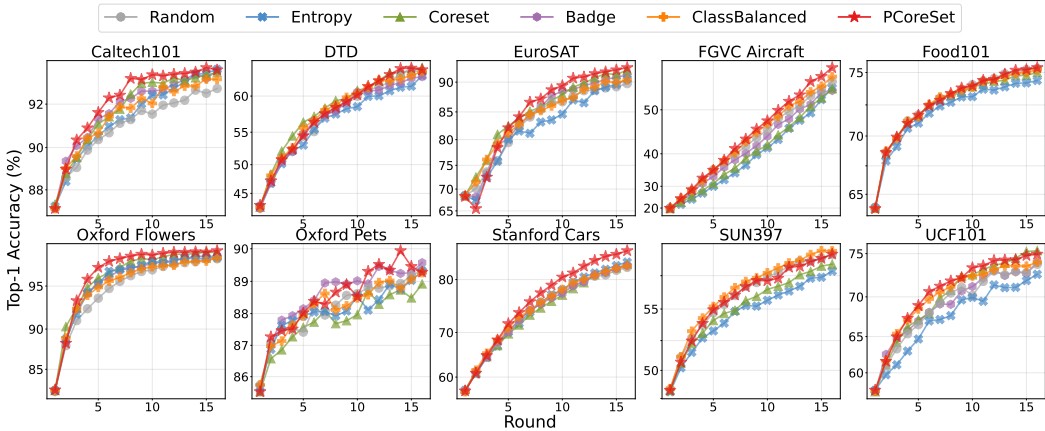

Figure 11: Results on 8 datasets using **MobileNetV2** under zero-shot distillation across 16 rounds.

# D   ADDITIONAL EXPERIMENTS

## D.1   EXPERIMENT ON MOBILENETV2

We further validate our methodology across diverse model architectures, including the lightweight MobileNetV2 (Sandler et al., 2018). The experimental results are presented in Fig. 11. For these experiments, we maintained the training configuration established for ResNet-18 (He et al., 2016), employing a ResNet-50 CLIP teacher (Radford et al., 2021) with identical hyperparameters. Our empirical results demonstrate that `PCoreSet` consistently outperforms alternative selection methods in the active distillation setting, exhibiting performance trends that align with our primary ResNet-18 (He et al., 2016) experiments.

## D.2   ACTIVE LEARNING WITHOUT KNOWLEDGE DISTILLATION

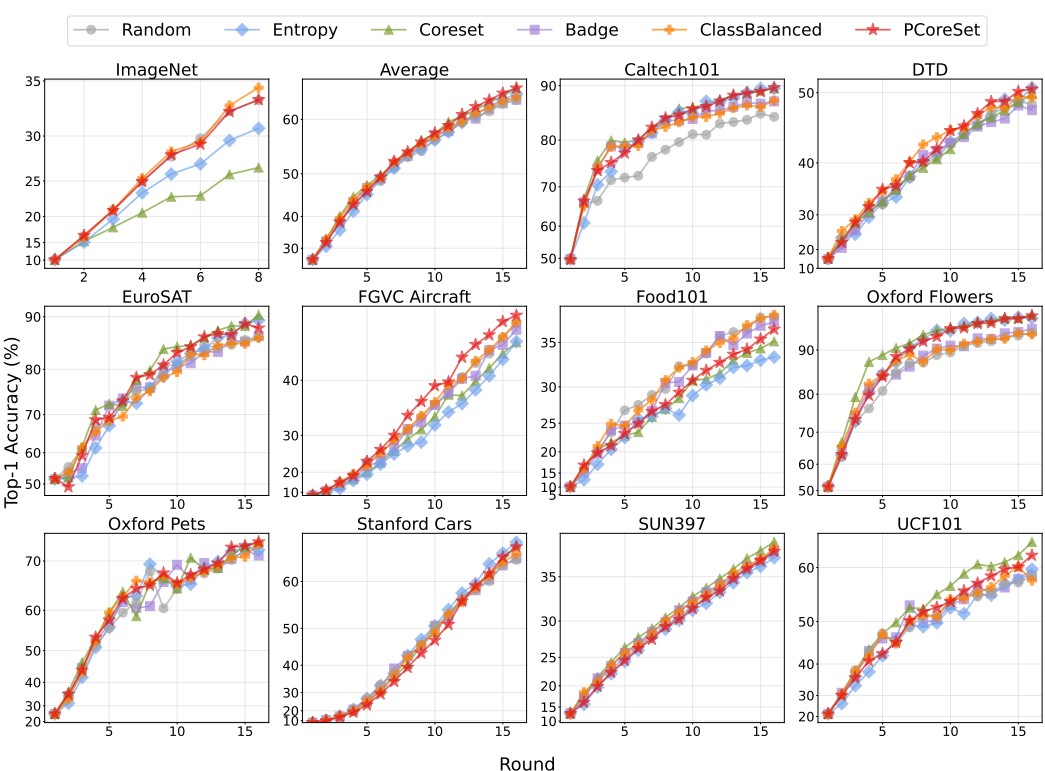

Figure 12: Results on 11 datasets including average of 10 datasets using **ResNet-18** without distillation.

The main purpose of our work is to propose utilizing a foundational teacher model directly in the knowledge distillation process (Hinton et al., 2015). Thus, we conducted active learning experiments without knowledge distillation as our baseline to enable direct comparison. Using ResNet-18 (He et al., 2016) and following our main experimental protocol in §B, we performed comparative analysis of knowledge distillation effects in the active learning process. The results are presented in Fig. 12. Interestingly, widely-adopted baselines such as uncertainty (Holub et al., 2008), coreset (Sener & Savarese, 2017), and badge (Ash et al., 2019) do not consistently outperform random or class-balanced selection methods Algorithm 5 on datasets such as FGVC (Maji et al., 2013), with our `PCoreSet` Algorithm 2 achieving superior performance. This occurs despite the approximately equal distribution of classes in these datasets as shown in Tab. 4 in the §C, suggesting that maintaining probabilistic balance may be beneficial even in traditional active learning scenarios. However, `PCoreSet` does not demonstrate clear effectiveness in this traditional active learning setting, indicating that maintaining probabilistic diversity is specifically effective in distillation scenarios, where it serves to leverage teacher's structured bias propagation.

## D.3 FEW-SHOT TEACHER DISTILLATION

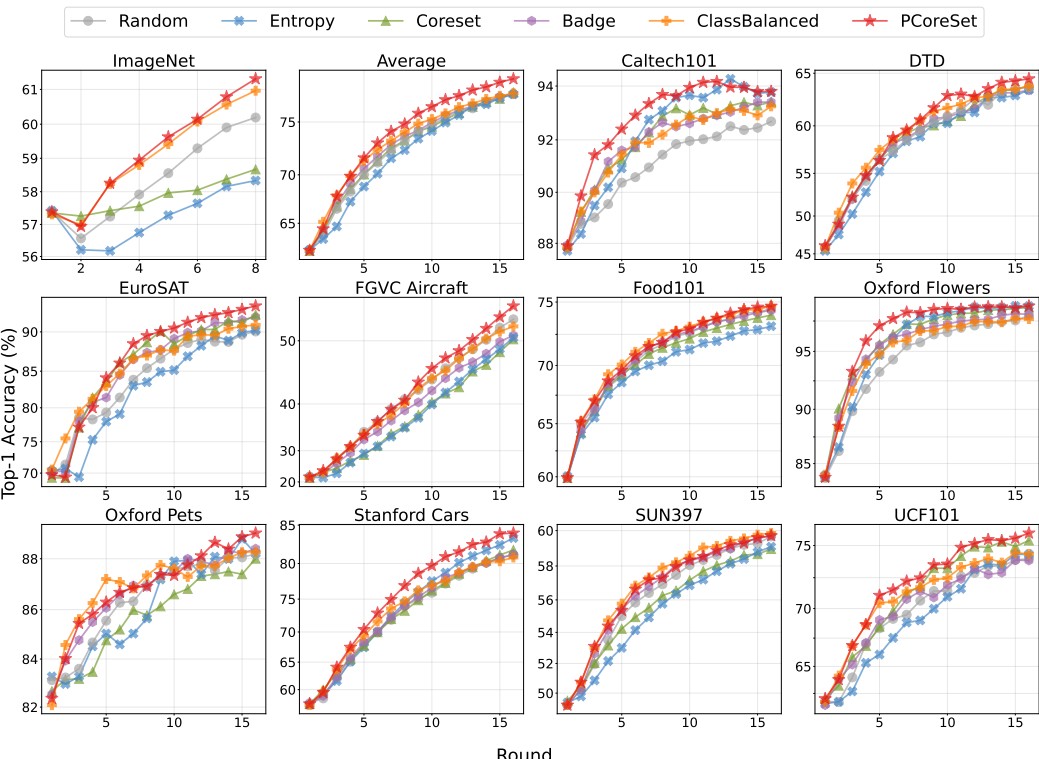

Figure 13: Results of **ResNet-18 student** on 11 datasets including average of 10 datasets under few-shot distillation.

In real-world applications, it is more natural to consider scenarios where both teacher and student models evolve concurrently. Specifically, we can apply few-shot learning methods to the generalist foundational teacher model with additional labeled samples acquired in each round, though our main experiments focused on zero-shot teachers to validate the effectiveness of integrating knowledge distillation into the active learning framework. Therefore, we extended our experiments to incorporate few-shot teachers that are updated with newly labeled samples in each acquisition round. For this purpose, we employed CLAP (Silva-Rodriguez et al., 2024), a few-shot learning method built upon CLIP that does not require a validation set—an appropriate choice for our setting where we assume extremely limited data availability (e.g., 1-shot training datasets) without access to validation data. The experimental results for student model performance are presented in Fig. 13, with corresponding teacher model performance shown in Fig. 14. In the few-shot teacher scenario, we observed consistent trends where probabilistic diversity effectively leverages bias, resulting in enhanced performance. Moreover, examining the few-shot teacher performance in Fig. 14 reveals that our `PCoreSet` outperforms alternative selection methods for teacher models as well. We attribute this to the fact that samples selected to leverage teacher bias naturally serve as beneficial examples for improving the teacher model, simultaneously enhancing performance on both the teacher and student sides.

## D.4 LOGIT ADJUSTMENT FOR CALIBRATED TEACHER PREDICTION

In this section, we investigate whether logit adjustment (Menon et al., 2020), a technique designed for imbalanced learning, can mitigate the structured prediction bias in teacher models. While logit adjustment (LA) provides a principled approach to handling class imbalance, we demonstrate that it is **orthogonal** to our `PCoreSet` method and can be used in conjunction to potentially enhance performance.

Logit adjustment (Menon et al., 2020) addresses **skewed, imbalanced** class distributions in training data by applying a **principled and effective** post-hoc adjustment based on empirical class frequencies.

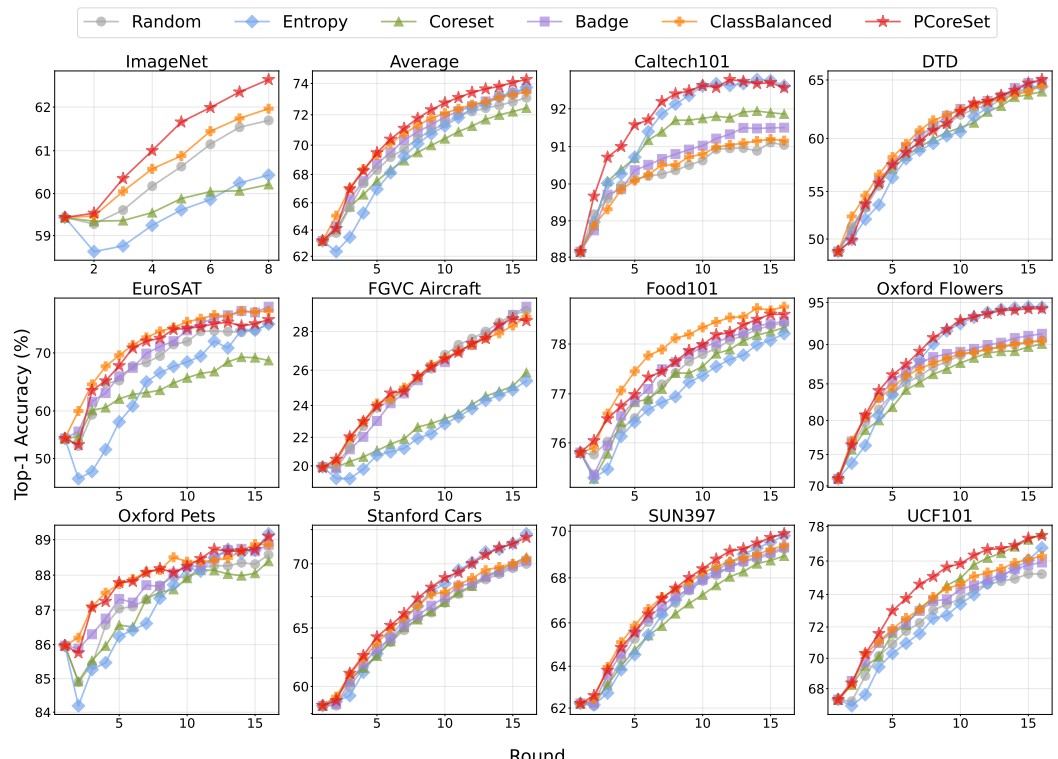

Figure 14: Results of **ResNet-50 few-shot teacher** on 11 datasets including average of 10 datasets under few-shot distillation.

Formally, $l' := l - \tau \log \pi \in \Delta^{C-1}$, where $l$ and $l'$ denote the original and adjusted logits, respectively. $\pi \in \Delta^{C-1}$ is the class prior estimated from the empirical distribution, i.e., $\pi := \mathbb{E}_{x \sim p(x)}[p(y \mid x)] \approx \frac{1}{N} \sum_{n=1}^{N} y_n$, with $y_n \in \{0,1\}^C$ one-hot labels in the supervised training dataset $\mathcal{D}^{(l)}$.

Given that teacher predictions in our setting are often skewed and biased, we explore whether adjusting **teacher logits** can provide an **improved inductive bias**. Our hypothesis is that PCoreSet can then leverage this calibrated teacher signal more effectively.

We consider two logit-adjustment variants:

1. **Hard LA.** Let $\mathcal{D} = \mathcal{D}^{(l)} \cup \mathcal{D}^{(u)}$. Estimate the class prior by **pseudo-label frequencies** on $\mathcal{D}$ with teacher predictions $f(x)$:

$$\pi_c \approx \frac{1}{|\mathcal{D}|} \sum_{x \in \mathcal{D}} \mathbf{1} \left[ \arg\max_{c'} [f(x)]_{c'} = c \right], \qquad \pi = [\pi_1, \ldots, \pi_C]^\top, \quad (23)$$

where $\mathbf{1}$ is an indicator function.

2. **Soft LA.** Estimate the class prior by the **average teacher probabilities**:

$$\pi \approx \frac{1}{|\mathcal{D}|} \sum_{x \in \mathcal{D}} f(x). \quad (24)$$

We conducted experiments on 8 of the 10 datasets by integrating LA into teacher predictions. Base results are averaged over 5 seeds, while Soft LA results are averaged over 3 seeds.

As detailed in Tab. 5, we observe that Soft LA **slightly improves PCoreSet** on average, even without exhaustive tuning of the hyperparameter $\tau$. These results indicate that while logit adjustment can provide marginal benefits, the core strength of PCoreSet lies in its ability to effectively leverage **structured teacher prediction bias** within the ActiveKD framework, yielding substantial gains across diverse datasets even without additional calibration techniques.

Table 5: Logit adjustment experiments across different active learning strategies. Numbers in parentheses show the difference from the baseline (no LA).

| Strategy | LA | Active Learning Round | | | | | | |
|---|---|---|---|---|---|---|---|---|
| | | 2 | 3 | 4 | 5 | 6 | 7 | 8 |
| Random | - | 65.1 | 67.4 | 69.5 | 71.3 | 72.4 | 73.9 | 74.5 |
| | Hard | 55.1(-10.1) | 59.9(-7.4) | 63.1(-6.5) | 66.0(-5.3) | 67.5(-5.0) | 69.4(-4.5) | 70.6(-3.9) |
| | Soft | 64.4(-0.7) | 67.6(+0.2) | 69.9(+0.3) | 71.5(+0.2) | 72.8(+0.4) | 74.0(+0.1) | 75.0(+0.5) |
| Coreset | - | 65.7 | 67.7 | 69.8 | 71.5 | 72.4 | 73.5 | 74.8 |
| | Hard | 55.5(-10.2) | 60.6(-7.1) | 63.6(-6.2) | 66.6(-4.9) | 68.8(-3.6) | 70.1(-3.4) | 71.9(-2.9) |
| | Soft | 65.3(-0.4) | 68.1(+0.4) | 70.4(+0.6) | 71.4(-0.1) | 73.0(+0.6) | 73.6(+0.2) | 75.3(+0.5) |
| Uncertainty | - | 63.8 | 66.5 | 67.9 | 69.8 | 71.1 | 72.4 | 73.4 |
| | Hard | 52.6(-11.2) | 57.4(-9.1) | 61.0(-6.9) | 63.7(-6.1) | 65.6(-5.5) | 68.5(-3.9) | 69.9(-3.5) |
| | Soft | 63.3(-0.5) | 66.3(-0.2) | 68.1(+0.2) | 70.0(+0.2) | 71.1(+0.0) | 72.3(-0.1) | 73.2(-0.2) |
| BADGE | - | 65.3 | 68.1 | 70.3 | 72.1 | 73.3 | 74.3 | 75.1 |
| | Hard | 54.6(-10.7) | 59.6(-8.4) | 63.7(-6.6) | 66.4(-5.7) | 68.0(-5.2) | 69.9(-4.4) | 71.1(-4.0) |
| | Soft | 64.5(-0.8) | 67.9(-0.2) | 70.2(-0.1) | 72.2(+0.1) | 73.4(+0.1) | 74.5(+0.2) | 75.6(+0.5) |
| ClassBalanced | - | 65.6 | 68.3 | 70.8 | 72.2 | 73.3 | 74.1 | 75.3 |
| | Hard | 55.4(-10.2) | 58.8(-9.4) | 62.6(-8.2) | 65.7(-6.5) | 67.2(-6.1) | 69.2(-4.9) | 70.3(-5.0) |
| | Soft | 65.1(-0.5) | 69.1(+0.8) | 71.3(+0.5) | 72.9(+0.7) | 74.1(+0.8) | 75.0(+0.9) | 75.7(+0.4) |
| **PCoreSet** | - | 65.7 | 68.6 | 71.0 | 72.8 | 74.1 | 75.3 | 76.2 |
| | Hard | 56.1(-9.5) | 61.8(-6.8) | 64.6(-6.4) | 67.8(-5.1) | 70.4(-3.6) | 71.7(-3.6) | 73.4(-2.8) |
| | Soft | **65.9**(+0.2) | **69.2**(+0.6) | **71.5**(+0.5) | **73.1**(+0.3) | **74.1**(+0.1) | **75.3**(+0.01) | **76.3**(+0.01) |

Table 6: Results on 8 datasets (average) when **excluding unlabeled data from KD** (denoted as w/o unlabeled).

| Method | KD | Round | | | | | | | | | | | | | | |
|---|---|---|---|---|---|---|---|---|---|---|---|---|---|---|---|---|
| | | 2 | 3 | 4 | 5 | 6 | 7 | 8 | 9 | 10 | 11 | 12 | 13 | 14 | 15 | 16 |
| Random | No Distill | 36.70 | 42.64 | 47.65 | 50.93 | 53.77 | 56.69 | 59.24 | 60.15 | 62.46 | 64.06 | 65.62 | 66.49 | 67.60 | 68.90 | 69.51 |
| | ActiveKD (w/o unlabeled) | 43.33 | 49.58 | 55.20 | 58.79 | 61.87 | 63.90 | 66.17 | 67.72 | 69.30 | 70.68 | 71.86 | 72.84 | 73.88 | 74.15 | 75.00 |
| | ActiveKD | 64.54 | 66.82 | 69.01 | 70.77 | 71.97 | 73.47 | 74.11 | 74.79 | 75.81 | 76.15 | 76.89 | 77.38 | 77.57 | 78.02 | 78.67 |
| Entropy | No Distill | 34.83 | 40.60 | 46.39 | 50.96 | 54.77 | 57.18 | 60.14 | 61.43 | 63.07 | 64.54 | 66.32 | 67.56 | 69.21 | 70.30 | 71.29 |
| | ActiveKD (w/o unlabeled) | 41.09 | 48.59 | 55.80 | 57.51 | 61.18 | 65.13 | 66.90 | 68.71 | 69.76 | 70.98 | 72.80 | 74.11 | 74.77 | 76.30 | 76.67 |
| | ActiveKD | 63.82 | 66.52 | 67.94 | 69.82 | 71.09 | 72.42 | 73.41 | 74.33 | 75.13 | 75.72 | 76.48 | 76.94 | 77.46 | 78.04 | 78.44 |
| Coreset | No Distill | 37.28 | 44.90 | 50.44 | 53.24 | 55.69 | 57.93 | 60.50 | 62.96 | 63.97 | 66.40 | 67.30 | 68.29 | 69.56 | 70.62 | 72.05 |
| | ActiveKD (w/o unlabeled) | 44.21 | 51.05 | 55.31 | 59.84 | 62.98 | 65.14 | 67.55 | 69.25 | 71.14 | 71.11 | 72.90 | 74.15 | 74.97 | 75.58 | 76.67 |
| | ActiveKD | 65.70 | 67.68 | 69.82 | 71.47 | 72.40 | 73.46 | 74.76 | 75.45 | 76.27 | 76.49 | 76.99 | 77.57 | 78.33 | 78.53 | 78.99 |
| Badge | No Distill | 36.61 | 42.58 | 48.73 | 52.44 | 54.67 | 57.70 | 59.37 | 61.28 | 63.47 | 64.07 | 66.21 | 66.43 | 68.05 | 69.06 | 69.60 |
| | ActiveKD (w/o unlabeled) | 43.43 | 50.42 | 56.56 | 59.17 | 61.60 | 63.91 | 66.44 | 67.32 | 68.94 | 70.54 | 71.48 | 72.16 | 73.09 | 74.47 | 74.91 |
| | ActiveKD | 65.29 | 68.05 | 70.31 | 72.12 | 73.27 | 74.29 | 75.11 | 75.80 | 76.44 | 76.88 | 77.44 | 77.84 | 78.08 | 78.71 | 78.92 |
| ClassBalanced | No Distill | 37.07 | 44.17 | 49.17 | 52.68 | 54.70 | 58.37 | 59.88 | 61.71 | 63.04 | 64.77 | 65.80 | 67.33 | 68.34 | 69.37 | 70.02 |
| | ActiveKD (w/o unlabeled) | 43.40 | 51.31 | 57.32 | 59.93 | 63.86 | 65.34 | 67.77 | 68.62 | 70.03 | 72.07 | 72.53 | 74.15 | 74.47 | 75.18 | 76.10 |
| | ActiveKD | 65.55 | 68.28 | 70.77 | 72.18 | 73.34 | 74.11 | 75.30 | 75.78 | 76.33 | 76.93 | 77.47 | 77.81 | 78.11 | 78.59 | 78.95 |
| **PCoreSet** | No Distill | 35.97 | 43.17 | 48.45 | 51.74 | 55.24 | 58.81 | 60.72 | 62.67 | 64.31 | 65.61 | 67.70 | 69.02 | 70.10 | 71.21 | 71.95 |
| | ActiveKD (w/o unlabeled) | 43.00 | 50.06 | 56.36 | 60.23 | 63.99 | 66.08 | 68.00 | 69.65 | 71.44 | 72.94 | 73.84 | 74.02 | 76.00 | 76.72 | 77.31 |
| | ActiveKD | 65.67 | 68.61 | 71.00 | 72.81 | 74.05 | 75.31 | 76.24 | 77.15 | 77.63 | 78.17 | 78.76 | 79.15 | 79.76 | 80.08 | 80.44 |

## D.5 ABLATION STUDY ON EFFECTIVENESS OF UNLABELED DATA FOR KD

To investigate the contribution of unlabeled data to knowledge distillation (KD) under ACTIVEKD, we conduct additional experiments by implementing **ActiveKD (w/o unlabeled)**, which excludes the unlabeled set $\mathcal{D}^{(u)}$ from the KD loss $\mathcal{L}_{\text{KD}}$ in Eq. 3. Specifically, under this variant, the KD loss reduces to $\mathcal{L}_{\text{KD}} = \frac{1}{N} \sum_n D_{\text{KL}} \left[ f(x_n^{(l)}) \| f_r(x_n^{(l)}) \right]$, without the unlabeled component $\frac{1}{M} \sum_m D_{\text{KL}} \left[ f(x_m^{(u)}) \| f_r(x_m^{(u)}) \right]$. We follow the same experimental setup as in Fig. 6, using zero-shot teachers and 8 out of the 10 datasets. Tab. 6 reports the results for three configurations: (1) **No Distill**, (2) **ActiveKD (w/o unlabeled)**, and (3) **ActiveKD**. Here, (2) serves as an ablation to isolate the effect of unlabeled data during KD. We observe that **ActiveKD (w/o unlabeled)** consistently improves over **No Distill** across all selection algorithms. However, the improvement is noticeably smaller than full **ActiveKD**. For example, at the 16th AL round with PCoreSet, the gain is 5.36 points, compared to 8.49 points with full ACTIVEKD. More importantly, even under this restricted setting, PCoreSet achieves the best performance in **11 out of 15** rounds (73.3%), demonstrating its robustness and effectiveness in *semi-supervised knowledge distillation using labeled samples only*.

# E  ADDITIONAL QUALITATIVE RESULTS

## E.1  HEATMAP OF SELECTED SAMPLES

We visualize the distribution of selected samples using heatmaps in Fig. 15.

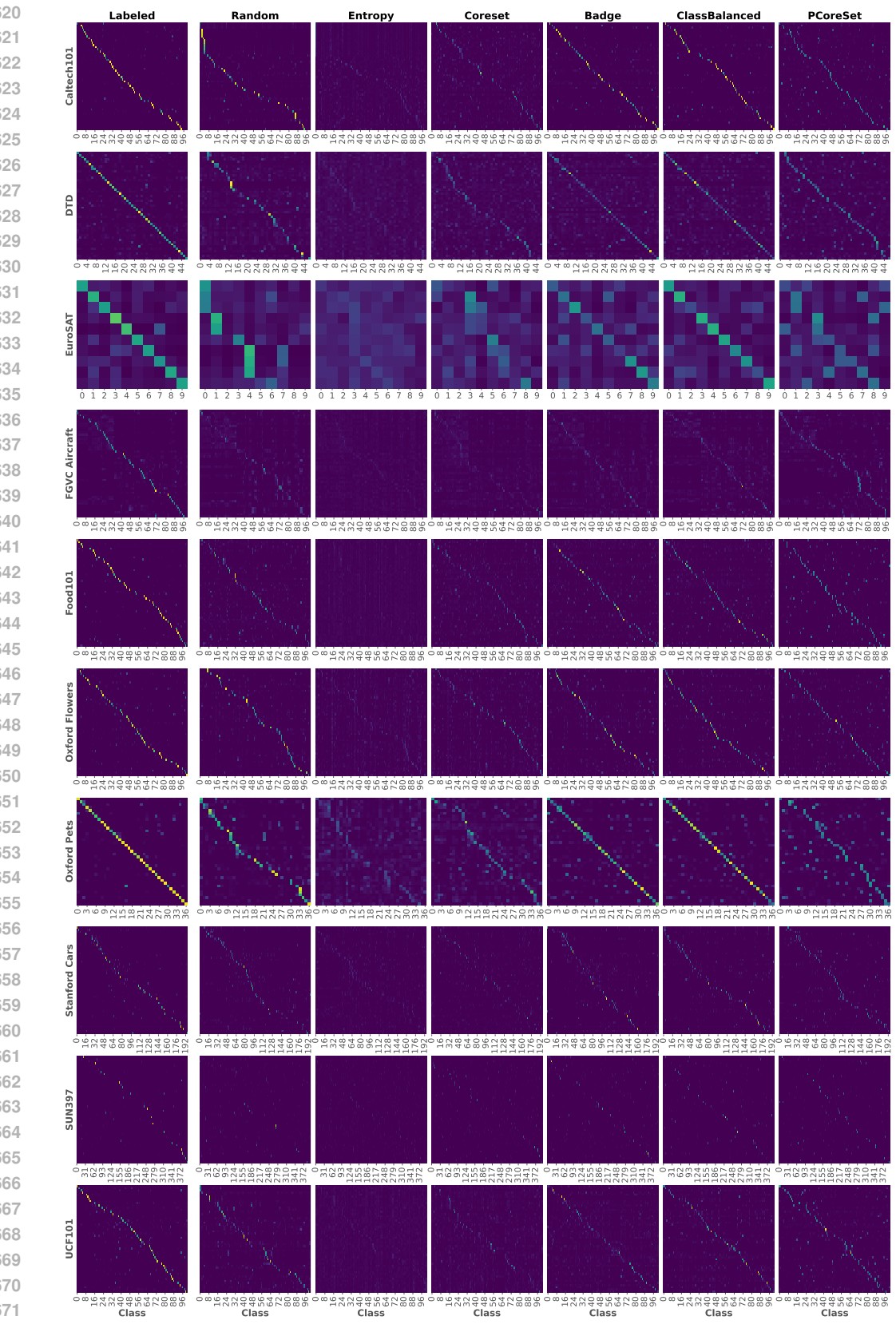

Figure 15: The heatmap of output probability vectors from different selection strategies in the first active learning round using 10 datasets.

## E.2 Qualitative Results on Selected Samples from PCoreSet

We visualize samples selected by **PCoreSet** from various datasets in Figures 16 through 22, showcasing how **PCoreSet** selects diverse and representative examples across different visual domains. These visualizations provide qualitative evidence of **PCoreSet**'s effectiveness in identifying informative samples that help leverage the structured bias of the teacher model.

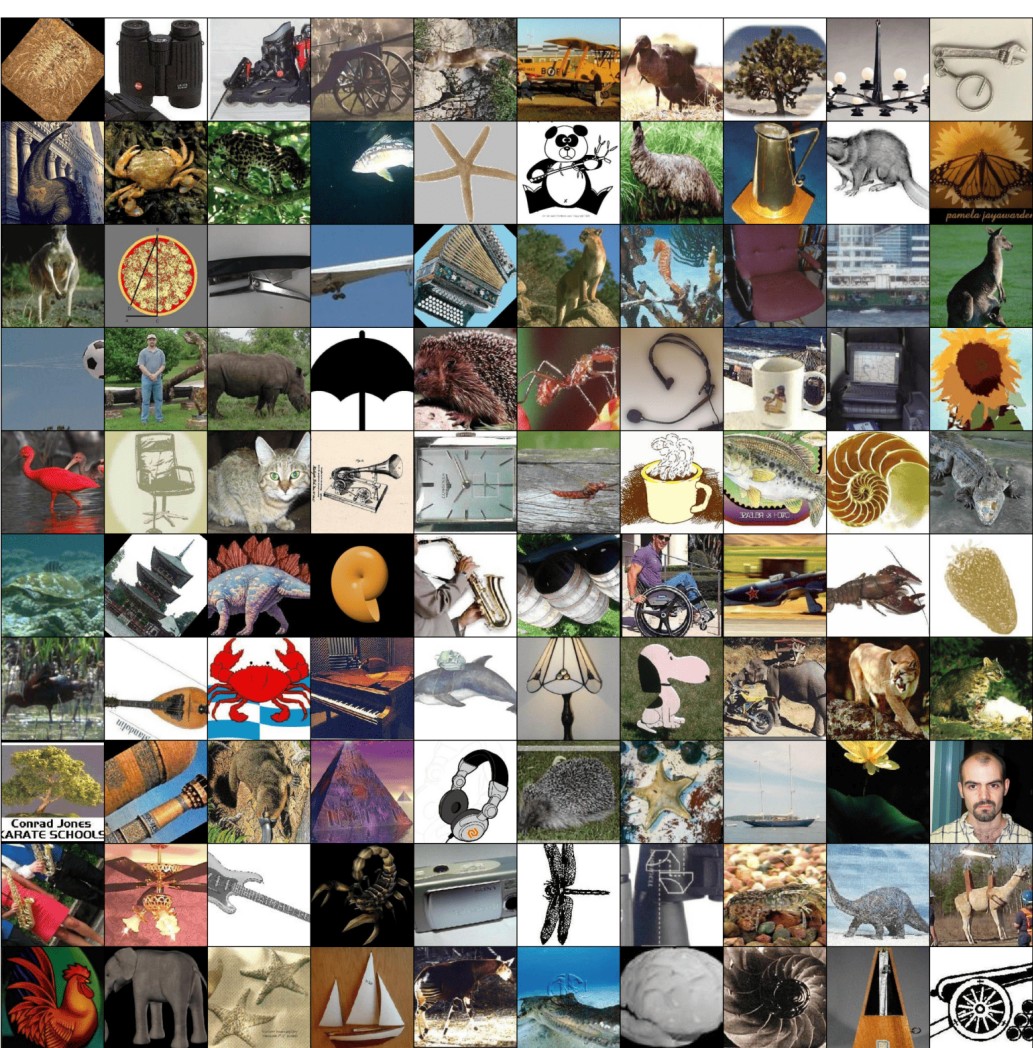

Figure 16: Visualization of samples selected by **PCoreSet** for Caltech101 dataset.

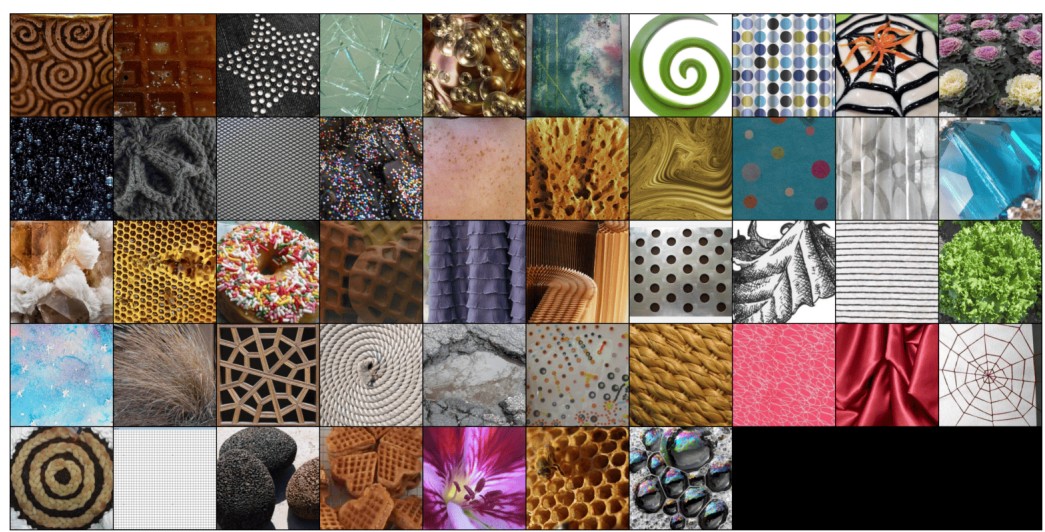

Figure 17: Visualization of samples selected by **PCoreSet** for DTD dataset.

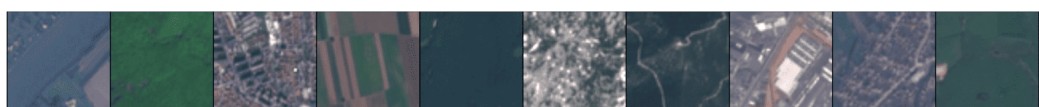

Figure 18: Visualization of samples selected by **PCoreSet** for EuroSAT dataset.

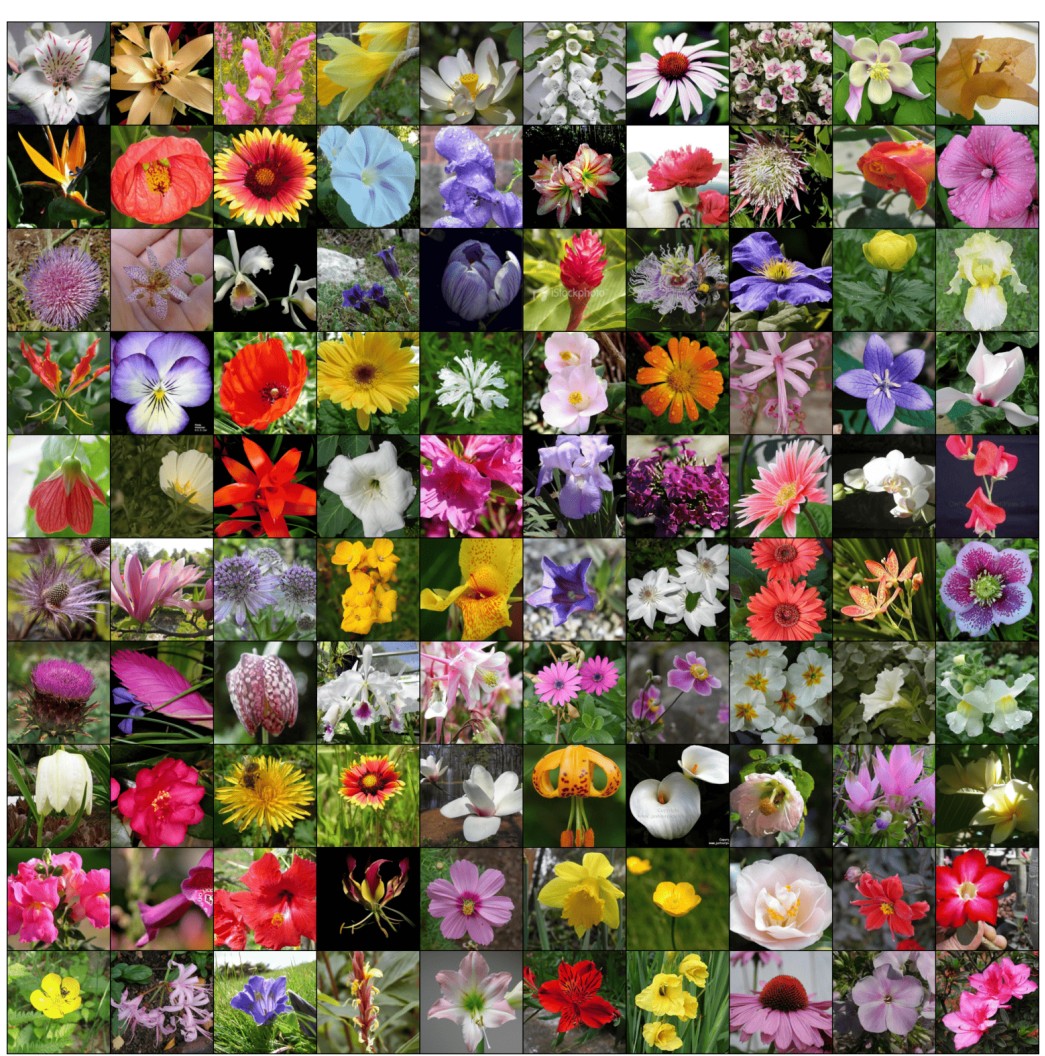

Figure 19: Visualization of samples selected by `PCoreSet` for Flowers102 dataset.

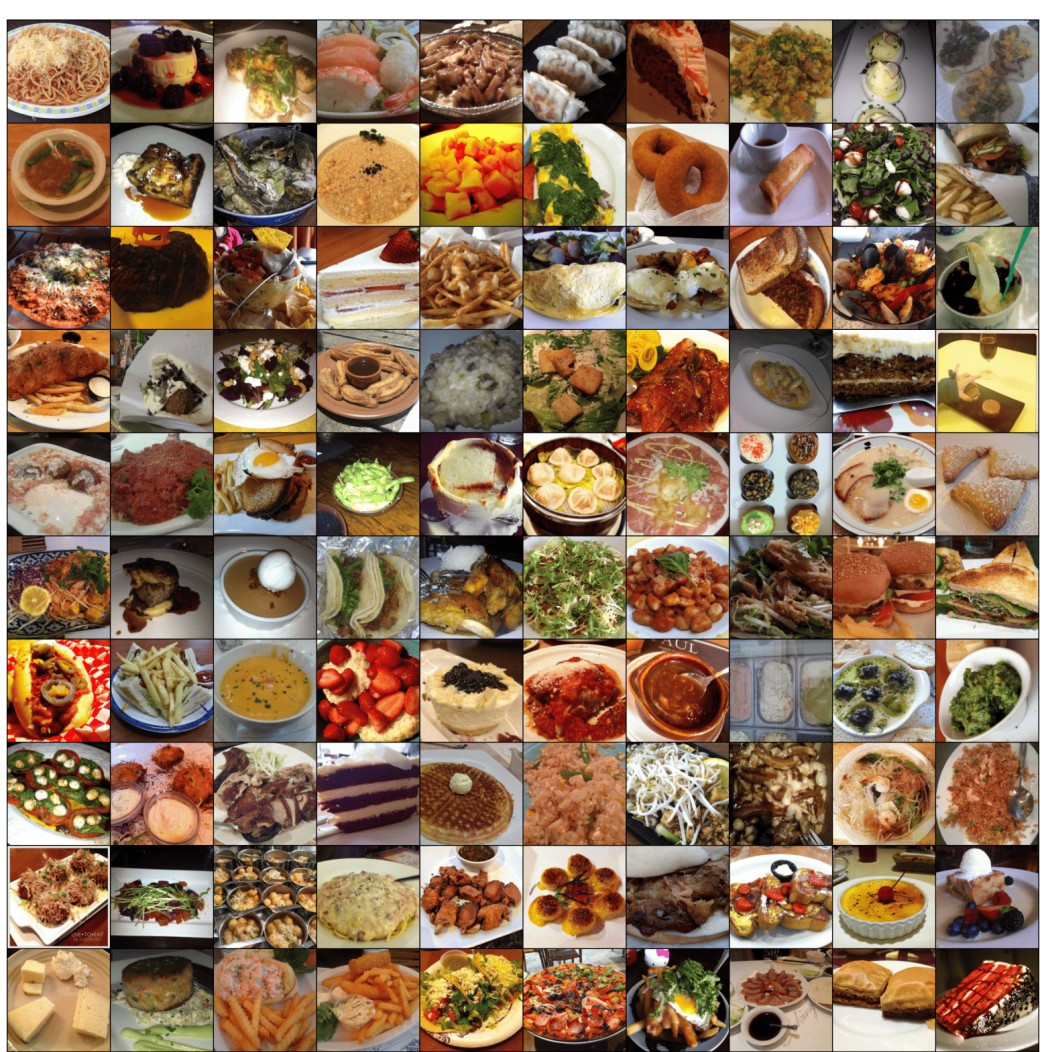

Figure 20: Visualization of samples selected by `PCoreSet` for Food-101 dataset.

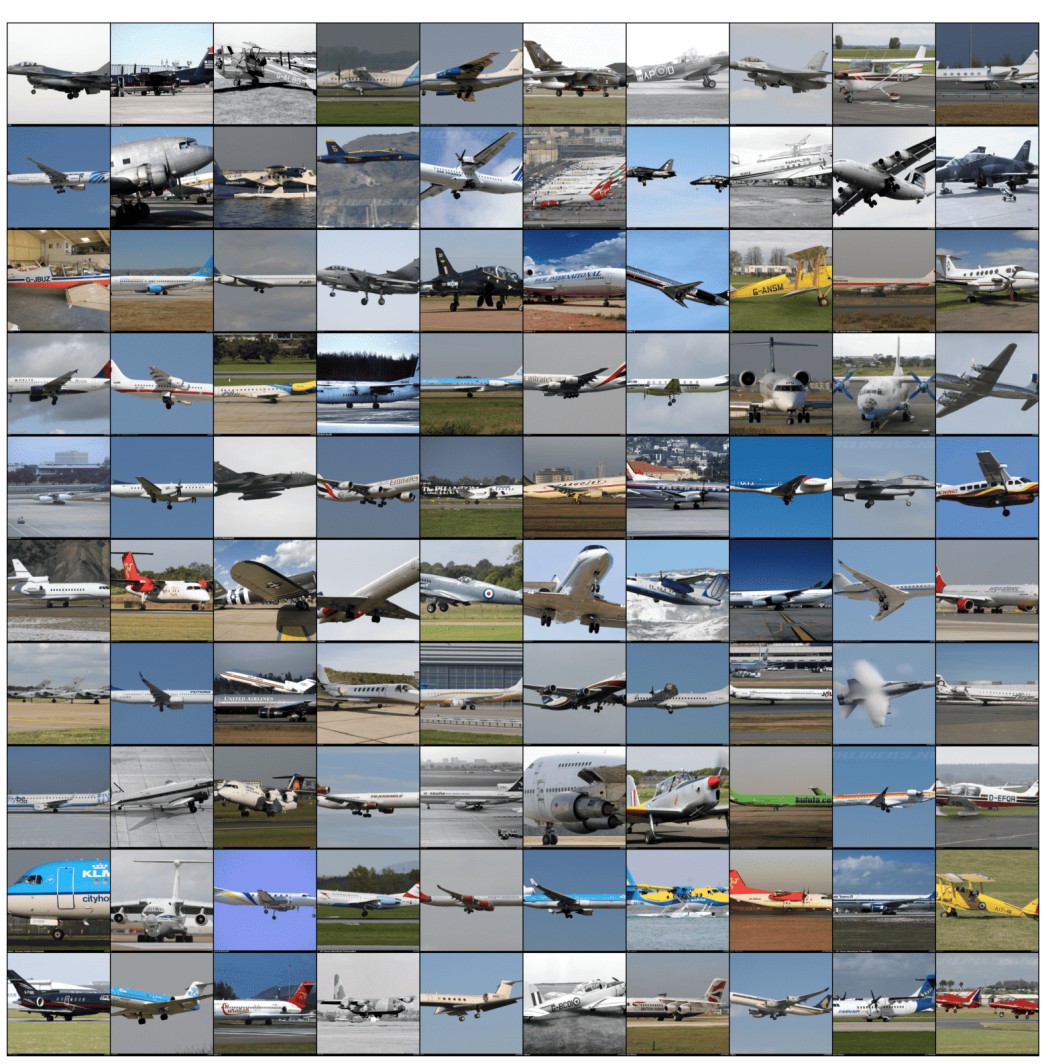

Figure 21: Visualization of samples selected by **PCoreSet** for FGVC dataset.

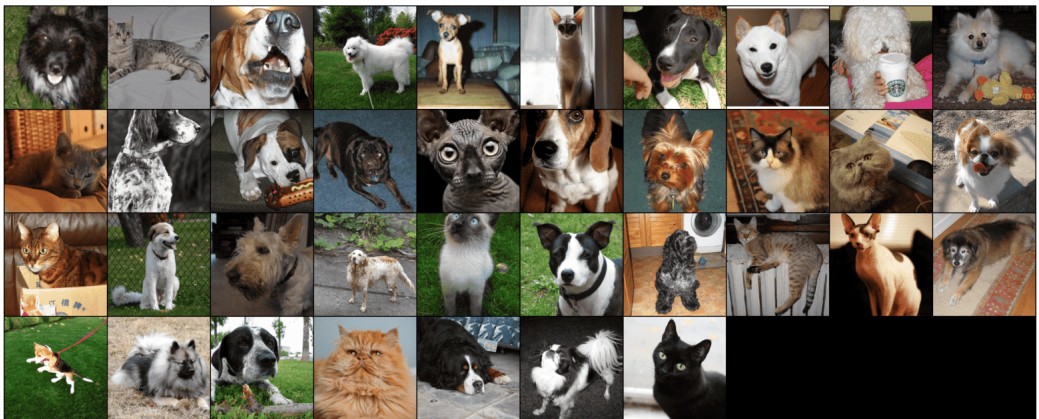

Figure 22: Visualization of samples selected by **PCoreSet** for Oxford-IIIT Pets dataset.

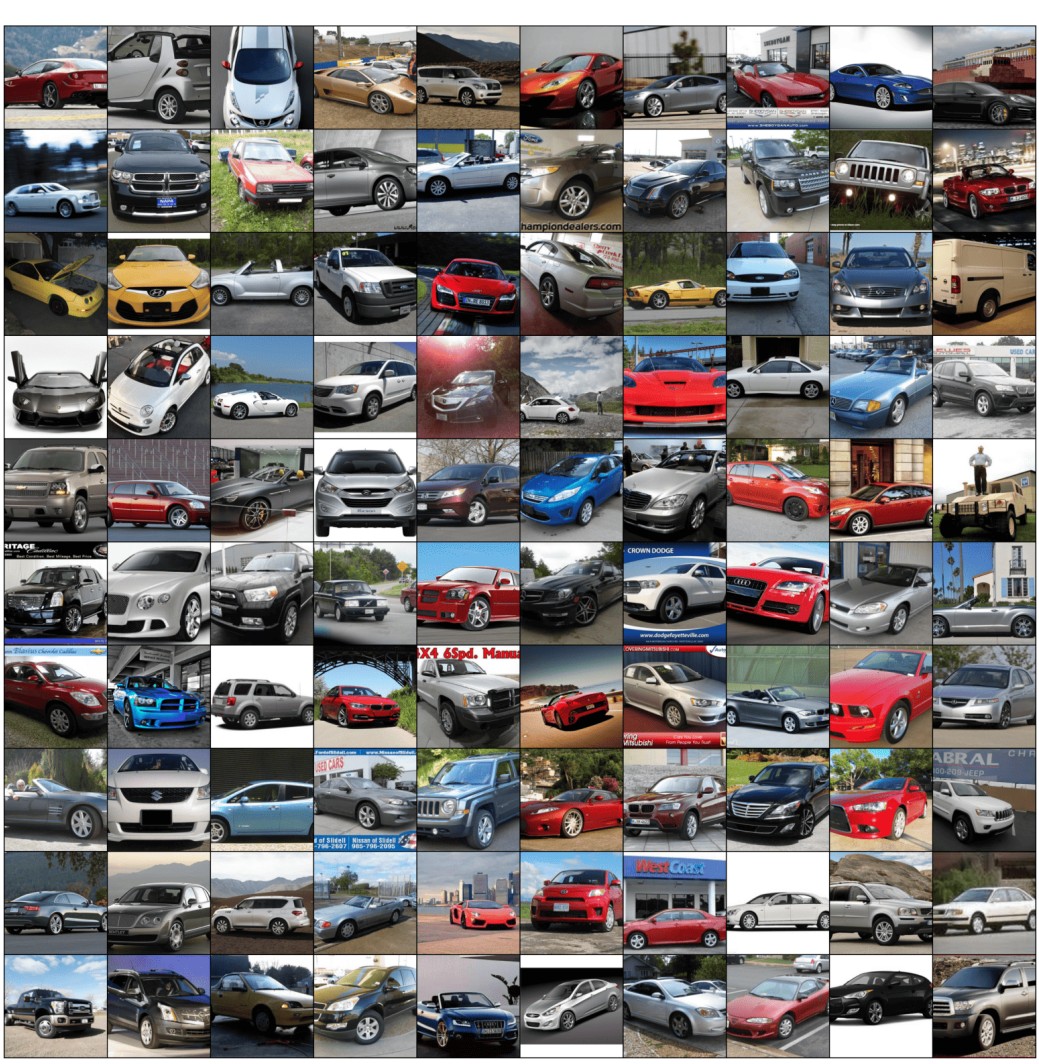

Figure 23: Visualization of samples selected by **PCoreSet** for Stanford Cars dataset.

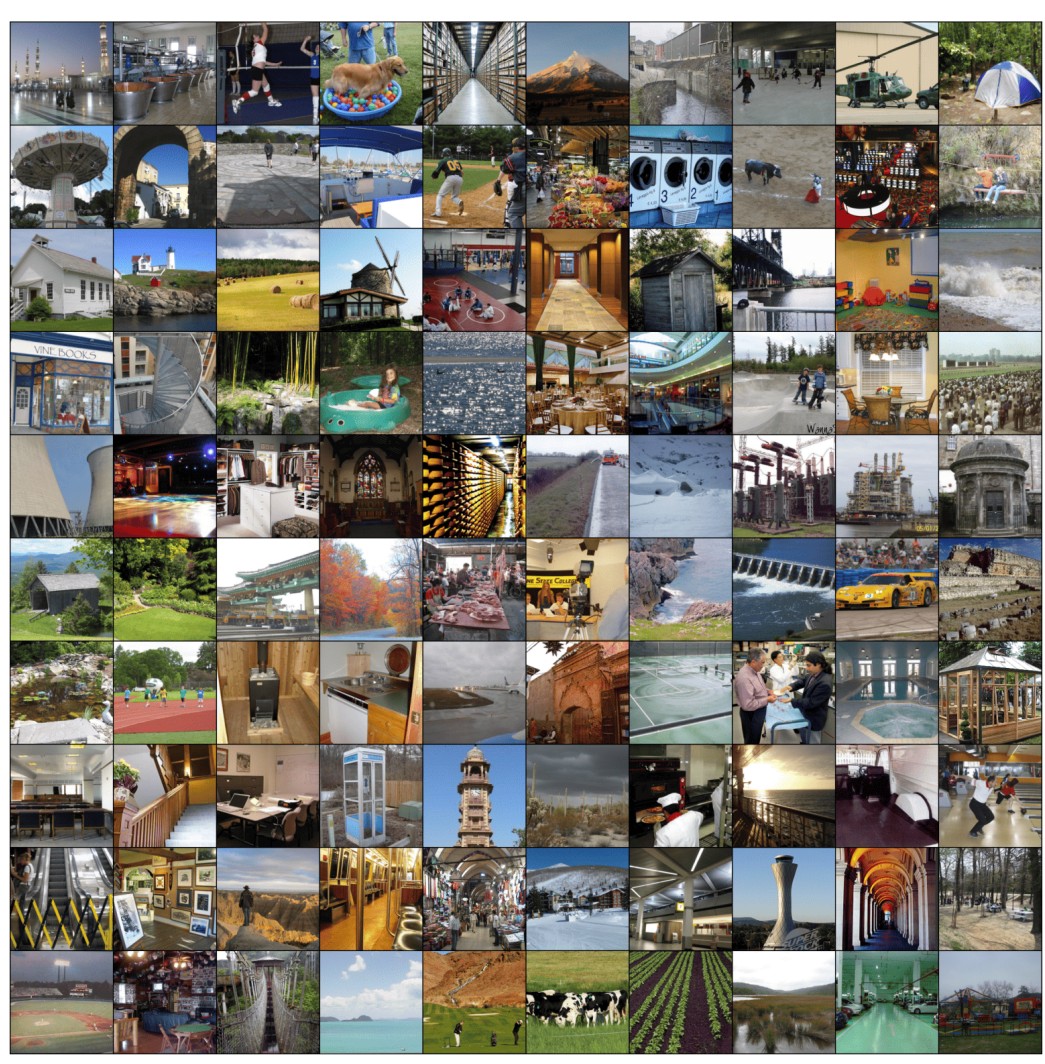

Figure 24: Visualization of samples selected by `PCoreSet` for SUN397 dataset.

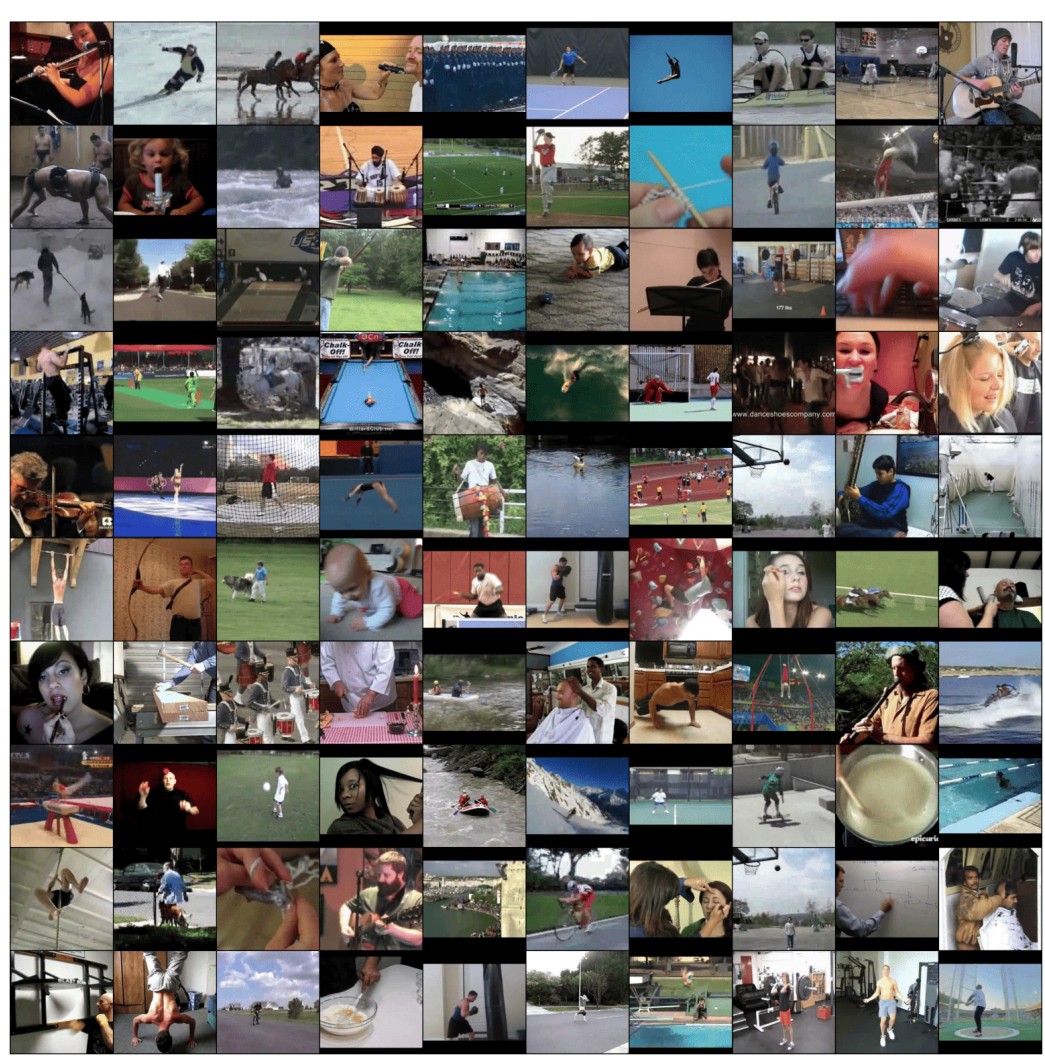

Figure 25: Visualization of samples selected by `PCoreSet` for UCF101 dataset.

# F   THE USE OF LLMs

We used LLMs solely for light editing such as correcting grammatical errors and polishing some words. They did not contribute to research ideation, experiments, analysis, or substantive writing.

