# OpenReview forum: "PCoreSet: Effective Active Learning through Knowledge Distillation from Vision-Langauge Models"
_ICLR.cc/2026/Conference — Submitted to ICLR 2026_

### Official Review · Reviewer_YbDT · 2025-10-29

**Soundness:** 3
**Presentation:** 3
**Contribution:** 2
**Rating:** 4
**Confidence:** 4

**Summary:**

The paper proposes ActiveKD, an active-learning framework that distills knowledge from a zero/few-shot vision–language model (VLM) into a compact student during each AL round, and introduces PCoreSet, a probability-space coreset selection rule that greedily maximizes the minimum l2 distance between a candidate's student prediction and those of labeled points. The authors argue VLM predictions exhibit structured clustering in the probability simplex and provide a simple analysis that such structure propagates to the student under distillation. Extensive experiments on 11 datasets show large gains from ActiveKD over "No Distill", and PCoreSet often ranks best among baselines.

**Strengths:**

1. The idea of employing VLM is advanced. The paper offers a theoretical argument for how the VLM's knowledge structure is passed to the student. It nicely connects this idea directly to the PCoreSet algorithm's goal of covering the probability space.
2. The PCoreSet selection rule is simple and computationally cheap. This makes it practical and should be easy to integrate into existing active learning pipelines.
3.  Strong empirical improvements from enabling KD with a VLM across many datasets.

**Weaknesses:**

1. The core concept of combining active learning with knowledge distillation from a powerful teacher, is a natural combination and not entirely new. The main algorithmic piece, using a VLM's outputs to select diverse data, is intuitive but also quite simple. It follows a classic active learning strategy, coreset, with the main novelty being the use of a VLM teacher and the focus on probability outputs rather than feature representations. The theory, while a nice addition, mainly formalizes an existing intuition rather than breaking new ground. Therefore, the novelty is limited.

2. When a powerful VLM is available, the empirical gains are substantial, so the framework could be practically valuable. However, these improvements are largely attributable to adding a strong teacher rather than to the selection rule itself; the marginal benefit of PCoreSet over other selectors is smaller and sometimes inconsistent in early rounds.

3. The method's simplicity is a double-edged sword: it's easy to implement, but the core idea of using a VLM to pseudo-label and select diverse outputs is fairly expected; the results match what one would anticipate when injecting a much stronger teacher into an AL loop.

**Questions:**

Can you provide computational analyses?

---

> ### Author Response · Authors · 2025-11-21
> **Response to Reviewer YbDT (1/4)**
>
> We sincerly appreciate you for your time and constructive comments which improve our paper. We respectfully address your concerns as following:
>
> ---
>
> > **[Q1-1]** The core concept of combining active learning with knowledge distillation from a powerful teacher is a natural combination and not entirely new.
>
> - We respectfully agree that combining active learning (AL) and knowledge distillation (KD) may appear **natural and intuitive**. However, we argue that integrating these two paradigms in our problem setting is **non-trivial** and more importantly **unexplored in prior work**.
>
> - In particular, AL and KD are **inherently difficult to combine** due to mismatches in their **underlying assumptions**. KD typically relies on **abundant labeled data** and a **well-trained, task-specific teacher**, whereas AL assumes **data-scarce regimes** with very limited or even zero initial labels, where such **task-specific teachers do not exist**.
>
> - As discussed in `L65–77`, only the **recent emergence of zero-/few-shot vision–language models (VLMs)** makes this integration feasible, as these models can serve as effective teachers **without sufficient task-specific supervision**. To the best of our knowledge, **this AL–KD integration using VLMs has not been explored** in prior work (`L141–156`), making our work the first to systematically examine this setting.
>
> - In summary, while the proposed ActiveKD framework may appear intuitive, it represents an **non-trivial and unexplored integration** of AL and VLM-based KD in label-scarce scenarios.
>
> ---
>
> > **[Q1-2]** The main algorithmic piece, using a VLM's outputs to select diverse data, is intuitive but also quite simple. It follows a classic active learning strategy (coreset), with the main novelty being the use of a VLM teacher and the focus on probability outputs rather than feature representations. The theory mainly formalizes an existing intuition rather than breaking new ground. Therefore, the novelty is limited.
>
> - We sincerely appreciate the reviewer for raising this concern. While the approach may appear **intuitive**, we respectfully argue that our contribution lies in **identifying and formalizing an unexplored challenge within the proposed ActiveKD framework**, and in proposing a **simple method specifically tailored to address this challenge**.
>
> - We also believe that our perspective, which shifts the selection criterion from **feature space to probability space**, is **well-motivated** and provides **meaningful insight** that has not been explored in prior AL work:
>
>     1. Prior work has reported **class imbalance** in VLM predictions [1]. We further observe that this imbalance **induces a geometric structure** in the probability simplex, forming distinct clusters (see `Fig. 4b`). This structure has direct implications for active sample selection, which, to our knowledge, has **not been investigated previously**.
>
>     2. We formalize this phenomenon as **structured prediction bias** (`Definition 1`) and show that it can **propagate through KD** from teacher VLMs to student models (`Proposition 1`). We also provide **empirical evidence** of this propagation in `Fig. 4c`.
>
>     3. Instead of mitigating this bias, we treat it as an **inductive bias** and propose **$\texttt{PCoreSet}$**, which **exploits this structure** for more effective AL query selection. This differs fundamentally from classic coreset methods that rely solely on feature-space geometry.
>
> - In conclusion, together with our response to **[Q1-1]**, our contribution lies in **identifying and formalizing a previously underexplored challenge** in combining AL and VLM-based KD, and in **introducing a well-motivated, specifically tailored method** that leverages structured prediction bias for sample-efficient active learning.
>
> ---

---

> ### Author Response · Authors · 2025-11-21
> **Response to Reviewer YbDT (2/4)**
>
> ---
>
> > **[Q2-1]** When a powerful VLM is available, the empirical gains are substantial, so the framework could be practically valuable. However, these improvements are largely attributable to adding a strong teacher rather than to the selection rule itself.
>
> - We sincerely appreciate the acknowledgment of the **effectiveness of the proposed ActiveKD** framework. We also agree that a substantial portion of the performance improvement comes from the **zero-/few-shot capabilities of VLM teachers**. However, we respectfully argue that the benefit of $\texttt{PCoreSet}$ is **not marginal** but **significant** for the following reasons:
>
>     - First, we recap the scope of our evaluation. Under **the same ActiveKD setup** (identical teacher, student architecture, hyperparameters, and query budgets), we compare $\texttt{PCoreSet}$ against existing AL selection methods across **73 configurations**, using **4 different VLM teachers** (CLIP ResNet-50, OpenCLIP ResNet-50, and CLIP ViT-L) and **5 different student architectures** (ResNet-18, ResNet-50, ViT-B, TinyViT, and MobileNet).
>
>     - Across these **fair** and **extensive** comparisons, as noted in the caption of `Fig. 5` and `L408-410`, $\texttt{PCoreSet}$ achieves the **best performance in 64 out of 73 settings (87.7%)**, which we believe is **substantial**, especially given that **only the data selection strategy differs** across methods.
>
>     - Moreover, as shown in `Fig. 6`, excluding $\texttt{PCoreSet}$, there is **no clear winner** among existing all AL baselines (Random, Entropy, Coreset, Badge, and ClassBalances), as they exhibit **dataset-dependent performance variations**.
>
>     - In practice, this means that practitioners would need to **run multiple strategies** to empirically identify the best-performing one for each target dataset. In contrast, $\texttt{PCoreSet}$ **consistently delivers strong performance across diverse datasets**, removing the need for manual effort in such exhaustive strategy searches.
>
> - We believe that the above discussion clearly addresses the concern regarding the marginal improvement of $\texttt{PCoreSet}$, and we have included the above discussion in `L401-410`.
>
> ---
>
> > **[Q2-2]** The marginal benefit of PCoreSet over other selectors sometimes inconsistent in early rounds.
>
> - We sincerly appreciate the reviewer for the observation. We would like to clarify that the occasional inconsistency in early AL rounds is an **expected and explainable behavior** rather than a fundamental limitation of $\texttt{PCoreSet}$:
>
>     - As noted in `§3.3`, $\texttt{PCoreSet}$ is designed to **exploit the structured prediction bias** of VLM teachers. However, VLM teachers are naturally **not perfect** and may exhibit **systematic errors** in zero-/few-shot settings. In the **very early rounds**, there are **insufficient ground-truth labels** to correct these teacher biases, which can lead to suboptimal selection.
>
>     - The **imperfection of VLM teachers** is empirically supported in `Fig. 1 (right)`. The results show that a student model trained with ActiveKD quickly **outperforms the zero-/few-shot teacher** after only **one or two AL rounds** on average across 10 datasets. This indicates that VLM teachers provide valuable **initial guidance**, but are not perfect oracles. $\texttt{PCoreSet}$ effectively **leverages this imperfect teacher signal** by selecting samples that better cover the structured probability space.
>
>     - This behavior is therefore **not inconsistent** with the design rationale, but rather an **expected outcome of adopting imperfect teachers** under limited supervision.
>
>     - Despite occasional early-round variation, $\texttt{PCoreSet}$ achieves the **best performance in 87.7% of configurations (64 out of 73)** across diverse architectures and datasets, demonstrating consistent improvements over existing AL baselines and validating its effectiveness as a practical AL selection strategy.
>
> - We believe that the above discussion clearly addresses the concern regarding the marginal improvement of $\texttt{PCoreSet}$, and we have included the above discussion in `L401-410`.
>
> ---

---

> ### Author Response · Authors · 2025-11-21
> **Response to Reviewer YbDT (3/4)**
>
> ---
>
> > **[Q3-1]** The method's simplicity is a double-edged sword: it is easy to implement, but the core idea of using a VLM to pseudo-label and select diverse outputs is fairly expected.
>
> - We sincerely appreciate the reviewer for acknowledging both the simplicity and the intuitive nature of our core idea. However, we respectfully argue that selecting diverse outputs under ActiveKD is **non-trivial**, and the key challenge lies in **how diversity should be defined** and **why certain definitions fail**, as illustrated by the example of Coreset [2]:
>
>     - Coreset [2] selects query samples by maximizing sample diversity in the **feature space**. However, as shown in `§4.2`, Coreset fails to achieve reasonable performance under ActiveKD; for example, it even **underperforms the Random baseline** under ActiveKD (Zero-Shot) in `Tab. 1`.
>
>     - This empirical failure suggests that **the choice of diversity measure is critical**, and we provide detailed justification throughout the paper for redefining diversity in a way that aligns with the behavior of VLM teachers as we discussed in response to **[Q1-2]**.
>
>     - Specifically, we identify a unique and crucial challenge in the proposed ActiveKD framework: **structured prediction bias**. Rather than attempting to mitigate this bias, we treat it as an **inductive bias** of VLM teachers. To leverage this property, we shift the selection criterion from **feature space to probability space**, and propose **$\texttt{PCoreSet}$**.
>
>     - Empirically, $\texttt{PCoreSet}$ not only outperforms the Random baseline (unlike Coreset) but also achieves the **best performance in 64 out of 73 settings (87.7%)** under the **same ActiveKD** framework and **experimental conditions**.
>
> - We believe the above discussion clarifies that, although $\texttt{PCoreSet}$ is indeed simple, its shift from feature space to probability space addresses a **non-trivial challenge** and is **not fairly expected** when compared to classical Coreset approaches.
>
> ---
>
> > **[Q3-2]** The results match what one would anticipate when injecting a much stronger teacher into an AL loop.
>
> - We sincerely appreciate the reviewer for raising this thoughtful concern. However, we respectfully note that **injecting a “much stronger” teacher into an AL loop is non-trivial** in our problem setting, for several reasons:
>
>     - Our work focuses on **AL scenarios**, where only **a small number of labels** can be acquired per AL round. Under such constraints, it is not only the student but also the **teacher** that lacks sufficient task-specific labeled supervision. As a result, the teacher **cannot readily become "much stronger"** within the AL setting.
>
>     - Although VLMs demonstrate strong **zero-shot and few-shot** capabilities, their performance is often **inferior to small models trained or distilled on even modest amounts of task-specific labeled data**. This gap is particularly pronounced in structured or domain-specific tasks.
>
>     - For example, in `Fig. 1 (right)`, the **ResNet-18 student** trained with **ActiveKD + $\texttt{PCoreSet}$** already **outperforms the CLIP ResNet-50 zero-shot teacher** averaged over 10 datasets **after the first AL round**, and surpasses the **few-shot teacher** after the **second round**. This shows that VLM teachers are **not inherently strong in AL environments**, and can be overtaken rapidly by an actively trained student.
>
>     - Therefore, we believe it is important to design AL selection methods that effectively utilize **imperfect VLM teachers** under the proposed ActiveKD framework.
>
> - In summary, while a much stronger teacher would indeed improve performance in principle, we respectfully argue that such an assumption **does not align with realistic AL scenarios**, where strong task-specific teachers are typically unavailable.
>
> - We have added a discussion on the **limitations and imperfections of VLM teachers**, and how students trained under ActiveKD can quickly surpass them, in `L63–65` of `Fig. 1`.
>
>
> ---

---

> ### Author Response · Authors · 2025-11-21
> **Response to Reviewer YbDT (4/4)**
>
> ---
>
> >**[Q4]** Can you provide computational analyses?
>
> - We sincerely appreciate the reviewer for raising this point. We kindly note that we have **already discussed** this in `L302–305`, where we compare the complexity of the proposed $\texttt{PCoreSet}$ with that of a strong and representative baseline that selects query samples based on feature space [2].
>
> - Specifically, the computational complexity of $\texttt{PCoreSet}$ is $\mathcal{O}(C \cdot M \cdot N)$, where $C$ is the number of classes, $M = |\mathcal{D}^{(u)}|$ is the number of unlabeled samples, and $N = |\mathcal{D}^{(l)}|$ is the number of labeled samples accumulated up to the current AL round.
>
> - In contrast, the computational complexity of Coreset is $\mathcal{O}(H \cdot M \cdot N)$, where $H$ denotes the dimensionality of the feature space. Therefore, in common scenarios where $C \ll H$, $\texttt{PCoreSet}$ is **more computationally efficient**.
>
> - We have included the discussion on the computational complexity of uncertainty-scoring baseline, i.e., Entropy in `L303-304` in `§3.1`.
>
> ---
>
> ### References
>
> [1] Bang, Jihwan, Sumyeong Ahn, and Jae-Gil Lee. "Active prompt learning in vision language models." CVPR. 2024.
>
> [2] Sener, Ozan, and Silvio Savarese. "Active learning for convolutional neural networks: A core-set approach." ICLR. 2018.
>
> ---

---

> > ### Author Response · Authors · 2025-11-25
> > **Gentle Reminder**
> >
> > Dear Reviewer YbDT,
> >
> > We sincerely appreciate your time and consideration.
> > We respectfully believe that our response has thoroughly addressed the concerns raised.
> > If you have any remaining concerns or questions, please feel free to contact us and we would be happy to discuss and clarify them.
> >
> > Best,
> >
> > The Authors

---

### Official Review · Reviewer_NApS · 2025-10-31

**Soundness:** 2
**Presentation:** 2
**Contribution:** 2
**Rating:** 2
**Confidence:** 3

**Summary:**

This paper proposes ActiveKD, a novel active-learning framework that distills knowledge from large vision–language models (VLMs) into compact task-specific networks. The key technical contribution is PCoreSet, a selection criterion that queries unlabeled samples lying in under-represented regions of the teacher’s probability simplex rather than in feature space. Extensive experiments on 11 datasets, 5 students and 3 teachers show consistent gains (+29 % on ImageNet over baselines) and top rank in 64/73 settings.

**Strengths:**

- The work is the first to tightly integrate knowledge distillation with pool-based active learning, addressing a realistic scenario where labeled data are extremely scarce.

- The insight that VLM predictions exhibit structured bias transferable to students is both novel and well-supported theoretically (bias-propagation proposition) and empirically.

- PCoreSet is simple, fast (O(CMN)) and complementary to existing acquisition functions;

**Weaknesses:**

- The motivation is not very clear. The initial motivation of this paper is more like a combination of two existing modules, more than a well-defined problem. For AL part, one powerful model like teacher model (e.g. VLM) is what we need so why we conduct KD? For KD part, ActiveKD seems more reasonable and may benefit efficient KD while the paper is more prone to AL settings.

- The problem settings are highly similar to semi-supervised learning while the SOTA SSL methods are not compared in the experimental parts.

**Questions:**

Please refer to the weakness.

---

> ### Author Response · Authors · 2025-11-21
> **Response to Reviewer NApS (1/4)**
>
> We sincerly appreciate you for your time and constructive comments which improve our paper. We respectfully address your concerns as following:
>
> ---
>
> > **[Q1-1]** The motivation is not very clear. The initial motivation of this paper is more like a combination of two existing modules, rather than a well-defined problem.
>
> - We respectfully agree that our framework combines **two existing components**, i.e., active learning (AL) and knowledge distillation (KD). However, we believe the motivation is **non-trivial and well-defined** for the following reasons:
>
>   - As noted in `L15-18` and `L53`, AL and KD are **inherently difficult to combine** due to fundamental differences in their underlying assumptions. KD typically requires **abundant labeled data** and a **well-trained, task-specific teacher**, whereas AL assumes **data-scarce scenarios** with extremely limited or no initial labels, where such **task-specific teachers are unavailable**.
>
>   - As noted in `L65–78`, the emergence of vision–language models (VLMs) has recently enabled effective **zero- and few-shot knowledge distillation (KD)** across tasks. Motivated by this success, we introduce **ActiveKD**, which integrates active learning (AL) with KD by leveraging VLM teachers. To the best of our knowledge, this combination has **not been explored in prior work** (`L151–156`).
>
>   - Prior work has reported **class imbalance** in zero- and few-shot predictions of VLMs [1]. As noted in `L244-252`, we further observe that **this imbalance manifests as a structured form of prediction bias** in the probability simplex, forming distinct clusters (`Fig. 4b`), which can directly impact active sample selection.
>
>   - We formalize this phenomenon as **structured prediction bias** (`Definition 1`) and theoretically show how this bias can **propagate through KD** from teacher VLMs to student models (`Proposition 1`). This analysis is further **supported empirically** in `Fig. 4c`.
>
>   - Instead of treating this bias as a drawback, we leverage it as an **inductive bias** to improve AL sample selection. Based on this insight, we propose **$\texttt{PCoreSet}$** to **exploit this structured prediction bias** for more effective querying and model training.
>
> - In summary, ActiveKD is motivated by investigating the previously **unexplored combination of AL and KD**. We **identify and formalize a new challenge** under ActiveKD, namely **structured prediction bias** in VLM teachers and its propagation to students. We **propose a simple yet effective solution**, $\texttt{PCoreSet}$, that leverages this as an inductive bias for more effective sample selection.
>
> - We believe that the above discussion clarifies our motivation.
>
> ---

---

> ### Author Response · Authors · 2025-11-21
> **Response to Reviewer NApS (2/4)**
>
> ---
>
> >**[Q1-2]** For AL part, one powerful model like teacher model (e.g. VLM) is what we need so why we conduct KD?
>
> - We sincerely appreciate the reviewer for raising this point. While one may be satisfied with using a strong VLM teacher directly, we respectfully argue that there are many practical scenarios where training a **task-specific** (thus more accurate) or **compact** (thus more efficient) model is desirable:
>
>     - Our work focuses on **active learning (AL)** scenarios where we can obtain only **limited labels per AL round**. In such settings, a student model can eventually **outperform the teacher VLM** by combining the small amount of ground-truth supervision with knowledge distillation (KD).
>
>     - Although VLMs demonstrate impressive zero/few-shot ability, their performance is still **inferior to models trained and distilled on task-specific labeled data**, even when the amount of labeled data is small.
>
>     - For example, in `Fig. 1 (right)`, the **ResNet-18 student** trained with the proposed **ActiveKD + $\texttt{PCoreSet}$** already **surpasses the CLIP ResNet-50 zero-shot teacher** on the average of 10 datasets after the **first AL round**, and also surpasses the **few-shot teacher** after the **second round**.
>
> - In summary, **KD is necessary** because our goal is **not simply to reuse VLMs**, but to **produce a task-specific and compact model** for practical deployment under limited labeling budgets, i.e., in realistic AL scenarios.
>
> - We have included the discussion on the imperfections of VLM teachers which can easily surpassed by students trained under ActiveKD in `L63-65` of `Fig. 1`.
>
> ---
>
> >**[Q1-3]** For KD part, ActiveKD seems more reasonable and may benefit efficient KD while the paper is more prone to AL settings.
>
> - We sincerely agree that ActiveKD could potentially benefit from more efficient knowledge distillation (KD) methods, as the ActiveKD framework is **orthogonal** to the underlying KD method. For example, the logit distillation objective [2] used in `Eq. 3` **can be replaced with any recent KD method** (e.g., feature distillation [3]).
>
> - However, the goal of this paper is **not to propose a new KD technique**, but rather to introduce ActiveKD framework **for active learning (AL) settings**, and evaluate its effectiveness across diverse AL scenarios. Therefore, investigating more efficient KD methods under ActiveKD is **beyond the scope** of this work:
>
>     - We focus on a direction specific to **active selection under ActiveKD**: designing a algorithm to **select queries** at each AL round. Under ActiveKD, we observe **clustered geometric structure in the probability space** in VLM teachers (see `Fig. 4b`).
>
>     - We formally define this phenomenon as **structured prediction bias** (`Definition 1`) and theoretically show how it can **propagate through KD** from teacher VLMs to student models (`Proposition 1`), with **empirical evidence** provided in `Fig. 4c`.
>
>     - Rather than treating this bias as a drawback, we leverage it as an **inductive bias** beneficial for active selection. Based on this insight, we propose **$\texttt{PCoreSet}$**, an AL selection strategy that **exploits structured prediction bias** for more effective querying and model training.
>
> - In summary, while ActiveKD may inspire **future work on more efficient or alternative KD methods**, since it is largely **orthogonal** to the underlying KD formulation, our contribution is to propose an **AL selection approach** specifically tailored to ActiveKD by leveraging structured prediction bias. This is supported by the strong empirical results shown in `Figs. 5 and 6`.
>
> - We have clarified that exploring more effective KD methods under ActiveKD is future work in `L484–485` of `§5`.
>
> ---

---

> ### Author Response · Authors · 2025-11-21
> **Response to Reviewer NApS (3/4)**
>
> ---
>
> > **[Q2-1]** The problem settings are highly similar to semi-supervised learning while SOTA SSL methods are not compared in the experiments.
>
> - We respectfully agree that our setting is **closely related** to semi-supervised learning (SSL). At each active learning (AL) round, once a batch of labels is acquired through the AL selection algorithm, the classifier is indeed trained in an **SSL manner** on the labeled set $\mathcal{D}^{(l)}$ and the unlabeled pool $\mathcal{D}^{(u)}$.
>
> - However, we believe that a direct comparison with state-of-the-art (SoTA) SSL methods is **out of scope** and also **orthogonal** for the following reasons:
>
>   - Our work focuses on **AL**, where the main research question is how to **select the most informative samples** under a limited annotation budget. SSL methods, in contrast, assume that labeled data is already given and do not address query efficiency or annotation constraints, which is the primary objective of our work.
>
>   - Our setting explicitly assumes access to **VLM teacher models** that provide zero-/few-shot knowledge distillation signals (`Eq. 3`) for training. This teacher-based supervision is a **core assumption** of the ActiveKD framework. Many state-of-the-art (SoTA) SSL methods do not assume access to a teacher model, and are therefore **not directly comparable**.
>
>   - More crucially, SoTA SSL methods that assumes **use teacher models** (e.g., [4]), while conceptually related, are **orthogonal** to ours. This is because SSL methods improve learning **after labels have already been selected**, whereas our contribution lies in **how to select samples** in the first place via $\texttt{PCoreSet}$ by leveraging **structured prediction bias** (see `Fig. 4b` and `Definition 1`). In other words, these SoTA SSL methods address (pseudo-)**label quality**, while our method addresses **label acquisition**.
>
> - For these reasons, while SSL techniques are **orthogonal** and could **potentially be integrated** into ActiveKD in future work, we consider benchmarking against SoTA SSL methods **beyond the scope of this paper**, whose focus is the ActiveKD framework and the proposed selection method $\texttt{PCoreSet}$.
>
> - We have clarified the above discussion in `L484–485` of `§5`.
>
> ---
>
> > **[Q2-2]** The problem settings are highly similar to semi-supervised learning while SOTA SSL methods are not compared in the experiments.
>
> - Instead, we conduct an additional ablation study to investivate **the SSL effect of ActiveKD** by **excluding unlabeled data**, i.e., computing KD loss in `Eq. 3` on **labeled samples alone**.
> - Specifically, `Eq. 3` is now $\mathcal{L}\_\text{KD}=\frac{1}{N}\sum_n D\_{KL}[f(x\_n^{(l)})||f\_r(x\_n^{(l)})]$. We use the same experimental setup as in `Fig. 6` with zer-shot teachers using 8 of 10 datasets.
> - `Table NApS-1` reports the performance of all selection algorithms under 1) No Distill, 2) **ActiveKD (w/o unlabeled)**, and 3) ActiveKD, where 2) ActiveKD (w/o unlabeled) is the ablation target by **excluding unlabeled data from KD**.
> - We observe that ActiveKD (w/o unlabeled) **significantly improves No Distill** across all selection methods, however, the improvement is **much less than ActiveKD** (e.g., **+5.36 vs. +8.49** at the 16-th AL round with $\texttt{PCoreSet}$).
> - More crucially, even under ActiveKD (w/o unlabeled), $\texttt{PCoreSet}$ **achieves the best performance in most cases**, demonstrating its effectiveness under this setting, i.e., semi-supervised learning via knowledge distillation on **labeled samples alone**.
> - In summary, the above results show that the **SSL effect of ActiveKD is significant**, and we believe it could indeed be **further improved by integrating it** with more effective SSL methods.
> - We have included the above discussion in `Appendix D.5`.
>
> ---
>
> ### References
>
> [1] Bang, Jihwan, Sumyeong Ahn, and Jae-Gil Lee. "Active prompt learning in vision language models." CVPR. 2024.
>
> [2] Hinton, Geoffrey, Oriol Vinyals, and Jeff Dean. "Distilling the knowledge in a neural network." arXiv 2015.
>
> [3] Yang, Chuanguang, et al. "Clip-kd: An empirical study of clip model distillation." CVPR 2024.
>
> [4] Wang, Xudong, et al. "Debiased learning from naturally imbalanced pseudo-labels." CVPR. 2022.
>
> ---

---

> ### Author Response · Authors · 2025-11-21
> **Response to Reviewer NApS (4/4)**
>
> **[Table NApS-1]** Results on 8 datasets (average) when **excluding unlabeled data from KD** (denoted as *w/o unlabeled*). The number in paranthesis denotes the improvement over No Distill.
>
> |Method|KD|2|3|4|5|6|7|8|9|10|11|12|13|14|15|16|
> |:-|:-|:-:|:-:|:-:|:-:|:-:|:-:|:-:|:-:|:-:|:-:|:-:|:-:|:-:|:-:|:-:|
> |Random|No Distill|36.70|42.64|47.65|50.93|53.77|56.69|59.24|60.15|62.46|64.06|65.62|66.49|67.60|68.90|69.51|
> |Random|ActiveKD (w/o unlabeled)|43.33 (**+6.63**)|49.58 (**+6.94**)|55.20 (**+7.55**)|58.79 (**+7.86**)|61.87 (**+8.10**)|63.90 (**+7.21**)|66.17 (**+6.93**)|67.72 (**+7.57**)|69.30 (**+6.84**)|70.68 (**+6.62**)|71.86 (**+6.24**)|72.84 (**+6.35**)|73.88 (**+6.28**)|74.15 (**+5.25**)|75.00 (**+5.49**)|
> |Random|ActiveKD|64.54 (**+27.84**)|66.82 (**+24.18**)|69.01 (**+21.36**)|70.77 (**+19.84**)|71.97 (**+18.20**)|73.47 (**+16.78**)|74.11 (**+14.87**)|74.79 (**+14.64**)|75.81 (**+13.35**)|76.15 (**+12.09**)|76.89 (**+11.27**)|77.38 (**+10.89**)|77.57 (**+9.97**)|78.02 (**+9.12**)|78.67 (**+9.16**)|
> |Entropy|No Distill|34.83|40.60|46.39|50.96|54.77|57.18|60.14|61.43|63.07|64.54|66.32|67.56|69.21|70.30|71.29|
> |Entropy|ActiveKD (w/o unlabeled)|41.09 (**+6.26**)|48.59 (**+7.99**)|55.80 (**+9.41**)|57.51 (**+6.55**)|61.18 (**+6.41**)|65.13 (**+7.95**)|66.90 (**+6.76**)|68.71 (**+7.28**)|69.76 (**+6.69**)|70.98 (**+6.44**)|72.80 (**+6.48**)|74.11 (**+6.55**)|74.77 (**+5.56**)|76.30 (**+5.99**)|76.67 (**+5.38**)|
> |Entropy|ActiveKD|63.82 (**+28.99**)|66.52 (**+25.92**)|67.94 (**+21.55**)|69.82 (**+18.86**)|71.09 (**+16.32**)|72.42 (**+15.24**)|73.41 (**+13.27**)|74.33 (**+12.90**)|75.13 (**+12.06**)|75.72 (**+11.18**)|76.48 (**+10.16**)|76.94 (**+9.38**)|77.46 (**+8.25**)|78.04 (**+7.74**)|78.44 (**+7.15**)|
> |Coreset|No Distill|37.28|44.90|50.44|53.24|55.69|57.93|60.50|62.96|63.97|66.40|67.30|68.29|69.56|70.62|72.05|
> |Coreset|ActiveKD (w/o unlabeled)|44.21 (**+6.93**)|51.05 (**+6.15**)|55.31 (**+4.87**)|59.84 (**+6.60**)|62.98 (**+7.29**)|65.14 (**+7.21**)|67.55 (**+7.05**)|69.25 (**+6.29**)|71.14 (**+7.17**)|71.11 (**+4.71**)|72.90 (**+5.60**)|74.15 (**+5.86**)|74.97 (**+5.41**)|75.58 (**+4.96**)|76.67 (**+4.62**)|
> |Coreset|ActiveKD|65.70 (**+28.42**)|67.68 (**+22.78**)|69.82 (**+19.38**)|71.47 (**+18.23**)|72.40 (**+16.71**)|73.46 (**+15.53**)|74.76 (**+14.26**)|75.45 (**+12.49**)|76.27 (**+12.30**)|76.49 (**+10.09**)|76.99 (**+9.69**)|77.57 (**+9.28**)|78.33 (**+8.77**)|78.53 (**+7.91**)|78.99 (**+6.94**)|
> |Badge|No Distill|36.61|42.58|48.73|52.44|54.67|57.70|59.37|61.28|63.47|64.07|66.21|66.43|68.05|69.06|69.60|
> |Badge|ActiveKD (w/o unlabeled)|43.43 (**+6.82**)|50.42 (**+7.84**)|56.56 (**+7.83**)|59.17 (**+6.73**)|61.60 (**+6.93**)|63.91 (**+6.21**)|66.44 (**+7.07**)|67.32 (**+6.04**)|68.94 (**+5.47**)|70.54 (**+6.47**)|71.48 (**+5.27**)|72.16 (**+5.73**)|73.09 (**+5.04**)|74.47 (**+5.41**)|74.91 (**+5.31**)|
> |Badge|ActiveKD|65.29 (**+28.68**)|68.05 (**+25.47**)|70.31 (**+21.58**)|72.12 (**+19.68**)|73.27 (**+18.60**)|74.29 (**+16.59**)|75.11 (**+15.74**)|75.80 (**+14.52**)|76.44 (**+12.97**)|76.88 (**+12.81**)|77.44 (**+11.23**)|77.84 (**+11.41**)|78.08 (**+10.03**)|78.71 (**+9.65**)|78.92 (**+9.32**)|
> |ClassBalanced|No Distill|37.07|44.17|49.17|52.68|54.70|58.37|59.88|61.71|63.04|64.77|65.80|67.33|68.34|69.37|70.02|
> |ClassBalanced|ActiveKD (w/o unlabeled)|43.40 (**+6.33**)|51.31 (**+7.14**)|57.32 (**+8.15**)|59.93 (**+7.25**)|63.86 (**+9.16**)|65.34 (**+6.97**)|67.77 (**+7.89**)|68.62 (**+6.91**)|70.03 (**+7.00**)|72.07 (**+7.30**)|72.53 (**+6.73**)|74.15 (**+6.82**)|74.47 (**+6.13**)|75.18 (**+5.81**)|76.10 (**+6.08**)|
> |ClassBalanced|ActiveKD|65.55 (**+28.48**)|68.28 (**+24.11**)|70.77 (**+21.60**)|72.18 (**+19.50**)|73.34 (**+18.64**)|74.11 (**+15.74**)|75.30 (**+15.42**)|75.78 (**+14.07**)|76.33 (**+13.29**)|76.93 (**+12.16**)|77.47 (**+11.67**)|77.81 (**+10.48**)|78.11 (**+9.77**)|78.59 (**+9.22**)|78.95 (**+8.93**)|
> |$\texttt{PCoreSet}$|No Distill|35.97|43.17|48.45|51.74|55.24|58.81|60.72|62.67|64.31|65.61|67.70|69.02|70.10|71.21|71.95|
> |$\texttt{PCoreSet}$|ActiveKD (w/o unlabeled)|43.00 (**+7.03**)|50.06 (**+6.89**)|56.36 (**+7.91**)|60.23 (**+8.49**)|63.99 (**+8.75**)|66.08 (**+7.27**)|68.00 (**+7.28**)|69.65 (**+6.98**)|71.44 (**+7.13**)|72.94 (**+7.33**)|73.84 (**+6.14**)|74.02 (**+5.00**)|76.00 (**+5.90**)|76.72 (**+5.51**)|77.31 (**+5.36**)|
> |$\texttt{PCoreSet}$|ActiveKD|65.67 (**+29.70**)|68.61 (**+25.44**)|71.00 (**+22.55**)|72.81 (**+21.07**)|74.05 (**+18.81**)|75.31 (**+16.50**)|76.24 (**+15.52**)|77.15 (**+14.48**)|77.63 (**+13.32**)|78.17 (**+12.56**)|78.76 (**+11.06**)|79.15 (**+10.13**)|79.76 (**+9.66**)|80.08 (**+8.87**)|80.44 (**+8.49**)|

---

> > ### Author Response · Authors · 2025-11-25
> > **Gentle Reminder**
> >
> > Dear Reviewer NApS,
> >
> > We sincerely appreciate your time and consideration.
> > We respectfully believe that our response has thoroughly addressed the concerns raised.
> > If you have any remaining concerns or questions, please feel free to contact us and we would be happy to discuss and clarify them.
> >
> > Best,
> >
> > The Authors

---

### Official Review · Reviewer_hCWn · 2025-11-01

**Soundness:** 3
**Presentation:** 3
**Contribution:** 3
**Rating:** 6
**Confidence:** 2

**Summary:**

This paper introduces ActiveKD, a new framework that integrates Active Learning (AL) with Knowledge Distillation (KD) to train compact, task-specific models in data-scarce environments. It addresses a key challenge: AL typically lacks the labeled data required to train a powerful "teacher" model for KD. ActiveKD solves this by using a large Vision-Language Model (VLM) as a zero-shot or few-shot teacher. In each AL round, the "student" model is trained using both the few human-labeled samples and soft labels (knowledge) distilled from the VLM teacher on the entire unlabeled data pool.

The authors observe that VLM teachers exhibit a "structured prediction bias," meaning their output probabilities form distinct clusters. They propose to leverage this as a useful inductive bias. To do this, they introduce Probabilistic CoreSet (PCoreSet), a novel AL selection strategy. Unlike traditional coreset methods that select samples for diversity in the feature space, PCoreSet selects samples that are diverse in the probability space, effectively covering the teacher's output clusters.

Extensive experiments on 11 datasets show that the ActiveKD framework itself provides a large performance boost to all AL methods. Within this new framework, PCoreSet is the most effective selection strategy, outperforming baselines in 64 out of 73 tested settings.

**Strengths:**

1. The paper introduces ActiveKD, a novel framework that successfully integrates Active Learning (AL) with Knowledge Distillation (KD). This is significant because it solves a key problem: AL operates in data-scarce settings where powerful task-specific "teacher" models are normally unavailable, but ActiveKD overcomes this by leveraging the zero- and few-shot capabilities of large Vision-Language Models (VLMs) as teachers.
2. The paper identifies that VLM teachers exhibit a "structured prediction bias," where their outputs cluster in the probability space, and reframes this not as a limitation but as a useful inductive bias. Based on this insight, it proposes the Probabilistic CoreSet (PCoreSet), a novel selection strategy that maximizes coverage in the probability space rather than the feature space, which is a key differentiator from standard coreset methods.

**Weaknesses:**

1. The paper reframes the VLM's "structured prediction bias" as a positive inductive bias to be exploited. However, it fails to investigate the negative case: what if this "structured bias" is just the VLM being confidently and systematically wrong about a whole cluster of data? The proposed PCoreSet method, by design, would actively sample from this erroneous cluster, potentially leading the student to efficiently learn the teacher's mistakes.
2.  The ActiveKD framework (Algorithm 1) applies knowledge distillation to both the small labeled set $\mathcal{D}^{(l)}$ and the large unlabeled pool $\mathcal{D}^{(u)}$. The large performance gains attributed to ActiveKD (e.g., +29.07% on ImageNet) are never ablated to separate the effect of distillation on the unlabeled pool (a form of semi-supervised learning) from the effect of distillation on the actively selected labeled set. It is unclear how much of the gain is simply from using the VLM teacher in a semi-supervised fashion versus the active learning component itself.

**Questions:**

1. You reframe the VLM's "structured prediction bias" as a positive inductive bias to be exploited. However, what if this bias simply represents a "blind spot" where the VLM teacher is confidently and systematically wrong about a whole cluster of data? Since PCoreSet is designed to find and sample from these clusters, how does your method prevent the student model from efficiently learning the teacher's errors (i.e., negative knowledge transfer)?
2. The ActiveKD framework applies knowledge distillation to both the small labeled set $\mathcal{D}^{(l)}$ and the entire unlabeled pool $\mathcal{D}^{(u)}$. Your results show that this framework provides a massive performance boost (e.g., +29.07% on ImageNet) over the "No Distill" baseline, even for Random selection. This suggests a large gain comes from the semi-supervised learning (SSL) component on $\mathcal{D}^{(u)}$. How much of the performance improvement is actually attributable to the active selection strategy versus this powerful SSL effect? Could you provide an ablation that applies KD only to the labeled samples?

---

> ### Author Response · Authors · 2025-11-21
> **Response to Reviewer hCWn (1/4)**
>
> We sincerly appreciate you for your time and constructive comments which improve our paper. We respectfully address your concerns as following:
>
> ---
>
> >**[Q1]** The paper reframes the VLM's "structured prediction bias" as a positive inductive bias to be exploited. However, it fails to investigate the negative case: what if this "structured bias" is just the VLM being confidently and systematically wrong about a whole cluster of data? The proposed PCoreSet method, by design, would actively sample from this erroneous cluster, potentially leading the student to efficiently learn the teacher's mistakes.
>
> - We sincerely appreciate the reviewer for the **in-depth understanding** of our method and agree that there is a potential risk of selecting samples from erroneous clusters. Since we assume zero-/few-shot VLMs as teacher models, it is expected that their predictions may sometimes be **overconfident or erroneous**.
>
> - We respectfully argue, however, that this risk can be mitigated by the use of **ground-truth (GT) labels** for the queried samples.
>
> - Suppose that $\texttt{PCoreSet}$ selects a query sample on which the VLM teachers are overconfident or systematically incorrect. If the classifier were trained **only** with the knowledge distillation (KD) objective in `Eq. 3`, it could indeed **misclassify** that sample (and potentially other visually similar samples), leading to a systematic error.
>
> - Fortunately, in the AL scenarios considered in our work, we assume access to **GT labels for the queried samples**, though under a limited budget (e.g., the 1-shot setting, $Q = C$). At each AL round, as noted in `L4` of `Alg. 1`, we train the classifier on both the **accumulated labeled set** $\mathcal{D}^{(l)}$ and the unlabeled pool $\mathcal{D}^{(u)}$ using both the **supervised loss** (`Eq. 2`) and the KD loss (`Eq. 3`).
>
> - Therefore, even when the predictions from teacher VLMs (i.e., the structured bias of VLMs) are **overconfident or systematically wrong**, the student model can **correct such errors through GT supervision**.
>
> - We believe this property contributes to the strong empirical effectiveness of $\texttt{PCoreSet}$ under ActiveKD, as demonstrated in `Figs. 5 and 6`.
>
> - We have incorparted the above discussion into `L298–300` in `§3.3`.
>
> ---

---

> ### Author Response · Authors · 2025-11-21
> **Response to Reviewer hCWn (2/4)**
>
> ---
>
> >**[Q2]** The ActiveKD framework (Algorithm 1) applies knowledge distillation to both the small labeled set $\mathcal{D}^{(l)}$ and the large unlabeled pool $\mathcal{D}^{(u)}$. The large performance gains attributed to ActiveKD (e.g., +29.07% on ImageNet) are never ablated to separate the effect of distillation on the unlabeled pool (a form of semi-supervised learning) from the effect of distillation on the actively selected labeled set. It is unclear how much of the gain is simply from using the VLM teacher in a semi-supervised fashion versus the active learning component itself.
>
>
> - We sincerely appreciate the reviewer for raising this point and apologize for the unclear presentation. However, we respectfully argue that we have **already discussed** how the **different components contribute to the overall improvement** in `§4`.
>
> - To clarify, we would first like to recap the distinctions between the components of our proposed approach as follows:
>
>   1) **ActiveKD** (`§3.2`): a *pool-based* active learning (AL) framework that leverages VLM teachers to train a classifier on both labeled data $\mathcal{D}^{(l)}$ and the unlabeled pool $\mathcal{D}^{(u)}$ in each AL round. Specifically, the classifier is trained with supervised loss (`Eq. 2`) on $\mathcal{D}^{(l)}$ and distillation loss (`Eq. 3`) on both $\mathcal{D}^{(l)}$ and $\mathcal{D}^{(u)}$.
>
>   2) **$\texttt{PCoreSet}$** (`§3.3`): an AL *selection strategy* designed for the ActiveKD framework. Any selection method can be used under ActiveKD (e.g., Random, Entropy, Coreset, Badge, and ClassBalanced). We propose $\texttt{PCoreSet}$ to **maximally leverage the structured bias** of VLM teachers for sample selection.
>
> - In `Tab. 1`, we demonstrate that **ActiveKD** improves classification performance compared to **No Distill** (i.e., the conventional AL framework) across **all** AL selection algorithms, including the **Random** baseline. This shows the **effectiveness of ActiveKD** in leveraging VLM teachers together with the unlabeled pool $\mathcal{D}^{(u)}$.
>
> - After establishing the effectiveness of ActiveKD, we then compare different AL selection strategies **under the same ActiveKD framework** in `Fig. 5` and perform an in-depth comparison across 10 datasets in `Fig. 6`. In `Fig. 5`, we observe strong empirical evidence that **$\texttt{PCoreSet}$ consistently outperforms other selection methods**, achieving the best performance in **64 out of 73 settings (87.7%)**.
>
> - In summary, we propose two contributions: (i) **ActiveKD**, and (ii) **$\texttt{PCoreSet}$**, and the above evaluation design **isolates the effects** appropriately: `Tab. 1` shows the **benefit of ActiveKD** across all selection methods, while `Figs. 5 and 6` demonstrate the **additional benefit** of $\texttt{PCoreSet}$ within the **same ActiveKD framework**.
>
> - We have clarified the fair evaluation of across different AL methods under ActiveKD in `L350-351` of `§4.1`.
>
> ---
>
> >**[Q3]** You reframe the VLM's "structured prediction bias" as a positive inductive bias to be exploited. However, what if this bias simply represents a "blind spot" where the VLM teacher is confidently and systematically wrong about a whole cluster of data? Since PCoreSet is designed to find and sample from these clusters, how does your method prevent the student model from efficiently learning the teacher's errors (i.e., negative knowledge transfer)?
>
> - Again, we sincerely appreciate the reviewer for raising this important concern of our method.
> - In summary, we believe that the use of **ground-truth (GT) labels** for queried samples in AL settings prevents the student model from learning **only from the overconfidence and systematic errors** of VLM teachers. This property contributes to the strong empirical effectiveness of $\texttt{PCoreSet}$ demonstrated in `Figs. 5 and 6`.
> - Please refer to our response to **[Q1]** for further details.
>
> ---

---

> ### Author Response · Authors · 2025-11-21
> **Response to Reviewer hCWn (3/4)**
>
> ---
>
> >**[Q4-1]** The ActiveKD framework applies knowledge distillation to both the small labeled set $\mathcal{D}^{(l)}$ and the entire unlabeled pool $\mathcal{D}^{(u)}$. Your results show that this framework provides a massive performance boost (e.g., +29.07% on ImageNet) over the "No Distill" baseline, even for Random selection. This suggests a large gain comes from the semi-supervised learning (SSL) component on $\mathcal{D}^{(u)}$. How much of the performance improvement is actually attributable to the active selection strategy versus this powerful SSL effect?
>
> - We sincerely appreciate this important question about the contribution of different components. We would first like to clarify the distinction between our two proposed components: (1) **ActiveKD**, which leverages VLM teachers, and (2) **$\texttt{PCoreSet}$**, which selects query points to maximally exploit the structured bias of VLM teachers. Please refer to our response to **[Q2]** for additional details.
>
> - To demonstrate the effectiveness of these two components, we design two complementary experimental evaluations:
>
>   1) **`Tab. 1`**: We show the effectiveness of the proposed **ActiveKD** framework over the conventional AL setup (denoted as *"No Distill"*). We observe that ActiveKD yields **substantial performance improvements** across all AL selection methods, including the Random baseline. This provides empirical support that ActiveKD is beneficial **regardless of the choice of AL selection algorithm**.
>
>   2) **`Figs. 5 and 6`**: **Under the same ActiveKD framework**, we **compare different AL selection algorithms** and observe that the proposed **$\texttt{PCoreSet}$** achieves the **best performance in 64 out of 73 settings (87.7%)**. This evaluation **isolates the SSL effect of ActiveKD** and clearly demonstrates the performance **advantage** of $\texttt{PCoreSet}$ over other **AL selection baselines**.
>
> - Therefore, in response to the question *"How much of the performance improvement is actually attributable to the active selection strategy versus this powerful SSL effect?"*, we note that our evaluation in **`Figs. 5 and 6`** already **isolates the SSL effect** of ActiveKD. The results show that **$\texttt{PCoreSet}$ provides additional and consistent gains beyond the SSL benefit of ActiveKD itself**, compared with other AL selection baselines.
>
> - We have clarified in the revision that the experiments in `Figs. 5 and 6` already isolate the SSL effect by using the same ActiveKD framework, as described in `L401–402` of `§4.2`.
>
> ---
>
> >**[Q4-2]** Could you provide an ablation that applies KD only to the labeled samples?
>
> - We sincerely appreciate the reviewer for suggesting important ablation experiment. We first kindly note our main focus is on practical AL scenarios where **unlabeled data is naturally available** and can be leveraged during training.
> - As noted in `§3.1`, this is because pool-based AL assumes the existence of an unlabeled data pool ($\mathcal{D}^{(u)}$), from which data points are actively selected and queried for annotation.
> - However, we sincerely agree that the suggested ablation would provide valuable insights into the contribution of semi-supervised learning via knowledge distillation on **labeled samples alone**.
> - To investigate this, we conduct additional experiments by **excluding unlabeled data** for computation of KD loss in `Eq. 3`, i.e, $\mathcal{L}\_\text{KD}=\frac{1}{N}\sum_n D\_{KL}[f(x\_n^{(l)})||f\_r(x\_n^{(l)})]$. We use the same experimental setup as in `Fig. 6` with zer-shot teachers using 8 of 10 datasets.
> - `Table hCWn-1` reports the performance of all selection algorithms under 1) No Distill, 2) **ActiveKD (w/o unlabeled)**, and 3) ActiveKD, where 2) ActiveKD (w/o unlabeled) is the ablation target by **excluding unlabeled data from KD**.
> - We observe that ActiveKD (w/o unlabeled) **significantly improves No Distill** across all selection methods, however, the improvement is **much less than ActiveKD** (e.g., **+5.36 vs. +8.49** at the 16-th AL round with $\texttt{PCoreSet}$).
> - More crucially, even under ActiveKD (w/o unlabeled), $\texttt{PCoreSet}$ **achieves the best performance in most cases**, demonstrating its effectiveness under this setting, i.e., semi-supervised learning via knowledge distillation on **labeled samples alone**.
> - We have included the above discussion in `Appendix D.5`.
>
> ---

---

> ### Author Response · Authors · 2025-11-21
> **Response to Reviewer hCWn (4/4)**
>
> **[Table hCWn-1]** Results on 8 datasets (average) when **excluding unlabeled data from KD** (denoted as *w/o unlabeled*). The number in paranthesis denotes the improvement over No Distill.
>
> |Method|KD|2|3|4|5|6|7|8|9|10|11|12|13|14|15|16|
> |:-|:-|:-:|:-:|:-:|:-:|:-:|:-:|:-:|:-:|:-:|:-:|:-:|:-:|:-:|:-:|:-:|
> |Random|No Distill|36.70|42.64|47.65|50.93|53.77|56.69|59.24|60.15|62.46|64.06|65.62|66.49|67.60|68.90|69.51|
> |Random|ActiveKD (w/o unlabeled)|43.33 (**+6.63**)|49.58 (**+6.94**)|55.20 (**+7.55**)|58.79 (**+7.86**)|61.87 (**+8.10**)|63.90 (**+7.21**)|66.17 (**+6.93**)|67.72 (**+7.57**)|69.30 (**+6.84**)|70.68 (**+6.62**)|71.86 (**+6.24**)|72.84 (**+6.35**)|73.88 (**+6.28**)|74.15 (**+5.25**)|75.00 (**+5.49**)|
> |Random|ActiveKD|64.54 (**+27.84**)|66.82 (**+24.18**)|69.01 (**+21.36**)|70.77 (**+19.84**)|71.97 (**+18.20**)|73.47 (**+16.78**)|74.11 (**+14.87**)|74.79 (**+14.64**)|75.81 (**+13.35**)|76.15 (**+12.09**)|76.89 (**+11.27**)|77.38 (**+10.89**)|77.57 (**+9.97**)|78.02 (**+9.12**)|78.67 (**+9.16**)|
> |Entropy|No Distill|34.83|40.60|46.39|50.96|54.77|57.18|60.14|61.43|63.07|64.54|66.32|67.56|69.21|70.30|71.29|
> |Entropy|ActiveKD (w/o unlabeled)|41.09 (**+6.26**)|48.59 (**+7.99**)|55.80 (**+9.41**)|57.51 (**+6.55**)|61.18 (**+6.41**)|65.13 (**+7.95**)|66.90 (**+6.76**)|68.71 (**+7.28**)|69.76 (**+6.69**)|70.98 (**+6.44**)|72.80 (**+6.48**)|74.11 (**+6.55**)|74.77 (**+5.56**)|76.30 (**+5.99**)|76.67 (**+5.38**)|
> |Entropy|ActiveKD|63.82 (**+28.99**)|66.52 (**+25.92**)|67.94 (**+21.55**)|69.82 (**+18.86**)|71.09 (**+16.32**)|72.42 (**+15.24**)|73.41 (**+13.27**)|74.33 (**+12.90**)|75.13 (**+12.06**)|75.72 (**+11.18**)|76.48 (**+10.16**)|76.94 (**+9.38**)|77.46 (**+8.25**)|78.04 (**+7.74**)|78.44 (**+7.15**)|
> |Coreset|No Distill|37.28|44.90|50.44|53.24|55.69|57.93|60.50|62.96|63.97|66.40|67.30|68.29|69.56|70.62|72.05|
> |Coreset|ActiveKD (w/o unlabeled)|44.21 (**+6.93**)|51.05 (**+6.15**)|55.31 (**+4.87**)|59.84 (**+6.60**)|62.98 (**+7.29**)|65.14 (**+7.21**)|67.55 (**+7.05**)|69.25 (**+6.29**)|71.14 (**+7.17**)|71.11 (**+4.71**)|72.90 (**+5.60**)|74.15 (**+5.86**)|74.97 (**+5.41**)|75.58 (**+4.96**)|76.67 (**+4.62**)|
> |Coreset|ActiveKD|65.70 (**+28.42**)|67.68 (**+22.78**)|69.82 (**+19.38**)|71.47 (**+18.23**)|72.40 (**+16.71**)|73.46 (**+15.53**)|74.76 (**+14.26**)|75.45 (**+12.49**)|76.27 (**+12.30**)|76.49 (**+10.09**)|76.99 (**+9.69**)|77.57 (**+9.28**)|78.33 (**+8.77**)|78.53 (**+7.91**)|78.99 (**+6.94**)|
> |Badge|No Distill|36.61|42.58|48.73|52.44|54.67|57.70|59.37|61.28|63.47|64.07|66.21|66.43|68.05|69.06|69.60|
> |Badge|ActiveKD (w/o unlabeled)|43.43 (**+6.82**)|50.42 (**+7.84**)|56.56 (**+7.83**)|59.17 (**+6.73**)|61.60 (**+6.93**)|63.91 (**+6.21**)|66.44 (**+7.07**)|67.32 (**+6.04**)|68.94 (**+5.47**)|70.54 (**+6.47**)|71.48 (**+5.27**)|72.16 (**+5.73**)|73.09 (**+5.04**)|74.47 (**+5.41**)|74.91 (**+5.31**)|
> |Badge|ActiveKD|65.29 (**+28.68**)|68.05 (**+25.47**)|70.31 (**+21.58**)|72.12 (**+19.68**)|73.27 (**+18.60**)|74.29 (**+16.59**)|75.11 (**+15.74**)|75.80 (**+14.52**)|76.44 (**+12.97**)|76.88 (**+12.81**)|77.44 (**+11.23**)|77.84 (**+11.41**)|78.08 (**+10.03**)|78.71 (**+9.65**)|78.92 (**+9.32**)|
> |ClassBalanced|No Distill|37.07|44.17|49.17|52.68|54.70|58.37|59.88|61.71|63.04|64.77|65.80|67.33|68.34|69.37|70.02|
> |ClassBalanced|ActiveKD (w/o unlabeled)|43.40 (**+6.33**)|51.31 (**+7.14**)|57.32 (**+8.15**)|59.93 (**+7.25**)|63.86 (**+9.16**)|65.34 (**+6.97**)|67.77 (**+7.89**)|68.62 (**+6.91**)|70.03 (**+7.00**)|72.07 (**+7.30**)|72.53 (**+6.73**)|74.15 (**+6.82**)|74.47 (**+6.13**)|75.18 (**+5.81**)|76.10 (**+6.08**)|
> |ClassBalanced|ActiveKD|65.55 (**+28.48**)|68.28 (**+24.11**)|70.77 (**+21.60**)|72.18 (**+19.50**)|73.34 (**+18.64**)|74.11 (**+15.74**)|75.30 (**+15.42**)|75.78 (**+14.07**)|76.33 (**+13.29**)|76.93 (**+12.16**)|77.47 (**+11.67**)|77.81 (**+10.48**)|78.11 (**+9.77**)|78.59 (**+9.22**)|78.95 (**+8.93**)|
> |$\texttt{PCoreSet}$|No Distill|35.97|43.17|48.45|51.74|55.24|58.81|60.72|62.67|64.31|65.61|67.70|69.02|70.10|71.21|71.95|
> |$\texttt{PCoreSet}$|ActiveKD (w/o unlabeled)|43.00 (**+7.03**)|50.06 (**+6.89**)|56.36 (**+7.91**)|60.23 (**+8.49**)|63.99 (**+8.75**)|66.08 (**+7.27**)|68.00 (**+7.28**)|69.65 (**+6.98**)|71.44 (**+7.13**)|72.94 (**+7.33**)|73.84 (**+6.14**)|74.02 (**+5.00**)|76.00 (**+5.90**)|76.72 (**+5.51**)|77.31 (**+5.36**)|
> |$\texttt{PCoreSet}$|ActiveKD|65.67 (**+29.70**)|68.61 (**+25.44**)|71.00 (**+22.55**)|72.81 (**+21.07**)|74.05 (**+18.81**)|75.31 (**+16.50**)|76.24 (**+15.52**)|77.15 (**+14.48**)|77.63 (**+13.32**)|78.17 (**+12.56**)|78.76 (**+11.06**)|79.15 (**+10.13**)|79.76 (**+9.66**)|80.08 (**+8.87**)|80.44 (**+8.49**)|

---

> > ### Author Response · Authors · 2025-11-25
> > **Gentle Reminder**
> >
> > Dear Reviewer hCWn,
> >
> > We sincerely appreciate your time and consideration.
> > We respectfully believe that our response has thoroughly addressed the concerns raised.
> > If you have any remaining concerns or questions, please feel free to contact us and we would be happy to discuss and clarify them.
> >
> > Best,
> >
> > The Authors

---

### Official Review · Reviewer_mf9U · 2025-11-01

**Soundness:** 3
**Presentation:** 3
**Contribution:** 2
**Rating:** 4
**Confidence:** 5

**Summary:**

This work proposes a method to construct a set of unlabeled data for labeling, aimed at improving model performance incrementally. The approach combines active learning and knowledge distillation, leveraging probability-based diversity coverage to guide selection. Performance gains are demonstrated through experiments using VLM models on selected benchmark datasets.

**Strengths:**

1. The proposed approach is carefully developed and supported by both theoretical analysis and experimental results.
2. The method is presented clearly, with evidence that illustrates its performance gain over selected benchmarks.

**Weaknesses:**

1. The method requires additional training and repeated querying to construct the core set. In practical applications, this overhead may limit scalability and reduce its real-world applicability, weakening the overall contribution.
2. The baseline comparisons are not fully fair, as they rely on different training and query budgets. In particular, some baselines require additional networks that must be trained separately, resulting in higher computational cost and an unequal evaluation setting.

**Questions:**

1. If the query and training costs are constrained, the benefits of the proposed method become less clear. In fact, some baseline approaches, such as uncertainty-based scoring, offer much more efficient computation for selecting core sets to be labeled, without requiring additional training overhead. This raises the question of whether the added complexity of the proposed method is justified when simpler, low-cost alternatives can achieve comparable results under realistic resource constraints.
2. The large-scale benchmark datasets used in the study exhibit limited diversity, raising concerns about the method’s ability to generalize in broader large-scale training scenarios.

---

> ### Author Response · Authors · 2025-11-21
> **Response to Reviewer mf9U (1/4)**
>
> We sincerly appreciate you for your time and constructive comments which improve our paper. We respectfully address your concerns as following:
>
> ---
>
> > **[Q1]** The method requires additional training and repeated querying to construct the core set. In practical applications, this overhead may limit scalability and reduce its real-world applicability, weakening the overall contribution.
>
> - We sincerely agree that active learning (AL) may appear computationally expensive. However, we respectfully believe this concern might come from a misunderstanding.
>
> - We first emphasize that AL remains one of the most **effective and practical frameworks** in real-world scenarios where **labeled data is expensive to obtain**. Specifically, the main focus of AL is **minimizing human annotation cost**, as its primary goal is to **maximize model performance under a limited labeling budget**.
>
> - More importantly, we clarify that the computational overhead in AL for additional training and repeated querying does **not apply only to our AL selection method ($\texttt{PCoreSet}$)**, but also to **all AL selection baselines evaluated in this paper** (e.g., Random, Entropy, Coreset, Badge, and ClassBalanced) for the following reasons:
>
>     - We first recap our problem setting. As shown in `Fig. 2` and noted in `L206–215` of `§3.1`, we consider a **pool-based AL scenario**, where we are given an unlabeled dataset $\mathcal{D}^{(u)}$ and aim to develop a classifier under a limited annotation budget. Here, the budget refers to the number of samples that can be labeled from human annotator.
>
>     - For example, suppose we aim to train a classifier for fine-grained recognition (e.g., aircraft types [1]). In such settings, unlabeled images are typically **abundant and easily obtained**, while **labels are scarce** and **annotation costs are expensive** (e.g., human expert labeling).
>
>     - In each AL round, we **request annotations** for selected data points (i.e., query samples) under a **predefined labeling budget** (e.g., $C$ samples per round, where $C$ is the number of classes). We then obtain a labeled set $\mathcal{D}^{(l)}$ and train the classifier on $\mathcal{D}^{(l)}$.
>
>     - Under this scenario, the goal of AL selection algorithms is to **select the most informative query samples** to maximize classifier performance under a **limited labeling budget**. Therefore, in AL settings, **all AL selection methods** require repeated querying and retraining across rounds. This requirement is **not specific** to our method.
>
> - Our proposed framework, ActiveKD, follows the **same AL procedure**. The key difference is that during classifier training, we **additionally leverage VLM teachers** (e.g., CLIP [2]) via knowledge distillation. Specifically, as noted in `L229-243`, we train the classifier with both supervised loss (`Eq. 2`) on $\mathcal{D}^{(l)}$ and distillation loss (`Eq. 3`) using predictions from the VLM on $\mathcal{D}^{(u)}$. This cost structure is **consistent with other AL selection algorithms**.
>
> - In summary, the computational overhead of **querying and retraining** is **inherent to the AL paradigm itself** and applies to **all AL selection baselines**, including Random, Entropy, Coreset, Badge, ClassBalanced, and our proposed $\texttt{PCoreSet}$.
>
> ---

---

> ### Author Response · Authors · 2025-11-21
> **Response to Reviewer mf9U (2/4)**
>
> ---
>
> >**[Q2]** The baseline comparisons are not fully fair, as they rely on different training and query budgets. In particular, some baselines require additional networks that must be trained separately, resulting in higher computational cost and an unequal evaluation setting.
>
> - We sincerely appreciate the reviewer for raising this point and apologize for our unclear presentation. However, we respectfully believe that this concern may stem from a misunderstanding.
> - To clarify, we first would like to distinguish the components of our proposed approach as follows:
>
>   1) **ActiveKD** (`§3.2`): a *pool-based* active learning (AL) framework that **leverages VLM teachers** to train a classifier on both labeled data $\mathcal{D}^{(l)}$ and the unlabeled pool $\mathcal{D}^{(u)}$ in each AL round. Specifically, the classifier is trained with supervised loss (`Eq. 2`) on $\mathcal{D}^{(l)}$ and distillation loss (`Eq. 3`) on both $\mathcal{D}^{(l)}$ and $\mathcal{D}^{(u)}$.
>
>   2) **$\texttt{PCoreSet}$** (`§3.3`): an AL *selection strategy* designed for the ActiveKD framework. Any selection method can be used under ActiveKD (e.g., Random, Entropy, Coreset, Badge, and ClassBalanced). We propose $\texttt{PCoreSet}$ to **maximally leverage the structured bias** of VLM teachers for sample selection.
>
> - If one compares ActiveKD against **No Distill**, which corresponds to a conventional AL setup **without VLM teachers** (i.e., `Alg. 1` without the underlined components, as noted in `L215` and `L314`), this may appear **unfair**. However, as shown in `Tab. 1`, ActiveKD consistently provides **large performance gains** over No Distill **across all AL selection baselines** (e.g., +29.07 on average).
>
> - More importantly, under the **same ActiveKD** framework (**even under No Distill**), comparisons between AL selection methods (e.g., $\texttt{PCoreSet}$ vs. Coreset) are **fair**, as all methods are evaluated under the **same training and querying budgets**.
>
> - Furthermore, all methods use the **same training objectives** (`Eq. 2` and `Eq. 3`), the **same hyperparameters** (learning rate, weight decay, etc.), and the **same query budget**, namely, the 1-shot setting where the number of query samples equals the number of classes ($Q=C$), as noted in `L350–351`.
>
> - Accordingly, under the **same ActiveKD** framework, all experimental settings are **identical and fair** across the **different AL selection methods**, with the **only exception being the labeled data** $\mathcal{D}^{(l)}$ that each AL selection method selects and queries for annotation.
>
> - Last but not least, none of the AL selection methods used in this paper (including the proposed $\texttt{PCoreSet}$) require any **additional trainable networks**. The only **trainable component is the classifier** (with a shared backbone, training objectives, and hyperparameters), and the only **additional component is the same VLM teacher** used within the ActiveKD framework.
>
> - In summary, under the **same ActiveKD framework**, **all AL selection methods** are evaluated under **identical conditions**, with the **only difference** being the **labeled data each method selects** for annotation.
>
> - We believe that, together with our response to **[Q1]**, the above discussion clarifies the fairness of our evaluation. We have incorporated the clarification into `L350-351` in `§4.1`.
>
> ---

---

> ### Author Response · Authors · 2025-11-21
> **Response to Reviewer mf9U (3/4)**
>
> ---
>
> >**[Q3]** If the query and training costs are constrained, the benefits of the proposed method become less clear. In fact, some baseline approaches, such as uncertainty-based scoring, offer much more efficient computation for selecting core sets to be labeled, without requiring additional training overhead. This raises the question of whether the added complexity of the proposed method is justified when simpler, low-cost alternatives can achieve comparable results under realistic resource constraints.
>
> - Again, we sincerely appreciate the reviewer for raising this thoughtful concern. Consistent with our response to **[Q2]**, we first kindly note that all AL selection methods considered in this paper (i.e., Random, Entropy, Coreset, Badge, ClassBalanced, and $\texttt{PCoreSet}$) **equally require training a classifier** with the **same query labeling budget** under ActiveKD in each AL round.
>
> - Regarding the **computational efficiency of query selection**, we **have already discussed** this in `L302–305` by comparing the computational complexity of the proposed $\texttt{PCoreSet}$ with that of a **strong and representative baseline** that selects query samples based on feature space (**Coreset**; [3]).
>
> - Specifically, the computational complexity of $\texttt{PCoreSet}$ is  $\mathcal{O}(C \cdot M \cdot N)$, where $C$ is the number of classes, $M = |\mathcal{D}^{(u)}|$ is the number of unlabeled samples, and $N = |\mathcal{D}^{(l)}|$ is the number of labeled samples accumulated up to the current AL round. In contrast, the computational complexity of Coreset is $\mathcal{O}(H \cdot M \cdot N)$, where $H$ denotes the dimensionality of the feature space. Therefore, in common scenarios where $C \ll H$, $\texttt{PCoreSet}$ is **more computationally efficient**.
>
> - For uncertainty-based scoring (i.e., Entropy), the computational complexity is $\mathcal{O}(C \cdot M)$, which is **indeed lower** than that of both $\texttt{PCoreSet}$ and Coreset. However, under ActiveKD, Entropy **underperforms even the simplest baseline** (Random) on ImageNet (e.g., 60.69 vs. 60.43, or 60.49 vs. 58.87), as shown in `Tab. 1`. Furthermore, `Fig. 6` shows that Entropy remains one of the **most underperforming baselines** across the 10 datasets. These results indicate that uncertainty-based AL selection consistently fails to capture informative samples in our settings.
>
> - More crucially, the primary goal of AL scenarios is to maximize performance **under a limited labeling budget**. In such settings, we respectfully argue that **human annotation costs typically outweigh the selection overhead**, making it reasonable to prioritize accuracy under a fixed labeling budget, even if the query step introduces additional computation.
>
> - In summary, the Entropy baseline is computationally more efficient but **less effective**, and we **have already discussed the computational complexity** of the proposed $\texttt{PCoreSet}$ in comparison to Coreset.
>
> - We have included the discussion on the computational complexity of uncertainty-scoring baseline, i.e., Entropy in `L303-304` in `§3.1`.
>
> ---
>
> >**[Q4]** The large-scale benchmark datasets used in the study exhibit limited diversity, raising concerns about the method’s ability to generalize in broader large-scale training scenarios.
>
> - We sincerely appreciate the reviewer for raising this concern. However, as far as we concern, the experimental evaluation in our work is among the **most diverse compared to prior AL literature**.
>
> - Earlier AL work, such as Coreset [3], primarily conducted experiments on CIFAR-10/100 [4] or SVHN [5], which is **not comparable to the scale of our evaluation**.
>
> - More recent AL studies (e.g., [6]) use datasets such as Flowers102, DTD, Oxford Pets, EuroSAT, Caltech101, Stanford Cars, and Aircraft, **all of which are included in our experiments**. In addition, [6] reports results on Food101, SUN397, and UCF101 in its Appendix, and these are **relatively large-scale datasets**, which we also **evaluated on**. Most importantly, we conduct experiments on **ImageNet-1K**, which is rarely used in AL due to its extreme scale (with the exception of [7]).
>
> - We further conduct extensive evaluations across **diverse architectures**:
>   1) **VLM teachers:** CLIP ResNet-50, OpenCLIP ResNet-50, and CLIP ViT-L
>   2) **Student models:** ResNet-18, ResNet-50, ViT-B, TinyViT, and MobileNet
> - These are **non-trivial architectures** and reflect realistic deployment settings.
>
> - With the above diverse architectures, we consider **seven teacher–student pairs** and **73 experimental setups** by accounting for different AL rounds. The proposed $\texttt{PCoreSet}$ achieves the **best performance in 64 out of 73 cases (87.7%)** across these diverse settings.
>
> - We believe the above evidence clearly addresses the concern regarding limited diversity in our experimental evaluation and have incorporated the clarifcation into `L312-313` in `§3.1`.
>
> ---

---

> ### Author Response · Authors · 2025-11-21
> **Response to Reviewer mf9U (4/4)**
>
> ---
>
> ### Reference
>
> [1] Maji, Subhransu, et al. "Fine-grained visual classification of aircraft." arXiv 2013.
>
> [2] Radford, Alec, et al. "Learning transferable visual models from natural language supervision." ICLR 2021.
>
> [3] Sener, Ozan, and Silvio Savarese. "Active learning for convolutional neural networks: A core-set approach." ICLR 2018.
>
> [4] Krizhevsky, Alex, and Geoffrey Hinton. "Learning multiple layers of features from tiny images." (2009): 7.
>
> [5] Netzer, Yuval, et al. "Reading digits in natural images with unsupervised feature learning." NeurIPS Workshop 2011.
>
> [6] Bang, Jihwan, Sumyeong Ahn, and Jae-Gil Lee. "Active prompt learning in vision language models." ICCV 2024.
>
> [7] Emam, Zeyad Ali Sami, et al. "Active learning at the imagenet scale." arXiv 2021.
>
> ---

---

> > ### Author Response · Authors · 2025-11-25
> > **Gentle Reminder**
> >
> > Dear Reviewer mf9U,
> >
> > We sincerely appreciate your time and consideration.
> > We respectfully believe that our response has thoroughly addressed the concerns raised.
> > If you have any remaining concerns or questions, please feel free to contact us and we would be happy to discuss and clarify them.
> >
> > Best,
> >
> > The Authors

---

### Author Response · Authors · 2025-11-21
**General Response**

We sincerely thank all reviewers for their time and constructive feedback, which have significantly improved our paper.

We are especially grateful that the reviewers recognize several strengths of our work, including the **novel integration** of active learning and knowledge distillation (mf9U, hCWn, YbDT), the **clear and well-organized presentation** (mf9U, hCWn, NApS), the **theoretical motivation** and **analysis of structured prediction bias** (hCWn, YbDT), and the **strong and consistent empirical gains** demonstrated across 11 datasets (mf9U, hCWn, YbDT).

We believe that we have successfully addressed all major concerns in this rebuttal. Below, we summarize **the changes** that have been included in the revised version (shown in blue):

---

- `L63-65` in `Fig. 1`: In response to reviewer **NApS** and **YbDT**, we included the discussion on the imperfections of VLM teachers which can easily surpassed by students trained under ActiveKD.

- `L298–300` in `§3.3`: In response to reviewer **hCWn**, we added a discussion explaining how **GT labels** can correct errors arising from the structured prediction bias of VLM teachers.

- `L303-304` in `§3.3`: In response to reviewers **mf9U** and **YbDT**, we added the discussion on the computational complexity of uncertainty-scoring baseline, i.e., Entropy.

- `L312-313` in `§4.1`: In response to reviewer **mf9U**, we clarified the **diversity and scale of the datasets** used in our evaluation.

- `L350-351` in `§4.1`: In response to reviewers **mf9U** and **hCWn**, we clarified the **fairness of our evaluation** among different AL selection methods (e.g., shared backbone, query size $Q$, and hyperparameters).

- `L401–402` in `§4.2`: In response to reviewer **hCWn**, we clarified that the experiments in `Figs. 5 and 6` already isolate the SSL effect by using the same ActiveKD framework.

- `L401–410` in `§4.2`: In response to reviewers **hCWn** and **YbDT**, we clarified that the improvement of $\texttt{PCoreSet}$ (ranking first in 87.7%) is not marginal but **significant**, given that all methods use essentially the same setup except for **data selection**.

- `L401–410` in `§4.2`: In response to reviewer **YbDT**, we added a discussion explaining that the degradation of $\texttt{PCoreSet}$ in the first two AL rounds is **not inconsistent** but **rather expected**.

- In `L484–485` of `§5`, in response to reviewer **NApS**, we clarified that while effective KD and semi-supervised learning methods are **orthogonal** to ActiveKD, investigating their effectiveness within this framework remains future work.

- `Appendix D.5`: In response to reviewers **hCWn** and **NApS**, we added additional experiments using **ActiveKD without unlabeled data** to clarify the importance of unlabeled data under ActiveKD.

---

---

### Author Response · Authors · 2025-11-29
**Letter to AC and Reviewers (2/2)**

---

### Reviewer NApS

> **[Q1]** The reviewer raised concerns about **motivation**, the **necessity of KD**, and **efficient KD**.

- We clarify our motivation by explaining how the **emergence of VLMs enables the integration** of AL and KD (i.e., ActiveKD) and how this reveals the **structured prediction bias** of VLMs under ActiveKD.
- We empirically demonstrate the necessity of KD, showing that **zero-/few-shot VLM teachers underperform compared to a compact student** trained with ActiveKD.
- We clarify the scope of this work and regard the development of more efficient KD methods for ActiveKD as **future work**.


> **[Q2]** The reviewer raised a concern about the absence of **comparison with semi-supervised learnin**g.

- We clarify that SSL focuses on **post-selection** learning, whereas our work focuses on **selecting which samples to label**, so direct comparison is out of scope and SSL is orthogonal.
- Instead, we conduct an ablation study to examine the effect of unlabeled data under ActiveKD, i.e., ActiveKD (w/o unlabeled), and observe that it **significantly improves No Distill** across all selection methods.

---

### Reviewer YbDT

> **[Q1]** The reviewer raised concerns about the **novelty of ActiveKD and $\texttt{PCoreSet}$**.

- We clarify that the **emergence of VLMs is what enables the integration** of AL and KD (i.e., ActiveKD), which has **not been explored in prior work**.
- We also emphasize that our perspective, which shifts the selection criterion from **feature space to probability space**, is **well-motivated** and provides **new insights** that have not been examined in previous AL research.


> **[Q2]** The reviewer raised concerns about **the effect of $\texttt{PCoreSet}$** within ActiveKD, and **the marginal benefit of $\texttt{PCoreSet}$ in early rounds**.

- We clarify that the effect of $\texttt{PCoreSet}$ is already discussed in `§4`:
  - `Tab. 1`: **ActiveKD significantly improves conventional AL** without VLM teachers (No Distill) across all AL selection methods.
  - `Fig. 5`: **Under the same ActiveKD framework**, $\texttt{PCoreSet}$ achieves **the best performance in 64 out of 73 settings (87.7%)**.

- We also clarify that the marginal benefit of $\texttt{PCoreSet}$ in early rounds is an **expected outcome of using imperfect teachers**, as it is designed to **maximally leverage the structured bias**.


> **[Q3]** The reviewer raised concerns about the **simplicity of using diverse outputs** and suggested that the results may be anticipated by injecting a much stronger teacher into the AL loop.

- We clarify that **how diversity is defined** is crucial. Shifting from feature space to **probability space** leads to meaningful differences, as demonstrated by our comparison with Coreset.

- We also clarify that injecting a much stronger teacher into an AL loop is non-trivial under severe label scarcity. For example, a compact student can outperform VLM teachers under ActiveKD, showing that simply using a stronger teacher does not guarantee better performance.

---

> **[Q4]** The reviewer raised a concern about **computational complexity and costs**.

- We clarify that we have **already discussed the computational complexity** of $\texttt{PCoreSet}$ compared to Coreset in `L302–305`.

---

---

### Author Response · Authors · 2025-11-29
**Letter to AC and Reviewers (1/2)**

Dear AC and Reviewers,

We sincerely appreciate your time and efforts for reviewing our paper.
Due to the unfortunate incident during the rebuttal period, ICLR decided to halt the discussion between reviewers and authors and to reassign the corresponding AC. We would like to express our sincere regret that the discussion ended without any responses, which we believe would have helped further clarify misunderstandings and improve our paper.

Accordingly, we **summarize below how we addressed each reviewer’s concerns**, which we believe have been successfully resolved. If the new AC has any remaining concerns or questions raised by the reviewers, we would be happy to address them.

---

### Reviewer mf9U

> **[Q1]** The reviewer raised a concern about **additional training and repeated querying**.

- We clarify that active learning (AL) is **effective and practical** when **labeled data is expensive to obtain**, and all AL selection baselines are **fairly evaluated**.

> **[Q2]** The reviewer raised a concern about **unfair comparisons** due to **different training and query budgets**.

- We clarify that all AL selection baselines are **fairly evaluated** under the same ActiveKD framework, including training objectives, hyperparameters, and query budgets.


> **[Q3]** The reviewer raised a concern about **computational complexity and costs**.

- We clarify that we have **already discussed the computational complexity** of $\texttt{PCoreSet}$ compared to Coreset in `L302–305`.
- We also provide the computational complexity of **uncertainty-based scoring** (Uncertainty), which is more efficient than $\texttt{PCoreSet}$ but **underperforms the simplest baseline** (Random).


> **[Q4]** The reviewer raised a concern about **limited diversity of datasets**.

- We clarify that the datasets used in our paper, including **ImageNet-1K**, are among the **most diverse** compared to prior AL literature.
- We also highlight that our experimental setups include extensive evaluations across **diverse student and teacher architectures**.


### Reviewer hCWn

> **[Q1 & Q3]** The reviewer raised a concern about **erroneous propagation of structured prediction bias**.

- We clarify that this bias is corrected once ground-truth (GT) labels are acquired through AL querying.


> **[Q2]** The reviewer raised a concern about **the effect of $\texttt{PCoreSet}$** within ActiveKD.

- We clarify that this effect is already discussed in `§4`:
  - `Tab. 1`: **ActiveKD significantly improves conventional AL** without VLM teachers (No Distill) across all AL selection methods.
  - `Fig. 5`: **Under the same ActiveKD framework**, $\texttt{PCoreSet}$ achieves **the best performance in 64 out of 73 settings (87.7%)**.

> **[Q4]** The reviewer demanded an ablation that applies **KD only to labled data**.

- We apply KD only to labeled samples, i.e., **ActiveKD (w/o unlabeled)**, and observe that it **significantly improves No Distill** across all selection methods.
- We also show that $\texttt{PCoreSet}$ **achieves the best performance in most cases** under ActiveKD (w/o unlabeled).


---

---

### Meta-Review · Area_Chair_Q7S3 · 2025-12-26

**Summary:**

This paper proposes ActiveKD, which integrates active learning and knowledge distillation using a zero- and few-shot VLM as the teacher. It also introduces PCoreSet, a probability-space acquisition strategy that promotes diversity and coverage in the teacher's prediction space. Reviewers acknowledged the strong and consistent empirical gains, but raised concerns about the limited novelty beyond a natural AL+KD combination, the extent to which the gains are driven by the strong VLM teacher and the distillation on the unlabeled pool (SSL-like effect) rather than the proposed selection method, and practical considerations such as computational cost and scalability.
While empirically effective, the work does not yet convincingly establish sufficient standalone novelty for the proposed selection strategy beyond combining existing components. Based on these considerations, I recommend Reject.
We appreciate the authors’ efforts in the rebuttal and revision, and encourage them to further refine the paper in line with the reviewers’ feedback.

**Reviewer Concerns:**

The rebuttal clarified evaluation fairness and the role of structured prediction bias, and added an ablation applying KD only to labeled data, which partially isolates the effect of unlabeled distillation. However, a systematic comparison with strong modern SSL methods is not provided (the authors argue it is out of scope / orthogonal), and skepticism remains regarding whether the main improvements stem from the proposed selection strategy or primarily from introducing a powerful VLM teacher.

**Reviewer Scores:**

This submission received initial scores of 2, 4, 4, and 6. Given the initial average and that several key concerns remain partially unresolved after the rebuttal, I recommend Reject.

---

### Decision · Program_Chairs · 2026-01-26

Reject